# Chromosome structure in *Drosophila* is determined by boundary pairing not loop extrusion

Xinyang Bing[1†‡], Wenfan Ke[2†], Miki Fujioka[3†], Amina Kurbidaeva[2§], Sarah Levitt[2], Mike Levine[1], Paul Schedl[2*], James B Jaynes[3]

[1]Lewis Sigler Institute, Princeton University, Princeton, United States; [2]Department of Molecular Biology, Princeton University, Princeton, United States; [3]Department of Biochemistry and Molecular Biology, Thomas Jefferson University, Philadelphia, United States

**\*For correspondence:**
pschedl@princeton.edu

[†]These authors contributed equally to this work

**Present address:** [‡]BlueRock Therapeutics, Cambridge, United Kingdom; [§]Center for Genomics and Systems Biology, New York, United States

**Abstract** Two different models have been proposed to explain how the endpoints of chromatin looped domains ('TADs') in eukaryotic chromosomes are determined. In the first, a cohesin complex extrudes a loop until it encounters a boundary element roadblock, generating a stem-loop. In this model, boundaries are functionally autonomous: they have an intrinsic ability to halt the movement of incoming cohesin complexes that is independent of the properties of neighboring boundaries. In the second, loops are generated by boundary:boundary pairing. In this model, boundaries are functionally non-autonomous, and their ability to form a loop depends upon how well they match with their neighbors. Moreover, unlike the loop-extrusion model, pairing interactions can generate both stem-loops and circle-loops. We have used a combination of MicroC to analyze how TADs are organized, and experimental manipulations of the *even skipped* TAD boundary, *homie*, to test the predictions of the 'loop-extrusion' and the 'boundary-pairing' models. Our findings are incompatible with the loop-extrusion model, and instead suggest that the endpoints of TADs in flies are determined by a mechanism in which boundary elements physically pair with their partners, either head-to-head or head-to-tail, with varying degrees of specificity. Although our experiments do not address how partners find each other, the mechanism is unlikely to require loop extrusion.

## eLife assessment

This **valuable** work presents elegant experimental data from the *Drosophila* embryo supporting the notion that interactions among specific loci, called boundary elements, contribute to topologically associated domain (TAD) formation and gene regulation. The evidence supporting boundary:boundary pairing as a determinant of 3D structures is **compelling**; however, an inability to deplete loop extruders formally leaves open a possible contribution of loop extrusion. This study will be of interest to the nuclear structure community, particularly those using *Drosophila* as a model.

## Introduction

The chromosomes of multicellular animals have a regular and inheritable physical organization. This was first recognized in studies on the lampbrush chromosomes in amphibian oocytes and the polytene chromosomes of insects (*Callan, 1963*; *Nora et al., 2020*; *Zhimulev and Belyaeva, 1991*; *Zhimulev et al., 1983*). Subsequent research has shown that the architectural features inferred from analysis of lampbrush and polytene chromosomes are present in chromosomes throughout much of the animal kingdom. The key organizational principle is the subdivision of the chromatin fiber into a

series of independent looped domains, commonly called 'TADs' (*Cavalheiro et al., 2021*; *Chetverina et al., 2017*; *Jerković et al., 2020*; *Matthews and White, 2019*; *Rowley and Corces, 2018*). With important exceptions, the arrangement of TADs along a given chromosome tend to be invariant and are largely independent of the cell type or developmental stage. This regular and inheritable organization is a reflection of the underlying mechanism of TAD formation. TADs are separated from each other by special elements called boundaries or insulators. While these elements have been found in many different species, they have been most fully characterized in *Drosophila* (*Cavalheiro et al., 2021*; *Chetverina et al., 2017*). Fly boundaries span DNA sequences of 150 bp to 1.5 kb in length and contain one or more nucleosome-free nuclease-hypersensitive regions. These nuclease-hypersensitive regions are targets for a large collection of DNA binding proteins that have been implicated in boundary function. Some of the fly boundary factors are widely conserved (CTCF, BEN family proteins, GAF), while others appear to be restricted to insect linages (Su(Hw), Pita, Zw5, Zipic, Mod(mdg4), BEAF; *Heger et al., 2013*; *Heger and Wiehe, 2014*; *Schoborg and Labrador, 2010*).

Boundary elements in flies are not only responsible for organizing the chromatin fiber, they also have genetic activities. When interposed between enhancers or silencers and target promoters, boundary elements block regulatory interactions (*Bell et al., 2001*; *Chetverina et al., 2014*; *Chetverina et al., 2017*). This insulating activity provides a mechanism for delimiting units of independent gene activity: genes located between a pair of compatible boundaries are subject to regulatory interactions with enhancers/silencers present in the same chromosomal interval, while they are insulated from the effects of enhancers/silencers located beyond either boundary in adjacent neighborhoods. Genetic studies suggest that the insulating activity of boundary elements is a consequence of subdividing the chromosome into a series of topologically independent domains (*Cai and Shen, 2001*; *Gohl et al., 2011*; *Muravyova et al., 2001*). Organizing the chromatin fiber into looped domains enhances contacts between sequences within the loop, while it suppresses contacts with sequences outside of the loop. While it is not known whether a similar insulation mechanism is at play in mammals, the fact that mammalian chromosomes are also subdivided into TADs by boundary-like elements suggests that it might be.

A critical question in chromosome biology is the mechanism(s) responsible for determining the endpoints of the loop domains, the TADs. In mammals, a novel loop-extrusion model has been proposed to not only generate chromosomal loops but also determine the endpoints of those loops (*Alipour and Marko, 2012*; *Fudenberg et al., 2016*; *Guo et al., 2012*; *Guo et al., 2015*; *Nuebler et al., 2018*; *Sanborn et al., 2015*). In this model, a cohesin complex initiates loop formation at a loading site within the 'loop-to-be' and then extrudes a chromatin loop until it bumps into CTCF-dependent roadblocks, one on each side of the extruding loop. The location of these roadblocks combined with the processive action of the cohesin complex determines the endpoints of each TAD (*Davidson and Peters, 2021*; *Ghosh and Meyer, 2021*; *Mirny and Dekker, 2022*; *Perea-Resa et al., 2021*). A key assumption of the loop-extrusion model is that mammalian boundaries are fully autonomous: they are roadblocks, and their physical presence in and of itself is sufficient to define the loop endpoint, independent of the functional properties of neighboring boundaries. In a more specific variant of this model, the relative orientation of the paired CTCF roadblocks is also important—in order to halt cohesin-mediated extrusion, the CTCF sites on each side of the loop must have a convergent orientation. In this model, the CTCF roadblocks are able to block an oncoming cohesin complex in only one direction; however, their intrinsic blocking activity is independent of the orientation of other CTCF sites in their neighborhood.

While this loop-extrusion mechanism is widely thought to be operative in mammals, the evidence regarding TAD formation and function in flies is seemingly inconsistent with this mechanism. Genetics studies have shown that fly boundaries are functionally non-autonomous and that their activities in both loop formation and gene regulation depend upon their ability to engage in direct physical interactions with other boundaries (*Chetverina et al., 2014*; *Chetverina et al., 2017*). That fly boundaries might function by physical pairing was first suggested by studies which showed that the *gypsy* transposon *su(Hw)* boundary and the bithorax complex (BX-C) *Mcp* boundary can mediate regulatory interactions (enhancer/silencer: reporter) between transgenes inserted at sites separated by 15–32 Mb or more (*Muller et al., 1999*; *Sigrist and Pirrotta, 1997*; *Vazquez et al., 1993*). Consistent with direct physical interactions, these distant transgenes co-localize *in vivo*, and in living tissue remain in contact for extended periods of time (*Chen et al., 2018*; *Li et al., 2011*; *Vazquez et al., 2006*).

Some of the parameters that govern pairing interactions have been defined in two different transgene assays, insulator bypass and boundary competition. In one version of the bypass assay, a set of enhancers are placed upstream of two different reporters (*Cai and Shen, 2001*; *Kyrchanova et al., 2008a*; *Muravyova et al., 2001*). When a single boundary is introduced between the enhancers and the closest reporter, the enhancers are unable to activate either reporter. However, if the same boundary is placed downstream of the closest reporter, bypass is observed. In this case the closest reporter, which is bracketed by the two boundaries, is still insulated from the enhancers; however, the downstream reporter is activated (*Muravyova et al., 2001*). *Kyrchanova et al., 2008a* showed that bypass activity is, in most cases, orientation-dependent. When the same boundary is placed in the upstream and downstream position, they typically need to be introduced in the opposite orientation. The reason for this is that self-pairing is head-to-head, and this configuration generates a stem-loop topology, bringing the upstream enhancers into close proximity to the downstream reporter. When the two boundaries are introduced in the same orientation, the enhancers do not activate the downstream reporter because a circle-loop rather than a stem-loop is formed (*Chetverina et al., 2017*; *Kyrchanova et al., 2008a*). Head-to-head pairing seems to be a common feature of self-pairing interactions between fly boundaries and was observed for *scs*, *scs'*, *iA2* and *wari* (*Kyrchanova et al., 2008a*). A preference for head-to-head interactions is also observed for heterologous pairing interactions between different boundaries in the *Abdominal-B* (*Abd-B*) region of the BX-C (*Kyrchanova et al., 2011*; *Kyrchanova et al., 2008b*). There are some exceptions. The BX-C *Fab-7* boundary pairs with itself and with its neighbor *Fab-8* in both orientations (*Kyrchanova et al., 2011*; *Kyrchanova et al., 2008b*). Bypass activity also seems to depend upon finding the proper match. For example, when multimerized dCTCF or Zw5 binding sites are placed both upstream and downstream of the closest reporter, bypass is observed. However, there is no bypass when multimerized dCTCF sites are tested with multimerized Zw5 sites (*Kyrchanova et al., 2008a*). Similar results have been reported for the Zipic and Pita factors (*Zolotarev et al., 2016*). Boundary:boundary pairing preferences are also observed in competition assays in which different boundaries are introduced into the same transgene (*Gohl et al., 2011*). In these assays, the insulating activity of a boundary in the blocking position (between an enhancer and a reporter that is flanked by a 3' boundary) is challenged by placing a heterologous boundary upstream of the enhancer. If the boundary upstream of the enhancer is a better match for the boundary located downstream of the reporter, then the insulating activity of the boundary in the blocking position can be compromised or lost altogether. These and other experiments argue that fly boundaries are functionally non-autonomous—that is, their activities depend upon their ability to physically interact with other boundaries.

However, the differences between the two models are not limited to the mechanisms for determining the endpoints of TADs. The models also differ in the possible topologies of the chromatin loops that form TADs. Only two loop topologies can be generated by the cohesin-extrusion mechanism. One is a stem-loop while the other is an unanchored loop. Depending upon the location of the roadblocks that halt cohesin extrusion and the cohesin loading sites, the chromatin fiber would be organized into a series of stem-loops. These stem-loops will be separated from each other by unanchored loops (see *Figure 1B*). This is the configuration that is often illustrated in articles discussing the loop-extrusion model. If the relevant roadblocks are very closely juxtaposed, then the unanchored connecting loops will disappear and be replaced by a series of stem-loops that point in opposite orientations, as illustrated in *Figure 1C*. Stem-loops are also possible in the boundary-pairing model, and they will be formed when two heterologous boundaries pair with each other head-to-tail in *cis* (see *Figure 1D*). Like loop-extrusion, the stem-loops could be separated from each other by unanchored loops (*Figure 1E*). If the boundaries in the neighborhood pair with each other head-to-tail, a series of connected stem-loops pointing in opposite orientations will be generated. In this case, the main axis of the chromosome would be a series of paired boundaries (*Figure 1F*).

Stem-loops are not, however, the only loop topology that can be generated in the boundary-pairing model. If the neighboring boundaries pair with each other head-to-head instead of head-to-tail, a circle-loop structure will be generated (*Figure 1G*). Like stem-loops, the circle-loops could be connected by unanchored loops (*Figure 1H*). Alternatively, if boundaries pair with both neighbors head-to-head, this will generate a series of linked circle-loops oriented in (more or less) the same direction (see *Figure 1I*).

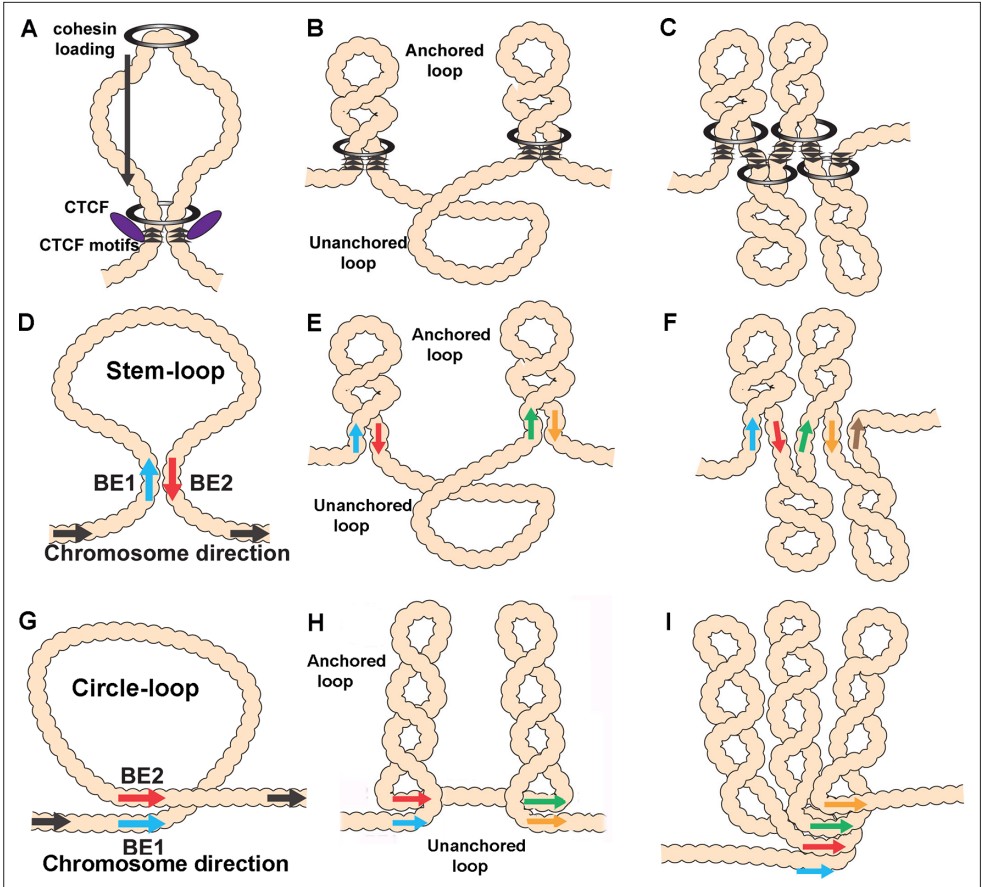

**Figure 1.** Diagrams of possible loop topologies generated by loop extrusion and boundary:boundary pairing. (**A**) Cohesin embraces a loop at a loading site somewhere within the TAD-to-be, and then extrudes a loop at an equal rate on both strands. Extrusion continues until cohesin encounters boundary roadblocks on both strands. (**B**) In one model, the orientation of the roadblock is important. As a consequence, the chromatin fiber will be organized into a series of stem-loops separated from each other by unanchored loops. (**C**) If the presence of a boundary roadblock, but not its orientation, is sufficient to halt extrusion, the chromatin fiber will be organized into a series of linked stem-loops. The main axis of the chromosome will be defined by the cohesin:CTCF roadblocks. (**D**) The pairing of two boundaries head-to-tail generates a stem-loop. (**E**) If boundary pairing interactions are strictly pairwise, head-to-tail pairing will generate a series of stem-loops separated from each other by unanchored loops. (**F**) If boundaries can engage in multiple head-to-tail pairing interactions, the chromosome will be organized into a series of linked stem-loops. The main axis of the chromosome will be defined by a series of paired boundaries. (**G**) The pairing of two boundaries head-to-head generates a circle-loop. (**H**) If boundary pairing interactions are strictly pairwise, there will be a series of circle-loops separated from each other by unanchored loops. (**I**) If boundaries can engage in multiple head-to-head pairing interactions, the chromatin fiber will be organized into a series of circle-loops connected to each other at their base. Pairing interactions between boundaries #1 and #2 need not be in register with the pairing of boundaries #2 and #3. In this case, the main axis of the chromosome may bend and twist, and this could impact the relative orientations of the circle-loops.

In the studies reported here, we have critically evaluated these two models. We have used a combination of MicroC to analyze how TADs are organized and experimental manipulations to test the predictions of the 'loop-extrusion' and the 'boundary-pairing' models. For these experimental manipulations, we have used the well-characterized *homie* boundary from the *even skipped* (*eve*) locus. *homie* together with *nhomie* flank the 16 kb *eve* locus, and these two elements share many of the properties of other fly boundaries (*Fujioka et al., 2013*; *Fujioka et al., 2009*). Like the *gypsy* and *Mcp* boundaries, *nhomie* and *homie* can mediate long-distance regulatory interactions. For example, a *lacZ* reporter transgene containing *homie* (or *nhomie*) inserted on one homolog can be activated by an *eve* enhancer in a *homie-* (or *nhomie-*) containing transgene inserted on the other homolog, over a distance of 2 Mb (*Fujioka et al., 2016*). Long-distance activation is also observed in heterologous

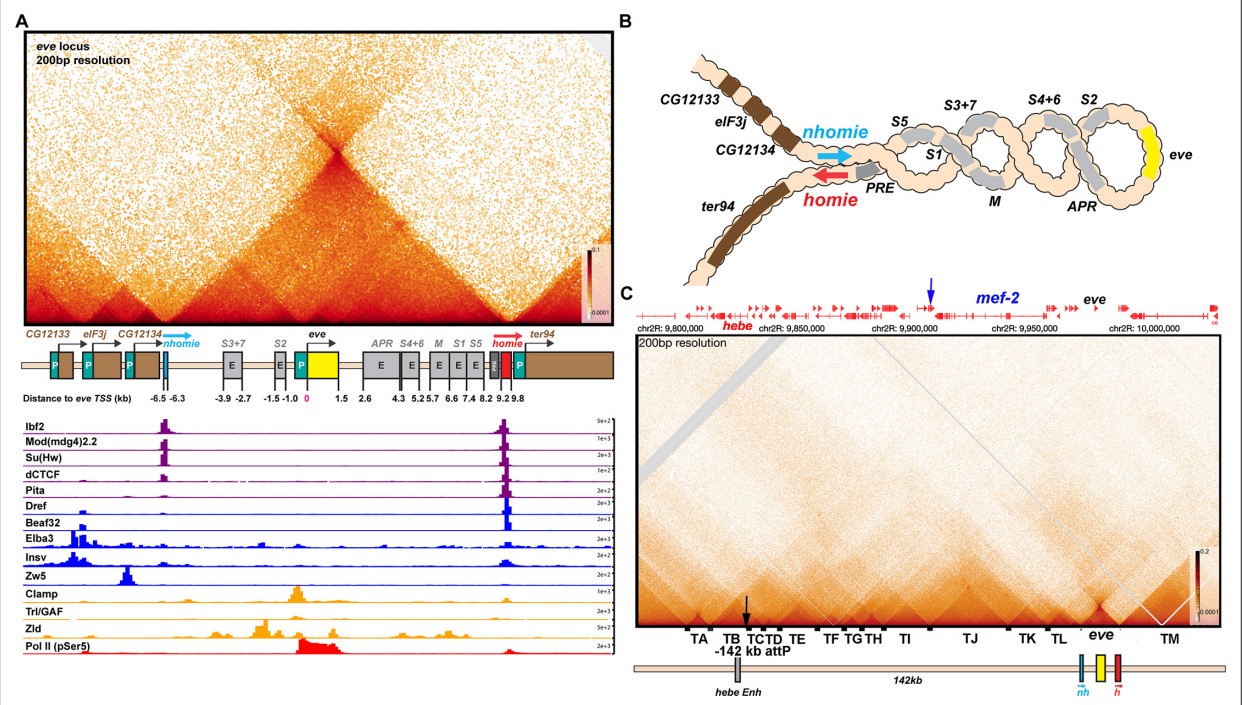

**Figure 2.** TAD organization of the *even-skipped* locus and neighboring sequences. (**A**) The *eve* TAD is a volcano with a plume that is anchored by *nhomie* and *homie*. ChIP-seq data below the MicroC map indicate that many of the known fly chromosomal architectural proteins are associated with the two *eve* boundary elements *in vivo* (**Cuartero et al., 2014**; **Duan et al., 2021**; **Gaskill et al., 2021**; **Li et al., 2015**; **Sun et al., 2015**; **Ueberschär et al., 2019**; **Zolotarev et al., 2016**). (**B**) The *eve* locus forms a stem-loop structure. In this illustration, *nhomie* pairs with *homie* head-to-tail, and this forms a stem-loop that brings sequences upstream of *nhomie* and downstream of *homie* into contact, as is observed in insulator bypass assays (**Kyrchanova et al., 2008a**). The *eve* locus is shown assembled into a coiled '30 nM' chromatin fiber. (**C**) MicroC contact pattern for the chromosomal region spanning the attP site at −142 kb and the *eve* locus. Like the *eve* volcano, this contact pattern was generated using aggregated previously published NC14 data (**Batut et al., 2022**; **Levo et al., 2022**). A black arrow indicates the −142 kb locus where transgenes are integrated into the genome. A blue arrow indicates the position of the *Etf-QO* gene. Note the numerous TADs between the −142 kb *hebe* locus from the *eve* 'volcano TAD'. Individual TADs are labeled TA—TM. Black boxes indicate positions of sequences that, based on ChIP experiments, are bound by one or more known insulator proteins *in vivo*.

The online version of this article includes the following figure supplement(s) for figure 2:

**Figure supplement 1.** TAD organization of the *even-skipped* gene and upstream region.

**Figure supplement 2.** Trouble ahead, trouble behind.

**Figure supplement 3.** Models for the formation of LDC domains.

combinations of *nhomie* and *homie*, but not in combinations with phage *lambda* DNA. Similarly, regulatory interactions are observed when a *homie*- or *nhomie*-containing reporter is inserted at an attP landing site in the *hebe* gene, 142 kb upstream of the *eve* locus (**Chen et al., 2018**; **Fujioka et al., 2016**; **Fujioka et al., 2009**). In this case, the *eve* enhancers drive reporter expression during development in a pattern that mimics the stage- and tissue-specific expression of the endogenous *eve* gene. Activation also depends upon the orientation of the boundary (5'→3') relative to the reporter in the transgene. For *homie*, the reporter must be located 'upstream' of the boundary (5'→3'), just like the *eve* enhancers (and *eve* gene) are 'upstream' of endogenous *homie* (see **Figure 2B**). Little or no activation is observed when the relative orientation is reversed, so that the reporter is located 'downstream' of *homie*. For *nhomie*, the reporter must be located 'downstream', in the same relative position as that in which the *eve* enhancers and *eve* gene are located with respect to endogenous *nhomie* (5'→3': **Figure 2B**).

# Results

## The *eve* TAD

We used MicroC to probe the TAD organization of the *eve* locus and the genomic region that extends from *eve* to just beyond the attP site at –142 kb. For this purpose, we chose nuclear cycle (NC) 14 embryos. DNA replication commences soon after nuclei exit the previous mitosis, and continues for nearly an hour until the nuclei arrest in G2, and the process of cellularization is completed. As sister chromosomes in flies pair with each other in register, there will be two juxtaposed copies of this region of the chromosome in most nuclei. The sister chromosomes should be linked to each other by cohesin in preparation for the asynchronous cellular divisions at the end of NC14 (**Collier and Nasmyth, 2022**; **Gligoris et al., 2014**). While homolog pairing is limited in pre-cellular blastoderm embryos, there are regions of the chromosome that already show evidence of pairing (**Erceg et al., 2019**; **Fung et al., 1998**). In fact, transvection mediated by transgenes containing the *gypsy* insulator and the *homie* and *Fab-8* boundaries has been observed in live imaging studies of nuclear cycle 14 embryos (**Lim et al., 2018**). Since the NC14 nuclei are just emerging from an earlier mitosis and are undergoing DNA synthesis, one would imagine that other key features of chromosome organization might also be in the process of being established. Thus, it should be possible to capture evidence of intermediates like those expected to be generated by extruding cohesin complexes as they move through the *eve* TAD and other nearby loci at this stage of development.

Shown in **Figure 2A** is a blow-up of the TAD organization of the *eve* locus. As expected, the left and right boundaries of the *eve* TAD correspond to *nhomie* and *homie*, respectively, and sequences containing these two boundaries converge at the apex of the *eve* interaction triangle. All the sequences within the TAD appear to come into contact with each other and define a high density of internal contact domain (HDIC). These contacts are expected to be generated, at least in part, by the sliding of the chromatin fiber within the TAD against itself, much like sequences in circular/supercoiled DNA can interact with each other (**Figure 1E**). There are also sequences within the *eve* TAD that show enhanced interactions. For example, there is a darker interaction triangle connecting *nhomie* and the promoter region, while another triangle connects *homie* and the PRE adjacent to *homie* to the promoter region. These enhanced internal contacts would be expected to help insulate the *eve* gene from external regulatory elements, and at the same time facilitate interactions between *eve* and its upstream and downstream enhancers.

As predicted by both the loop-extrusion and boundary-pairing models, the *eve* TAD is a stem-loop, not a circle-loop (**Figure 2B**). This is evident from the prominent 'volcano plume' at the apex of the *eve* TAD (volcano triangle). A volcano plume is observed because sequences flanking *eve* are brought into close proximity when *homie* and *nhomie* are linked to each other by either cohesin or by head-to-tail pairing (see **Figure 1**: see also **Ke et al., 2024**). **Figure 2—figure supplement 1** shows the hierarchy of interactions between TADs to either side of *eve* (L-M, K-M, and J-M in **Figure 2—figure supplement 1**). This conclusion is supported by previous transgene studies. As first described in the boundary bypass experiments of **Cai and Shen, 2001** and **Muravyova et al., 2001**, the formation of a stem-loop structure when two boundaries are linked together brings sequences immediately beyond the linked boundaries into close proximity with each other. In this assay, a stem-loop structure is required, as it enables enhancers flanking the 'upstream' boundary to interact with reporters flanking the 'downstream' boundary (**Kyrchanova et al., 2008a**). In contrast, a circle-loop configuration does not bring the neighbors to either side of *eve* into close proximity.

While a stem-loop is the topology expected for the *eve* TAD in both models, one feature of the TAD is inconsistent with the loop-extrusion model. According to this model, loops (TADs) are generated by a cohesin complex that forms a small bubble at a loading site somewhere in the TAD-to-be and then extrudes both strands of the growing loop at an equal rate until it bumps into the boundary roadblocks (**Davidson and Peters, 2021**; **Ghosh and Meyer, 2021**; **Mirny and Dekker, 2022**; **Perea-Resa et al., 2021**). If the loading site were located in the middle of the TAD-to-be, and the extrusion intermediates in a population of nuclei were captured by crosslinking, one should observe a crosslink-generated stripe that begins at the loading site and extends perpendicular to the chromosomal DNA until it reaches the apex that links the two TAD boundaries with imaginary lines of 45° (**Figure 2—figure supplement 2B**). This is the pattern that would be generated by a 'broken zipper' when it transiently links the zipper teeth equidistant from the initial 'loading site' (**Figure 2—figure supplement 2A**). If the loading site is off center to the left, the crosslinking of the passing cohesin complex

in a population of nuclei will generate a perpendicular stripe until cohesin come to a halt when it encounters the roadblock on the left (*Figure 2—figure supplement 2C*). Assuming that the complex continues to extrude the right strand, it will then generate a stripe at a 45° angle that extends to the apex of the TAD. If the loading site corresponds to one of the CTCF roadblocks or is located close to the roadblock, the cohesin complex will generate a 45° stripe that comes to a halt at the apex of the triangle when it encounters the neighboring CTCF roadblock (*Figure 2—figure supplement 2D and E*). If there are multiple loading sites within the TAD, there should be a series of perpendicular stripes that generate and sequentially reinforce the stripe(s) at 45° (*Figure 2—figure supplement 2F*). However, in spite of the fact that NC14 fly embryos should provide by far the best prospect of actually capturing loop extrusion intermediates, there are no vertical stripes in the *eve* TAD, nor are there stripes at 45° that extend to the apex of the *eve* TAD. Nor do we observe a population of perpendicular stripes that convert into a series of reinforced 45° stripes. There is also no evidence of stripes that are initially tilted to the left or right of a 'loading site', as might be observed if the rate of extrusion were unequal on the two strands, and then subsequently convert to a 45° stripe as cohesin comes to a halt at the first boundary encountered.

## TADs in the *eve* environs

The attP insertion site at –142 kb is near the end of the first exon of the *hebe* transcription unit (*Figure 2C*). It is located just within a ~14 kb TAD, TB, that has a high density of internal contacts, and extends from the *hebe* promoter to the 3' end of the neighboring *dila* gene. In between *hebe* and the *eve* locus, there are at least 10 distinct TADs, TC-TL (*Figure 2C*). The endpoints of most of these TADs correspond to sequences that are associated with one or more known chromosomal architectural factors (dots along the horizontal axis in *Figure 2C*).

These TADs correspond to the fundamental building blocks for organizing the 3D architecture of this chromosomal segment. Superimposed upon these TADs are regions that exhibit lower density of contacts (LDC). For example, the *mef2* gene spans two TADs, TJ and TK. TK contains the *mef2* distal promoter region and extends to the internal *mef2* promoter, while TJ extends to near the promoter of the divergently transcribed *Etf-QO* gene (blue arrow). Both of these TADs are linked together by a rectangular LDC domain, J-K (see *Figure 2—figure supplement 1*). Likewise, TJ is linked to its immediate neighbor TI by the LDC domain I-J. In the next level, TADs separated by a single TAD are linked together. In the region immediately above TJ, the LDC domain I-K links TI to TK (*Figure 2—figure supplement 1*). Similarly, TJ is linked to TL by J-L, while TJ is also linked to the large TAD TM (which contains *TER94* and the neighboring gene, *pka-R2*) on the other side of *eve* by J-M. The same pattern of a hierarchical series of LDC domains linking TAD neighbors, TAD next-door neighbors and next-next-door neighbors is observed in the DNA segment that includes the *hebe* gene.

### Cohesin-mediated loop extrusion

Like *eve*, the TADs in the region between *eve* and the attP site in the *hebe* locus are defined by right-angle triangles (volcanos) with a high density of internal contacts. However, unlike *eve*, these volcano triangles do not have a plume, and instead are surrounded by a series of LDC rectangular 'clouds'. This is not the contact pattern that one might expect for either a series of stem-loops connected by unanchored loops or a series of stem-loops connected to each other. Moreover, there are no perpendicular or angled crosslinking stripes (broken zippers) emanating from the base of these triangles, nor are their stripes along their 45° legs. Thus, in spite of the fact that the NC14 nuclei used in this analysis are just emerging from an earlier mitosis and a round of DNA synthesis, and should be in the process of assembling TADs, the crosslinking signatures that should be generated by the embrace of passing cohesin complexes are completely absent.

In the loop-extrusion model, the LDC domains would arise in a subset of nuclei because the extruding cohesin complex breaks through one or more boundary roadblocks before its progression is halted (*Figure 2—figure supplement 3A* versus B, C, and D; *Hsieh et al., 2020*; *Krietenstein et al., 2020*). Breakthroughs could occur at only one roadblock on one side of the extruding loop (*Figure 2—figure supplement 3B, C*), or at multiple roadblocks on one or both sides (*Figure 2—figure supplement 3D*), giving rise to a set of overlying LDC domains like that evident in the region between *hebe* and *eve*. However, while there would have to be multiple breakthrough events to account for the hierarchical array of LDC domains observed in these NC14 embryos, there is no

evidence of the 45° crosslinking stripes that would be expected to mark the legs of the LDC domains following the breakthrough. For example, the I-J LDC domain (*Figure 2—figure supplement 1*) could be generated by cohesin complexes that initiated extrusion in either TI or TJ and then failed to halt at the TI:TJ boundary (green arrowhead). In either case, there should be a 45° stripe of crosslinking that marks (at least part of) the left (TI) and/or the right (TJ) leg of the I-J LDC domain and extends to the apex of the TI-TJ LDC triangle; however, there is no evidence of such stripes. Nor is there evidence of crosslinking stripes marking one or both legs of the J-K or I-K LDC domains (*Figure 2—figure supplement 1*).

While the LDC domains associated with the TADs located between *eve* and the *hebe* locus could potentially be explained by cohesin breaking through one or more boundaries (even though we do not observe the expected stripe intermediates), it is important to note that the *eve* boundaries, *nhomie* and *homie,* do not appear to be subject to as frequent break-through events. This is because, unlike the other TADs in the neighborhood, the LDC domains that link *eve* to TA and TL are much more lightly populated compared to the LDC domain that links TA to TL.

## Boundary:boundary pairing

At least two different mechanisms are expected to account for the TADs and LDC domains in the region between *eve* and the attP insertion site at –142 kb. First, since none of the TADs in this DNA segment (with the possible exception of TA) are topped by volcano plumes, they could correspond to circle-loops. As circle-loops are generated by head-to-head pairing, the boundaries in the region between *nhomie* and *hebe* would be expected to pair with each other head-to-head. (The TAD-to-TAD contact maps generated by circle-loops and stem-loops are considered further in the accompanying paper, *Ke et al., 2024*).

As illustrated in *Figure 2—figure supplement 3E–G*, the coiled circle-loop TADs, though topologically independent, are expected to be in relatively close proximity to each other. This means that in addition to crosslinking events within the TADs, there will be crosslinking events between neighboring TADs. These crosslinking events could generate the LDCs seen in *Figure 2C* and *Figure 2—figure supplement 1*. Since TADs next to each other (*Figure 2—figure supplement 3E, F*; *Figure 2—figure supplement 1I–J or J–K*) would typically be expected to interact more frequently than TADs separated by one, two, or more TADs (*Figure 2—figure supplement 3G*; *Figure 2—figure supplement 1I–K or J–L*), there should be a progressive reduction in contact frequency at each step. This is generally what is seen (see *Figure 2—figure supplement 1*). While connected stem-loops generated by loop extrusion should also bump into one another, contacts between next-door neighbors might be expected to be less frequent than contacts between next-next-door neighbors (*Figure 1B and C* and *Figure 2—figure supplement 3A*). (TAD-to-TAD crosslinking and the impact of loop topology is further examined in *Ke et al., 2024*).

The second mechanism for generating LDC domains would be switching and/or combining pairing partners (*Figure 2—figure supplement 3H–J*). Though imaging studies suggest that pairing interactions can be of relatively long duration (>30 min: *Chen et al., 2018*; *Vazquez et al., 2006*), they are not permanent, and other nearby boundaries can compete for pairing interactions (*Gohl et al., 2011*). If switching/combining occurs in NC14 embryos (see *Figure 2—figure supplement 3H–J*), then the precise pattern of TADs and LDC domains could also be impacted by the relative avidity of potential pairing partners in the neighborhood, and also the distances separating potential partners. For example, the rectangle linking TH and TI (H-I in *Figure 2—figure supplement 1*) could represent partner switching, so that the boundary between TI and TJ (green arrowhead in *Figure 2—figure supplement 1*) sometimes pairs with the boundary separating TH and TI (blue arrowhead in *Figure 2—figure supplement 1*) and sometimes with the boundary separating TG and TH (red arrowhead in *Figure 2—figure supplement 1*). Distant boundary elements could also occasionally interact with each other, generating a supra-TAD that contains multiple TAD domains. For example, at the apex of the LDC domain B-I, there is an interaction dot (small purple arrow, *Figure 2—figure supplement 1*) that links the TA-TB boundary to the TI-TJ boundary (green arrowhead).

## Activation of a distant reporter by the *eve* enhancers

In previous studies, we found that a minimal 367 bp *homie* fragment can orchestrate regulatory interactions between the *eve* enhancers and reporters inserted at an attP site in the first intron of the *hebe*

transcription unit, 142 kb upstream of the *eve* gene (*Fujioka et al., 2016*). This minimal fragment lacks the one potential *homie* CTCF site, but has sites for several other generic boundary proteins, including BEAF, Zipic, Su(Hw), Pita, and Ibf2. In blastoderm stage embryos, *lacZ* reporter expression was observed in a 7-stripe pattern that coincided with the stripes of the endogenous gene. However, at that stage only a subset of *eve*-expressing nuclei also expressed the reporter. Later in development, during mid-embryogenesis, reporter expression was observed in cells in the dorsal mesoderm, the ventral CNS, and in the anal plate region, in a pattern that recapitulated *eve* gene expression (*Fujioka et al., 2016*). At these later stages, most of the *eve*-expressing cells also appear to express the reporter. The *eve*-dependent activation of the reporter in a stripe pattern has also been visualized in live imaging of pre-cellular blastoderm-stage embryos (*Chen et al., 2018*). Within the 7 *eve* stripes, the majority of the nuclei (~80–85%) did not express the *lacZ* reporter. In these nuclei, the transgene was found to be at a considerable average physical distance from the *eve* locus (~700 nM). In the remaining nuclei, the transgene was located in closer proximity to the *eve* locus (~330 nM, on average). Within this subset, the transgene was expressed in most of the nuclei. Moreover, in most cases in which the transgene was active, it remained in close proximity and continued to express *lacZ* for the duration of the experiment (~30 min).

In the two models for TAD formation, quite different mechanisms must be invoked to account for activation of the reporter at –142 kb by the *eve* enhancers. In the boundary-pairing model, the transgene *homie* boundary at –142 kb loops over the intervening TADs and pairs with the *nhomie:homie* complex flanking the *eve* TAD (c.f., *Figure 3B and C*). In the loop-extrusion model, a cohesin complex initiating loop extrusion in the *eve* TAD must break through the *nhomie* roadblock at the upstream end of the *eve* TAD. It must then make its way past the boundaries that separate *eve* from the *attP* site in the *hebe* gene, and come to a halt at the *homie* boundary associated with the *lacZ* reporter. This would generate a novel TAD, *eveMammoth* (*eveMa*), that extends from the *eve homie* all the way to the *homie* fragment at –142 kb, and encompasses both the reporter and the *eve* gene, including its enhancers (c.f., *Figure 3D*). Of course, the *eveMa* TAD could also be generated by a cohesin complex that initiated in, for example, TF. However, in this case, the runaway cohesin complex would have to break through the intervening boundary roadblocks in both directions. In both the boundary-pairing and loop-extrusion models, the configuration of the chromatin fiber would lead to the activation of *lacZ* expression by the *eve* enhancers, while the reporter would be protected from the *hebe* enhancers by the *homie* boundary.

To test these two models, we used a transgene that has two divergently transcribed reporters, *lacZ* and *gfp*, that are each under the control of an *eve* promoter (*Figure 3A*). Two different versions of the transgene were generated. In the first, the *eve-lacZ* reporter is located "upstream" of *homie* (with respect to the 5'→3'orientation of *homie* in the *eve* locus), while *eve-gfp* is located downstream. In the second, the orientation of *homie* within the transgene is reversed, so that *eve-gfp* is upstream of *homie*, while *eve-lacZ* is downstream. These two transgenes were then individually inserted into the *attP* site at –142 kb. For the pair shown in *Figure 3A*, *GeimohL* and *GhomieL*, the transgenes were inserted so that *eve-lacZ* is on the *eve* side of the *homie* boundary. The *eve-gfp* reporter is on the opposite side of the *homie* boundary and is farther away from the *eve* locus. This places the *eve-gfp* reporter next to a series of *hebe* enhancers located farther upstream (*Figure 2C*), in the intron of the *hebe* gene, and thus it should be subject to regulation by these enhancers. In the other pair, *LhomieG* and *LeimohG*, the entire transgene is inserted in the opposite orientation in the chromosome so that *eve-gfp* is on the *eve* side of *homie*, while *eve-lacZ* is next to the *hebe* enhancers (illustrated in Figure 6A). As a control, we also generated a transgene in which *lambda* DNA instead of *homie* was inserted between the *eve-gfp* and *eve-lacZ* reporters. This transgene was oriented so that *eve-lacZ* was on the *eve* side of the *lambda* DNA, and *eve-gfp* was close to the *hebe* enhancers.

### The *eve* enhancers drive *lacZ* reporter expression in the *GeimohL* insert

The results for transgenes *GeimohL* and the control *GlambdaL* are most straightforward and will be considered first. In both the loop-extrusion and boundary-pairing models, the *GlambdaL* control is not expected to display any regulatory interactions with the *eve* locus. *Figure 4C* shows that this is the case: the *eve-lacZ* and *eve-gfp* reporters in *GlambdaL* embryos are silent at the blastoderm stage, and transcripts are only very rarely detected. Later in development, both reporters are expressed in

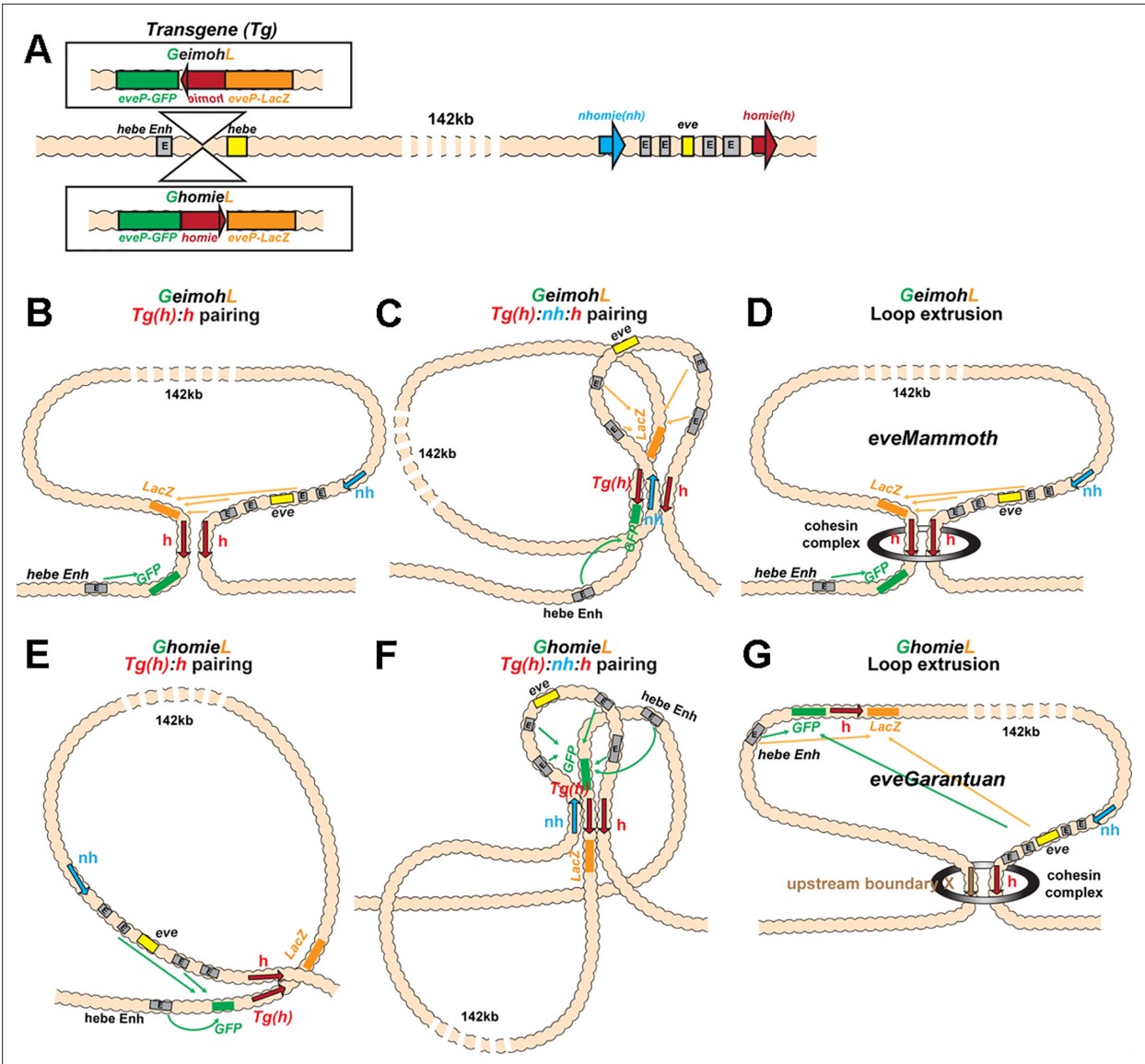

**Figure 3.** Schematics of boundary pairing and loop extrusion. (**A**) *GeimohL* and *GhomieL* transgenes and the *eve* locus on a linear map. The same color codes are used throughout. In the *eve* locus, *nhomie* (blue arrow) and *homie* (red arrow) are oriented, by convention, so that they are pointing toward the right. This convention is maintained in the two transgenes. In *GeimohL*, *homie* (top) is in the opposite orientation from *homie* in the *eve* TAD and so is pointing away from the *eve* TAD. In *GhomieL* (bottom), *homie* is in the same orientation as *homie* in the *eve* locus, and so is pointing toward the *eve* TAD. (**B**) Predicted boundary pairing interactions between *GeimohL* and the *eve* TAD. *homie* in the transgene pairs with *homie* in the *eve* TAD head-to-head. Since *homie* in the transgene is pointing in the opposite orientation from *homie* in the *eve* TAD, a stem-loop will be generated. (**C**) If *homie* in the transgene also pairs with *nhomie* in the *eve* TAD head-to-tail, a loop structure like that shown in C will be generated. In this topology, *eve-lacZ* is in close proximity to the *eve* enhancers, and *eve-gfp* is in contact with the *hebe* enhancer. (**D**) Loop-extrusion model for *GeimohL*. Transgene *homie* and endogenous *homie* determine the endpoints of the extruded *eveMammoth* (*eveMa*) loop. As in B, the topology of *eveMa* is a stem-loop. This topology brings *eve-lacZ* into close proximity to the *eve* enhancers, while *eve-gfp* is in another loop that contains the *hebe* enhancers. (**E**) Predicted boundary pairing between *GhomieL* and the *eve* locus. The *GhomieL* transgene is inserted in the same chromosomal orientation as *GeimohL*; however, the *homie* boundary is inverted so that it is in the same orientation as the *eve homie*, and so it is pointing toward the *eve* TAD. *homie* in the transgene will pair with *homie* in the *eve* locus head-to-head, and this generates a circle-loop. (**F**) If *homie* in the transgene also pairs with *nhomie* in the *eve* TAD, a loop structure like that shown in F will be generated. In this topology, *eve-gfp* will be activated by both the *eve* and *hebe* enhancers. (**G**) Loop-extrusion model for *GhomieL*. The cohesin complex bypasses transgenic *homie* and is stopped at an upstream boundary X, to generate a novel *eveGarantuan* (*eveGa*) loop. Both *eve-gfp* and *eve-lacZ* are located within the same *eveGa* TAD, and thus should interact with *eve*. Since the *hebe* enhancers are within this TAD as well, they would also be able to activate both reporters.

The online version of this article includes the following figure supplement(s) for figure 3:

**Figure supplement 1.** Loop extrusion: *eveElephant*.

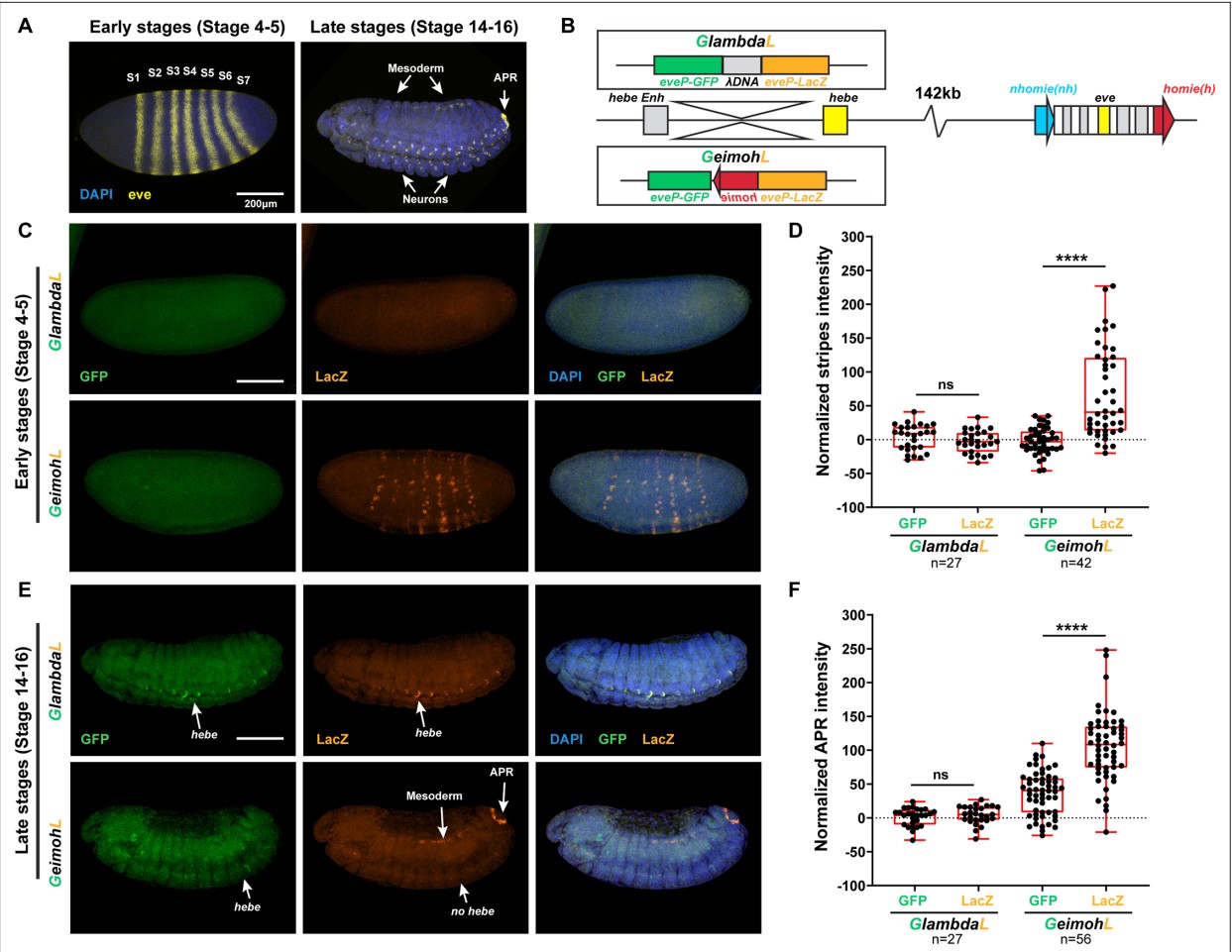

**Figure 4.** The *lacZ* reporter is activated by *eve* enhancers in the *GeimohL* insert. (**A**) Expression patterns of endogenous *eve* in early (stage 4–5) and late (stage 14–16) embryos by smFISH. *eve* (Atto 633) probes were used to hybridize with *eve* mRNA. *eve* is shown in yellow and DAPI in blue. N>3. (**B**) Schematics of transgenes. (**C**) Expression patterns of transgenic reporters in early-stage embryos. Top: *GlambdaL*; bottom: *GeimohL*. GFP is in green, *lacZ* is in orange, and DAPI is in blue, here and in E. (**D**) Quantification of normalized stripe signals for transgenes shown in C. The reporter smFISH intensity (signal to background), here and in F, was measured, normalized, and plotted as described in Methods. N>3. n=27 for *GlambdaL* and n=42 for *GeimohL*. (**E**) Expression patterns of transgenic reporters in late-stage embryos. Top: *GlambdaL*; bottom: *GeimohL*. (**F**) Quantification of normalized stripe signals for transgenes shown in E. N>3. n=27 for *GlambdaL*, n=56 for *GeimohL*. Scale bars = 200 µm. N = # of independent biological replicates. n = # of embryos. The paired two-tailed t-test was used for statistical analysis. ****p<0.0001, ns: not significant. Raw measurements are available in ***Supplementary file 3***.

a repeating pattern along the dorsal midline (***Figure 4E***). This expression is driven by the *hebe* gene enhancers located just beyond the *gfp*-reporter (***Fujioka et al., 2009***).

In the case of the *GeimohL* insert, regulatory interactions between the transgene and the *eve* locus will be observed if a stem-loop is established that links the transgene to *eve*. In the loop-extrusion model, the breakthrough cohesin complexes would generate a stem-loop, *eveMammoth* (*eveMa*), by coming to a halt at the *homie* boundary in the transgene and the *homie* boundary in *eve*, as illustrated in ***Figure 3D***. This will place the *lacZ* reporter in close proximity to the *eve* enhancers. It will also disrupt the *eve* TAD, as *nhomie* is no longer linked to the *homie* boundary by cohesin. In the boundary-pairing model, the *lacZ* reporter in the transgene is 'upstream' of *homie* just like the *eve* gene and its enhancers are 'upstream' of *homie* in the *eve* locus. Since *homie* pairs with itself head-to-head, pairing between *homie* in the transgene and *homie* in the *eve* locus will generate the same stem-loop as that generated by loop-extrusion (***Figure 3B***). However, because *homie* pairs with *nhomie* head-to-tail, a more complex multi-loop structure like that in ***Figure 3C*** would be expected if the pairing interactions with the transgene do not disrupt the *eve* TAD. Consistent with a tripartite structure of the sort shown here, previous studies have shown that three *Mcp* transgenes on three

different chromosomes can interact with each other genetically (**Muller et al., 1999**). Subsequent direct visualization of *Mcp*-mediated pairing interactions in imaginal discs by **Vazquez et al., 2006** showed that four *Mcp*-containing transgenes separated by Mbs (at Bridges cytogenetic intervals ~65 on the left arm of chromosome 3, and ~83 and~95 on the right arm of chromosome 3) and/or located on different homologs, clustered in the same nuclear foci in 94% of the nuclei.

As predicted from the formation of a stem-loop linking the transgene to the *eve* locus, *lacZ* is expressed in blastoderm-stage embryos in 7 stripes that coincide with the 7 stripes of the endogenous *eve* gene. However, unlike *eve*, not all nuclei in the seven stripes express *lacZ* (compare with the *eve* control in **Figure 4A**). Later, during mid-embryogenesis, *lacZ* is expressed in the mesoderm, the CNS, and the anal plate in a pattern that mimics the endogenous *eve* gene. At this stage, almost all *eve*-positive cells are also positive for *lacZ*. Unlike the *GlambdaL* transgene, the *hebe* enhancers do not drive *eve-lacZ* expression, as they are beyond the *homie* boundary (**Figure 4E**). The *eve-gfp* reporter in the *GeimohL* transgene responds differently. As was observed for the *GlambdaL* control, the *eve* enhancers drive little, if any, *gfp* expression, either at the blastoderm stage or later in development (**Figure 4C and E**). Instead, *gfp* is expressed in the midline during mid-embryogenesis under the control of the *hebe* enhancers (**Figure 4E**), which should be in the same TAD as *eve-gfp*. These results would be consistent with both the loop-extrusion (*eveMa*) and boundary-pairing models (**Figure 3B–D**).

## The *eve* enhancers drive *GFP* reporter expression in the *GhomieL* insert

The situation is more complicated for the *GhomieL* transgene. This transgene is inserted in the same chromosomal orientation as *GeimohL*: *eve-lacZ* is on the *eve* side of the *homie* boundary and *eve-gfp* is on the *hebe* enhancer side of the boundary. However, the orientation of the boundary within the transgene is switched so that *eve-gfp* rather than *eve-lacZ* is 'upstream' of *homie*. In the boundary-pairing model, flipping the orientation of *homie* in the transgene but keeping the same orientation of the transgene in the chromosome will have two consequences. First, as illustrated in **Figure 3E**, the topology of the chromatin loop connecting the *homie* boundary in the transgene and the *homie* boundary in the *eve* locus will change from a stem-loop to a circle-loop. The reason for the change in loop topology is that *homie* pairs with *homie* head-to-head (**Fujioka et al., 2016**). As before, the transgene *homie* will pair with *eve homie* head-to-head to generate a simple circle-loop (**Figure 3E**); however, if, as expected, it also pairs with *nhomie* so that the *eve* TAD remains intact, a more complicated loop structure would be generated (**Figure 3F**). Second, the reporter that is preferentially activated by the *eve* enhancers will switch from *eve-lacZ* to *eve-gfp*. In order to be activated by the *eve* enhancers, the reporter must be "upstream" of the transgene *homie*. This physical relationship means that the *eve-gfp* reporter will be activated by the *eve* enhancers independent of the orientation of the transgene in the chromosome. On the other hand, the *eve-lacZ* reporter will not be activated by the *eve* enhancers. In addition, it will still be insulated from the *hebe* enhancers by the *homie* boundary.

It is not possible to form a circle-loop in the loop-extrusion model. Instead, the breakthrough cohesin complex will come to a halt, as before, when it encounters the inverted *homie* boundary at −142 kb (**Figure 3D**). This means that the *eve-lacZ* reporter will be in the same TAD (*eveMa*) as the *eve* enhancers and will be activated by the *eve* enhancers. In contrast, the *eve-gfp* reporter will be in the neighboring TAD and will not be activated by the *eve* enhancers. It will, however, be regulated by the nearby *hebe* enhancers.

**Figure 5** shows that the four predictions of the boundary-pairing model are correct: (a) the *eve* enhancers drive *eve-gfp* expression, (b) the *hebe* enhancers drive *eve-gfp* expression, (c) *eve-lacZ* is not subject to regulation by the *eve* enhancers, and (d) *eve-lacZ* is insulated from the *hebe* enhancers. This would also mean that pairing of the transgene *homie* in the *GhomieL* insert with *homie* in the *eve* locus generates a circle-loop (or a more complicated multi-loop structure if it also pairs with *nhomie*), not a stem-loop, as was the case for *GeimohL*. In contrast, since loop-extrusion cannot generate circle-loops, the key predictions of this model, namely that *eve-gfp* will be silent while *eve-lacZ* will be regulated by the *eve* enhancers, are not satisfied. Instead, *eve-gfp* is activated by the *eve* enhancers, while *eve-lacZ* is not.

While the results described above for the *GhomieL* transgene are inconsistent with a 'simple' loop-extrusion model, there is one potential caveat: the 5'→3' orientation of the *homie* boundary is inverted so that it is 'pointing' towards the *eve* locus, rather than away from it like the endogenous

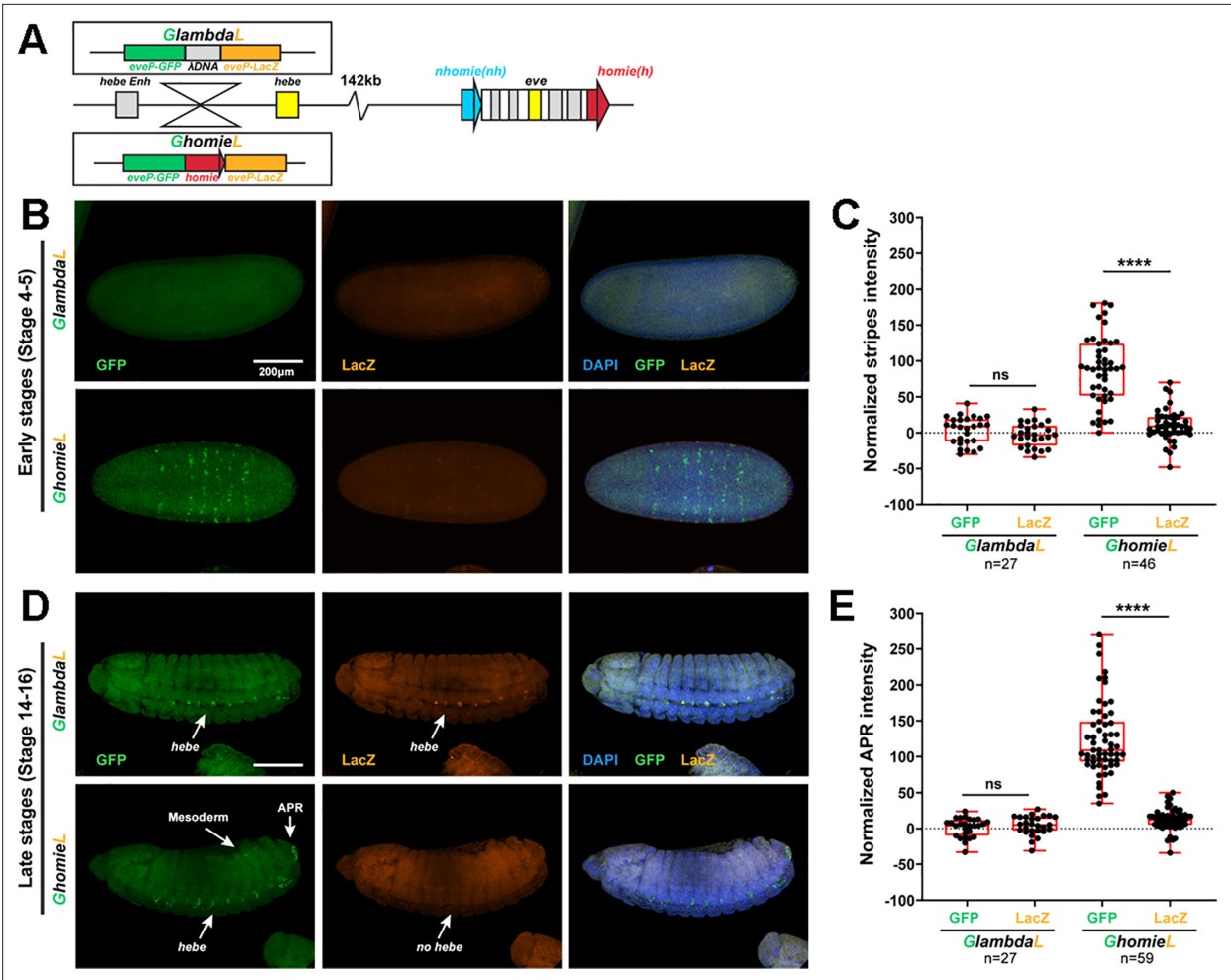

**Figure 5.** The *GFP* reporter is activated by both *eve* and *hebe* enhancers in the *GhomieL* insert. (**A**) Schematics of transgenes. (**B**) Expression patterns of transgenic reporters in early-stage embryos. Top: *GlambdaL*; bottom: *GhomieL*. *GFP* is in green, *lacZ* is in orange, and DAPI is in blue, here and in D. (**C**) Quantification of normalized stripe signals as represented in B. The reporter smFISH intensity (signal to background) was measured, normalized, and plotted as described in Methods, here and in E. N>3. n=27 for *GlambdaL*, n=46 for *GeimohL*. (**D**) Expression patterns of transgenic reporters in late-stage embryos. Top: *GlambdaL*; bottom: *GhomieL*. (**E**) Quantification of normalized stripe signals for transgenes shown in D. N>3. n=27 for *GlambdaL*, n=59 for *GeimohL*. Scale bars = 200 µm. N = # of independent biological replicates. n = # of embryos. The paired two-tailed t-test was used for statistical analysis. ****p<0.0001, ns: not significant. Raw measurements are available in **Supplementary file 3**.

*homie* boundary. In this orientation, it is possible that *homie* no longer functions as an effective roadblock, and instead the cohesin complex emanating from the *eve* locus (or from a loading site somewhere in between) slips past this *homie* fragment and continues until it is blocked by a properly oriented boundary further upstream. In this case, the newly formed stem-loop, *eveGargantuan* (*eveGa*; **Figure 3G**), would include the *eve-gfp* reporter, and it would then be activated by the *eve* enhancers. However, there are two problems with postulating this novel *eveGa* TAD. First, *eve-lacZ* would be in the same *eveGa* TAD as *eve-gfp*, and it should also be activated by the *eve* enhancers. Second, since the *homie* boundary was bypassed by the cohesin complex to form the larger *eveGa* TAD, it should not be able to block the *hebe* enhancers from activating *eve-lacZ*. Neither of these predictions is correct.

## *LeimohG* and *LhomieG*: Orientation of *homie* in the transgenes determines which reporter is activated

To confirm these conclusions, we examined the expression patterns of the reporters when the transgene is inserted in the opposite orientation in the –142 kb attP site. In this transgene orientation,

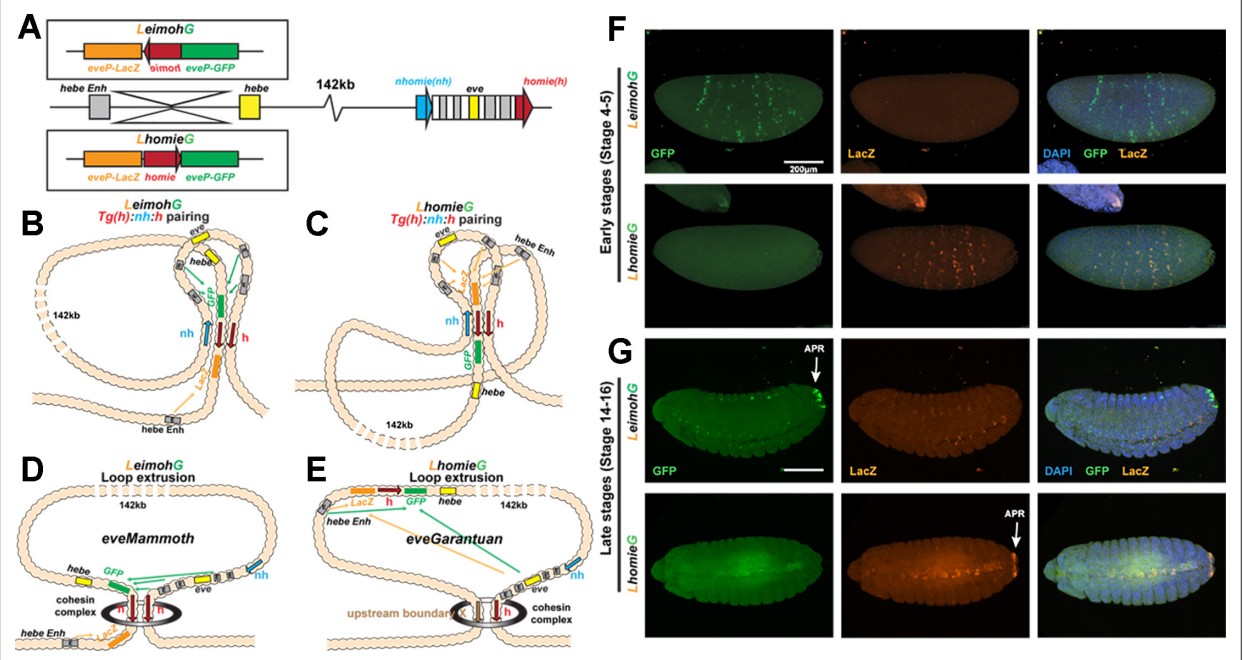

**Figure 6.** Expression of reporters in the *LeimohG* and *LhomieG* inserts. (**A**) Schematics of GE transgenes. (**B**) Boundary pairing for *LeimohG* transgene long-range interactions. In this topology, *GFP* is in close proximity to *eve* enhancers, and *lacZ* is in contact with the *hebe* enhancer region. (**C**) Boundary pairing for the *LhomieG* transgene. In this topology, *lacZ* is in close proximity to both the *eve* enhancers and the *hebe* enhancers. (**D**) Loop-extrusion model for *LeimohG*. Transgenic *homie* and endogenous *homie* flank the *eveMammoth* loop. In this topology, *GFP* is in close proximity to *eve* enhancers, and *lacZ* is close to the *hebe* enhancers. (**E**) Loop-extrusion model for the *LhomieG* transgene. The cohesin complex bypasses transgenic *homie* and is stopped at an upstream boundary X, to create the *eveGarantuan* loop. Both *GFP* and *lacZ* are at a similar physical distance from *eve* enhancers and *hebe* enhancers in this topology. (**F**) Expression patterns of transgenic reporters in early-stage embryos. Top: *LeimohG*; bottom: *LhomieG*. *GFP* is in green, *lacZ* is in orange, and DAPI is in blue. N=3, n=24 for both *LeimohG* and *LhomieG*. (**G**) Expression patterns of transgenic reporters in late-stage embryos. Top: *LeimohG*; bottom: *LhomieG*. N=3, n=24 for both *LeimohG* and *LhomieG*. Scale bars = 200 µm, N = # of independent biological replicates, n = # of embryos. Statistical analysis is in **Supplementary file 1**.

The online version of this article includes the following figure supplement(s) for figure 6:

**Figure supplement 1.** MicroC of *LeimohG*.

**Figure supplement 2.** MicroC of *LhomieG*.

---

*eve-gfp* is on the *eve* side of the *homie* boundary while *eve-lacZ* is close to the *hebe* enhancers. In the boundary-pairing model, the orientation of *homie* in the transgene determines which reporter will be preferentially activated, while the orientation of the transgene *homie* in the chromosome determines the topology of the resulting loop(s). The transgene *homie* will pair with the endogenous *homie* head-to-head. Since the transgene *homie* in *LeimohG* is in the opposite orientation from the endogenous *homie*, this pairing interaction will generate a stem-loop, if pairing interactions are exclusively pairwise. A more complicated multi-loop structure will be formed if the transgene *homie* interacts simultaneously with both endogenous boundaries (**Figure 6B**). In both cases, *eve-gfp* will be activated by the *eve* enhancers. On the other hand, since this orientation places *eve-lacZ* on the same side of the transgene *homie* as the *hebe* enhancers, they will activate it, rather than the *eve-gfp* reporter (**Figure 6B**). In *LhomieG*, the topology of the loop (or multi-loop) will switch, and *eve-lacZ* will be activated by both the *eve* and *hebe* enhancers (**Figure 6C**).

In the first version of the loop-extrusion model, the *eve-gfp* reporter in both *LeimohG* (**Figure 6D**) and *LhomieG* (not shown) should be activated by the *eve* enhancers, since it is included in the *eveMa* TAD when the transgenes are inserted so that *eve-gfp* is on the *eve* side of the *homie* boundary, and *eve-lacZ* will be regulated by the *hebe* enhancers (**Figure 6D**). In the second version of the loop-extrusion model, the functioning of the *homie* roadblock depends upon its orientation relative to endogenous *homie*. In this model, *LeimohG* is expected to form the *eveMa* TAD (**Figure 6D**), while *LhomieG* will form the larger *eveGa* TAD (**Figure 6E**).

As would be predicted by all three models, the *eve* enhancers activate the *eve-gfp* reporter in the *LeimohG* insert (**Figure 6F and G**). Likewise, *homie* blocks the *hebe* enhancers from activating the *eve-gfp* reporter, while they turn on *eve-lacZ* expression instead. As predicted by the boundary-pairing model, *eve*-dependent expression of the reporters switches from *gfp* to *lacZ* in the *LhomieG* insert (**Figure 6F and G**). This would not be expected in the first version of the loop-extrusion model, but it is predicted in the second version. However, as indicated in the diagram in **Figure 6E**, the two reporters, along with the *hebe* enhancers, are included in the larger *eveGa* TAD, and for this reason, both reporters should be activated equally by both the *eve* and *hebe* enhancers. This is not the case. The *hebe* enhancers are blocked by the intervening *homie* boundary, while there is little or no *eve*-dependent *gfp* expression.

## TAD formation by transgenes inserted at –142 kb

We used MicroC to examine the TAD organization of the genomic region extending from upstream of the attP site through the *eve* locus in 12–16 hr embryos carrying different transgene insertions. At this stage, *eve* is expressed in only a small number of cells in the CNS, mesoderm, and anal plate, and *hebe* is also expressed in only a small subset of cells. This means that most of the interactions detected by MicroC are in cells in which both the *eve* gene and the two reporters are inactive. In addition, while the transgene co-localizes with *eve* in less than about a fifth of blastoderm stage nuclei (based on the fraction of cells within *eve* stripes that activate a transgene reporter) the frequency of physical contact is expected to be much higher in 12–16 hr embryos (**Vazquez et al., 2006**). This is because at this stage of development, cell cycles are much longer, providing more time for transgene – endogenous *eve* pairing to occur. The key findings are summarized below.

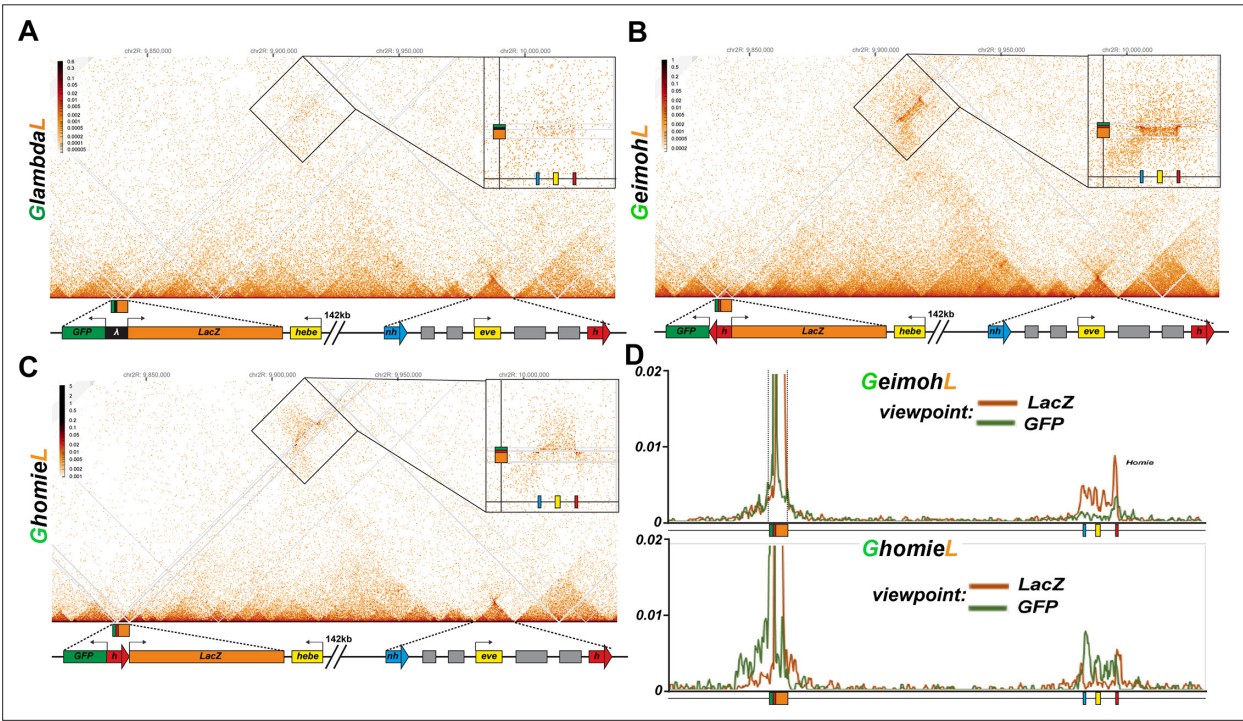

**Figure 7.** *homie* in transgenes *GeimohL* and *GhomieL* mediates long-range physical interactions with the *eve* TAD. Diagrams for (**A-C**) yellow box: 5' end of the *hebe* transcription unit. Transgene: green box: *eve-gfp* reporter, orange box: *eve-lacZ* reporter, black box: *lambda* DNA, red block arrow: *homie* DNA. *eve* TAD: blue block arrow: endogenous *nhomie*, red block arrow: endogenous *homie*; the directions of block arrows follow the established convention for *nhomie* and *homie*; gray boxes: *eve* enhancers, yellow box: *eve* transcription unit. (**A**) MicroC contact profile of the control *GlambdaL*. Inset shows a blow-up of the contacts between the *GlambdaL* transgene and the *eve* TAD. Note a slight increase in interaction frequency (compare to **Figure 1A**). (**B, C**) MicroC map of *GeimohL* and *GhomieL*, respectively. The key difference between the two inserts is the contacts between transgenes and sequences in the *eve* TAD. Note the changed pattern of interaction with the endogenous *eve* locus due to the orientation switch. (**D**) 'Virtual 4C' maps obtained from MicroC maps of 'B' (top panel) and 'C' (bottom panel). Viewpoints are shown in both panels from either the *lacZ* gene (orange) or the *GFP* gene (green).

### GlambdaL

While there is no evidence of contact between sequences in *eve* and sequences around the attP site at −142 kb in wild-type embryos, this not true for embryos carrying the dual reporter transgene. Perhaps the most unexpected finding is a weak, but clearly discernable, pattern of interaction linking the *GlambdaL* transgene to the *eve* locus (***Figure 7A***). However, since the *GlambdaL* transgene does not respond to the *eve* enhancers, this low level of physical interaction is not sufficient to drive detectable expression of the reporters. Although our experiments do not allow us to unequivocally identify which sequences in the transgene are responsible for this long-distance interaction, the most likely candidates are the two *eve* promoters in the *GlambdaL* transgene.

### GeimohL

Unlike *GlambdaL*, the *eve-lacZ* reporter in the *GeimohL* insert is activated by the *eve* enhancers, and this regulatory interaction is perfectly paralleled by the strong pattern of crosslinking between the transgene and the *eve* locus (***Figure 7B***). On the *eve* side, the interactions between the transgene and the *eve* locus generate a heavily populated band of crosslinked sequences that span the 16 kb *eve* TAD. On the transgene side, there is an unequal distribution of crosslinked sequences to either side of the transgene *homie* boundary. As shown in the inset, the heaviest density of crosslinked sequences is on the *eve-lacZ* side of the transgene, consistent with the activation of this reporter by *eve* enhancers. To confirm that the interaction bias mimics the difference in activity of the two reporters in cells in which the reporters are expressed, we analyzed the interaction profiles from viewpoints within either the *eve-lacZ* or *eve-gfp* genes. ***Figure 7D*** shows that *eve-lacZ* interacts with sequences extending across the *eve* TAD, while *eve-gfp* interactions are largely restricted to the *eve homie* element. While these results would seemingly be consistent with both the boundary-pairing and loop-extrusion models, there is no evidence of cohesin breaking through multiple intervening boundaries into order to establish the novel *eve(Mam)* TAD. Instead, the organization of the TADs and LDC domains in the interval in between the transgene and *eve* (***Figure 7B***) resemble that seen in wild-type NC14 embryos (see ***Figure 2***) (with the caveat that there are significantly fewer reads in the transgene experiment). Equally important, the same is true for the *eve* TAD. In order to generate the *eve(Mam)* TAD, the cohesin complex must break through the *nhomie* boundary and, in so doing, disrupt the *eve* TAD. However, there are no obvious alterations in either the *eve* volcano or plume with respect to their structure or relative contact density compared to the *GlambdaL* and wild type NC14 embryos. Likewise, there is no evidence of novel vertical and 45° crosslinking stripes that would reflect the boundary breakthrough events that are expected to accompany the formation of a physical connection between the transgene and the *eve* locus in the loop-extrusion model.

We next used the transgene *homie* as the viewpoint, to test the predictions of the boundary-pairing and loop-extrusion models. Since *homie* can pair with itself (head-to-head) and with *nhomie* (head-to-tail) in transvection assays (***Fujioka et al., 2016***), a prediction of the boundary-pairing model is that there will be physical interactions between the transgene *homie* and <u>*both*</u> *homie* and *nhomie* in the *eve* locus (***Vazquez et al., 2006***). Also based on what is known about how fly boundaries pair with each other (***Chetverina et al., 2014***; ***Chetverina et al., 2017***; ***Kyrchanova et al., 2008a***), self-interactions are expected to be stronger than heterologous interactions. In contrast, in the loop-extrusion model, the cohesin complex forms the *eveMa* TAD by linking *homie* in the transgene to *homie* in the *eve* locus, and this requires the cohesin complex to break through *nhomie* and disrupt the connections between *nhomie* and *homie* in the *eve* locus. In this case, transgene:*homie*←→*eve:homie* interactions should be observed, while transgene:*homie*←→*eve:nhomie* should not. ***Figure 8A*** shows that, as predicted by the boundary-pairing model, the transgene *homie* interacts with both *homie* and *nhomie* in the *eve* locus.

Since the *eve* TAD does not appear to be disturbed by the presence of the *GeimohL* transgene, these findings would suggest that the transgene *homie* likely pairs with both *nhomie* and *homie,* in a pattern like that illustrated in ***Figure 3C***. While the physical interactions between the transgene and *eve* boundaries are inconsistent with the loop-extrusion model, one could make the *ad hoc* assumption that runaway cohesin complexes form not one, but two different novel TADs. One would be *eveMa*. In the other, *eveElephant* (*eveEl*), a cohesin complex that initiates in, for example, the TE TAD would come to a halt on one side at the *homie* boundary in the transgene, and on the other side at the *nhomie* boundary in the *eve* locus (***Figure 3—figure supplement 1***). However, this configuration

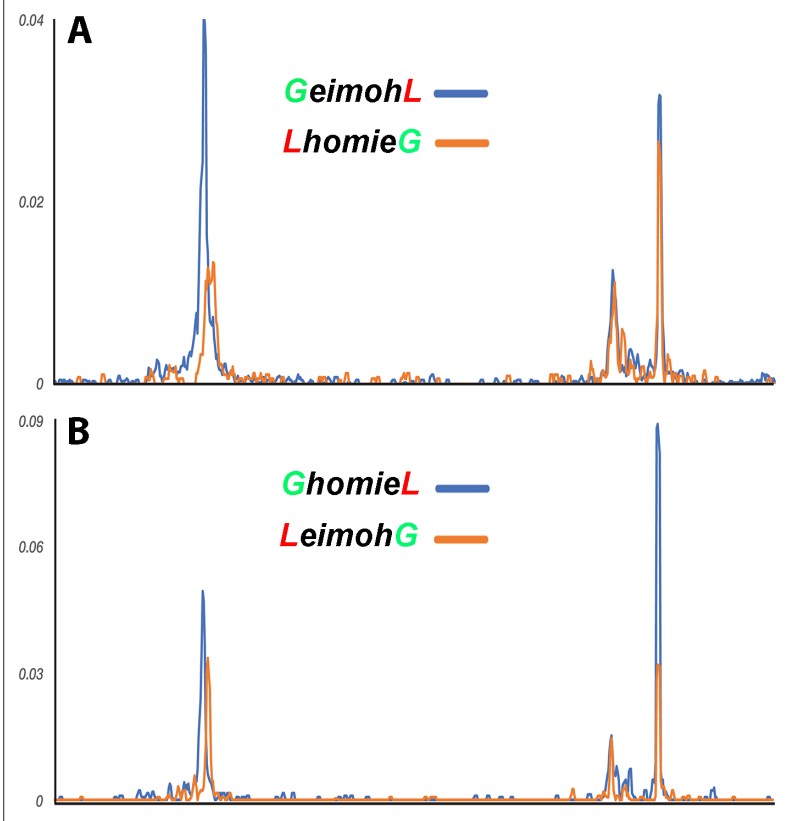

**Figure 8.** *homie* in the transgene interacts with *eve homie* and *nhomie*. 'Virtual 4C' maps of the *GeimohL* and *LhomieG* (**A**) or *GhomieL* and *LeimohG* (**B**) transgenes, oriented so that either *lacZ* (**A**) or *gfp* (**B**) is located upstream of the *homie* boundary in the transgene. (In the paired configuration, the reporter gene that is upstream of transgenic *homie* is the one that is brought closer to the endogenous *eve* enhancers, and so is the one that is expressed in an *eve* pattern.) Viewpoints are taken from the transgenic *homie* sequence. The major transgene peak (located toward the left end of each trace) is over the transgenic *homie* sequence in each case (it is in slightly different places in each pair of traces because the *lacZ* and *gfp* reporter genes are different sizes). Note that, in each case, transgenic *homie* interacts with endogenous *homie* (far right peak) more than it interacts with endogenous *nhomie*.

The online version of this article includes the following figure supplement(s) for figure 8:

**Figure supplement 1.** Blowup of interactions generated by the transgene with sequences neighboring −142 kb and the *eve* TAD.

would not explain how the transgene reporter is activated by *eve* enhancers. Since the *eve* enhancers in *eveEl* are located in a different TAD than the TADs containing *eve-lacZ* and *eve-gfp*, they should not be able to activate either reporter.

### GhomieL

When *homie* is reversed in the *GhomieL* transgene, the *eve-gfp* reporter, instead of *eve-lacZ*, responds to the *eve* enhancers. In the boundary-pairing model, *eve-gfp* is activated because self- and heterologous-pairing interactions between *homie* and *nhomie* are orientation-dependent. As a result, the topology of the loop formed between the transgene and *eve* is transformed in a way that brings the *eve-gfp* reporter, instead of *eve-lacZ*, into close proximity to the *eve* enhancers (see *Figure 3E and F*). Since only stem-loops can be generated in the loop-extrusion model, activation of the distal *eve-gfp* reporter would require the formation of a larger *eveGa* TAD in order to encompass both the transgene and the *eve* enhancers in the same looped domain (*Figure 3G*). However, as shown in *Figure 7C*, we do not observe the cross-linking pattern expected for a new *eveGa* TAD that includes the entire transgene at −142 kb and then extends to an even more distal boundary. Nor are there any novel vertical/45° crosslinking stripes or obvious perturbations

in the basic TAD organization in the region beyond –142 kb, or between –142 kb and the *eve* locus, as might be expected for cohesin complexes breaking through multiple intervening road-blocks. Instead, as was the case for *GeimohL*, the *GhomieL* transgene is physically linked to the *eve* locus, and there is a prominent band of crosslinked transgene sequences that spans the *eve* TAD (*Figure 7C*). Consistent with the expression patterns of the two reporters, the heaviest density of crosslinked sequences is on the *eve-gfp* side of the transgene (see inset). This is confirmed by the interaction profiles generated using either the *eve-gfp* or *eve-lacZ* gene body as the viewpoint: the most frequent contacts are between sequences within *eve* and the *eve-gfp* reporter (*Figure 7D*). In addition, as was the case for *GeimohL*, when the transgene *homie* is used as the viewpoint, inter-actions are observed between both *homie* and *nhomie* in the *eve* locus, with the most frequent corresponding to self-pairing interactions (*Figure 8B*). Other than postulating the *ad hoc eveEl* TAD (*Figure 3—figure supplement 1*), this would not be expected for either version of the loop-extrusion model.

### LhomieG and LeimohG

To confirm that the orientation of the *homie* boundary in the transgene is the key determinant of the physical interactions between the transgene reporters and the *eve* TAD, we used MicroC to analyze the TAD organization of *LhomieG* and *LeimohG* (*Figure 6—figure supplements 1 and 2*). Like their counterparts *GhomieL* and *GeimohL*, both show a strong band of interaction between the transgene and sequences spanning the *eve* locus. In the case of *LeimohG*, the pairing of the transgene *homie* with the endogenous *homie* is predicted to generate a stem-loop, while a more complicated multi-loop structure will be generated if it also pairs with *nhomie* (see *Figure 6B*). As shown in *Figure 6—figure supplement 1A*, these pairing interactions bring *eve-gfp* into contact with the *eve* enhancers and, as expected, the *eve-gfp* sequences interact more frequently with the *eve locus* than does *eve-lacZ* (inset *Figure 6—figure supplement 1A*). When *homie* is inserted in the transgene in opposite orientation, *homie* self-pairing is expected to generate a circle-loop, while the more complicated structure shown in *Figure 6C* will be formed if it also pairs with *nhomie*. While the sample preparation for MicroC was of poorer quality than the others, the inset in *Figure 6—figure supplement 2* shows that, as expected, *eve-lacZ* interacts more frequently with sequences in the *eve* locus than does *eve-gfp*. On the other hand, there is no evidence of a larger *eveGa* TAD, even though *eve-lacZ* is activated by the *eve* enhancers. These interac-tions are confirmed when either *eve-lacZ* or *eve-GFP* is used as viewpoint to plot the physical contacts between the *eve* TAD and the two reporters in *LhomieG* and *LeimohG* (*Figure 6—figure supplements 1B and 2B*). In addition, as was the case for the *GhomieL* and *GeimohL* transgenes, when the transgene *homie* is used as the viewpoint, interactions are observed between *homie* in the transgene and both *homie* and *nhomie* in the *eve* locus (*Figure 8A and B*). Again, the most frequent interactions are self-pairing interactions between *homie* in the transgene and the *eve homie*.

### Loop topology and interactions with neighbors

The results in the previous sections demonstrate that the orientation of *homie* in the transgene deter-mines which of the two reporters physically interacts with sequences in the *eve* TAD, and this is inde-pendent of the orientation of the transgene in the chromosome. On the other hand, the orientation of the transgene *homie* in the chromosome impacts how sequences flanking the transgene *homie* and the *eve* TAD interact with each other. With the important caveat that the depth of our MicroC sequencing is limited, the interaction profiles reflect the topology of the transgene-induced loop. For *homie* pointing away from *eve* (*GeimohL* and *LeimohG*), sequences to the *TER94* side of *eve* and to the *hebe* enhancer side of the transgene are more frequently crosslinked with each other (*Figure 8—figure supplement 1*). These interactions would be expected for either a stem-loop structure (as in *Figure 3B*) or a more complicated multi-loop structure like that shown in *Figure 3C*. For *homie* pointing towards *eve*, sequences on the *TER94* side of *eve* interact with sequences on the *eve* side of the transgene. This would be expected for either a circle-loop structure (as in *Figure 3E*) or the more complicated multi-loop structure shown in *Figure 3F* (neither of which can be generated by a simple loop-extrusion mechanism).

## Discussion

Two different mechanisms have been proposed to explain how TADs are formed and their endpoints determined. In the first, a cohesin complex first induces a small loop at a loading site within the TAD-to-be, and then progressively extrudes one or both strands until the complex encounters boundary roadblocks. If the loading site is located asymmetrically within the TAD-to-be, extrusion will continue on the unblocked strand until a roadblock is encountered. The endpoints of the loops were initially thought to be determined by the location of the first two roadblocks encountered by the cohesin complex (*Alipour and Marko, 2012*). In a refinement of this model, the orientation of a given road-block relative to the direction of movement of the cohesin complex determines whether it will come to a halt (*Davidson and Peters, 2021*; *Fudenberg et al., 2016*; *Guo et al., 2012*; *Guo et al., 2015*; *Sanborn et al., 2015*). It has also been suggested that cohesin complexes can break through a road-block and come to a halt at more distant roadblocks, so that in a population of cells analyzed by Hi-C or MicroC, the basic units of organization, the TADs, are overlaid by a hierarchy of LDC domains (*Hsieh et al., 2020*; *Krietenstein et al., 2020*). However, the only loop topologies that are possible in the loop-extrusion model are a stem-loop and an unanchored loop.

In the other mechanism, TAD formation is governed by boundary:boundary pairing interactions. In this case, the critical factors are the pairing and orientation preferences of potential partners, together with proximity. Since boundary pairing interactions often exhibit an orientation preference, the topology of the loop formed by paired boundaries can be either a stem-loop or a circle-loop, depending on their relative orientation in the chromosome. In this respect, it differs from the loop-extrusion model in that switching the orientation of the boundary does not preclude the formation of a TAD, whereas it can in some versions of the loop-extrusion model.

In the studies reported here, we have used MicroC to analyze the TAD organization of an ~150 kb chromosomal segment flanking the *eve* locus, and have experimentally tested the predictions of these two models for the mechanisms of TAD formation. Our studies do not support a loop-extrusion mechanism and are inconsistent with this model on multiple levels.

## Manipulating 'TAD' formation

In our experimental paradigm, a transgene containing two divergently transcribed reporters and the *eve* boundary *homie* is inserted 142 kb upstream of the *eve* TAD. Depending upon the orientation of the *homie* boundary within the transgene, either *eve-gfp* or *eve-lacZ* is activated by developmentally regulated enhancers in the *eve* TAD. Since there are a series of well-defined TADs and their associated boundary elements separating the transgene from *eve*, a version of the loop-extrusion model in which the cohesin complex invariably arrests upon encountering *nhomie* and *homie* at the ends of the *eve* TAD cannot account for this long-distance regulation. Instead, the extruding cohesin complex must be able to break through the boundaries located between *homie* in the transgene and *homie* in the *eve* TAD. This would generate a novel TAD, *eveMa*, with endpoints corresponding to the two *homie* boundaries (*Figure 3D*). In this case, the reporter on the *eve* side of the transgene *homie* would be in the same domain as the *eve* enhancers, and thus potentially subject to regulation. While *eveMa* could account for reporter expression in inserts in which the transgene *homie* is pointing away from *eve* (and also away from the reporter on the *eve* side of the transgene insert), it does not explain reporter expression when the transgene *homie* is pointing towards *eve* (and in this case towards the reporter on the *eve* side of the transgene insert). In the latter case, the reporter activated by the *eve* enhancers lies outside of the *eveMa* (cf., *Figure 3D*), and thus is not included in the same TAD as the *eve* enhancers. In order to account for activation of the distal reporter, one must imagine that the cohesin complex breaks through the *homie* boundary in the transgene and continues on until extrusion is halted by an even more distal boundary. In this revision of the loop-extrusion model, both reporters would be included in the same TAD as the *eve* enhancers (c.f., *Figure 3G*, *eveGa*), and both reporters should be activated. However, this is not observed.

## TAD organization: loop extrusion versus boundary pairing

Analysis of the TAD organization in the region spanning the transgene insertion site and *eve* also does not fit with the expectations of the loop-extrusion model. To account for reporter activation, the cohesin complex must be able to break through the *nhomie* boundary. This breakthrough could be engineered by a cohesin complex that initiated loop extrusion either from within the *eve* TAD or from

one of the TADs located between the *nhomie* boundary and the transgene at −142 kb. However, in the absence of the transgene, there is no indication that *nhomie* is particularly prone to break-through events. In fact, the *eve* TAD differs from most of the TADs in between *eve* and the transgene in that it is incorporated into larger LDC domains much less frequently. There is also no reason to imagine that the presence of a transgene carrying *homie* (or *nhomie*: see *Fujioka et al., 2016*) at −142 kb would somehow induce *nhomie* break-through events.

As expected from the regulatory interactions, *homie*-containing transgenes at −142 kb generate a strong band of crosslinking events that physically link the transgene to the *eve* locus. In the loop-extrusion model, this strong band of interaction should generate a novel interaction triangle (LDC or HDIC) with endpoints corresponding to the *homie* elements in the transgene and *eve*. Contrary to this expectation, a triangle of increased interaction frequencies spanning the entire region between −142 kb and *homie* is not observed for any of the transgene inserts. In viewpoints from the *homie* element in the *GeimohL* and *LeimohG* transgenes, there are two peaks in the *eve* TAD. The more prominent peak maps to *homie*, while the less prominent peak maps to *nhomie*. To account for these interaction peaks, the loop-extrusion model would have to assume that the presence of the *GeimohL* and *LeimohG* transgenes induces the formation of two novel TADs, the *eveMa* (*homie:homie*) TAD in *Figure 3D* and another TAD, *eveElephant*, that links the transgene *homie* to the *nhomie* element in the *eve* locus (*Figure 3—figure supplement 1B*). However, while the *eveMa* TAD could potentially explain the activation of reporters on the proximal side of the transgene *homie* by the *eve* enhancers, these TADs do not explain how the *eve* enhancers are able to activate reporters on the distal (*hebe* enhancer) side of the *GhomieL* and *LhomieG* transgene. This would require the formation of yet another TAD, *eveGa*, that encompasses *eve*, the transgene, and one or more TADs beyond the *hebe* gene (*Figure 3G*). However, there is no indication of physical interactions between the *homie* boundary in *eve* and a boundary located beyond the *hebe* gene. Moreover, like *eveMa*, *eveGa* would also have to be accompanied by the formation of *eveEl*, since the *homie* element in *GhomieL* and *LhomieG* interacts with both *nhomie* and *homie*.

While these results are inconsistent with the expectations of the loop-extrusion model, they dovetail nicely with many of the predictions of the boundary-pairing model. First, in all four transgenes, *homie* interacts with both *homie* and *nhomie* in the *eve* locus, but does not interact with the multiple boundary elements located in between. This is consistent with other studies which show that boundaries often have strong partner preferences (*Gohl et al., 2011*; *Kyrchanova et al., 2011*; *Kyrchanova et al., 2008b*). Second, interaction frequencies are greater for self-pairing than for heterologous pairing. This fits with what is known about how partner preferences are determined (*Blanton et al., 2003*; *Chetverina et al., 2017*; *Erokhin et al., 2021*; *Kyrchanova et al., 2008a*; *Zolotarev et al., 2016*). Third, the pattern of reporter activation in the four transgenes is precisely as expected from previous studies on the orientation dependence of *homie* pairing with itself (head-to-head) and with *nhomie* (head-to-tail) (*Chen et al., 2018*; *Fujioka et al., 2016*). This orientation dependence means that the reporter located 'upstream' of *homie* is the one that is activated, independent of the orientation of the transgene in the chromosome. Fourth, the pattern of physical interactions between the *eve* locus and each of the transgenes matches the pattern of reporter activation. That is, when *eve-lacZ* sequences are preferentially crosslinked to sequences in the *eve* locus, then the *eve-lacZ* reporter is activated by the *eve* enhancers. Likewise, *eve-gfp* is preferentially crosslinked to sequences in *eve* when it is activated by the *eve* enhancers. Again, this is dependent on the orientation of *homie* in the transgene, but not on the orientation of the transgene in the chromosome. Fifth, unlike the loop-extrusion model where only stem-loops or unanchored loops can be generated, stem-loops, circle-loops and even more complicated structures can be, and, as we have shown here, actually are generated by boundary:boundary pairing.

## Are the pairing interactions of *homie* and *nhomie* unusual?

There are two reasons to think that the properties of most fly boundaries are similar to those of *homie* and *nhomie*. First, all the fly boundaries that have been tested in assays that are expected to require pairing do have this activity (*Blanton et al., 2003*; *Cai and Shen, 2001*; *Erokhin et al., 2011*; *Gohl et al., 2011*; *Kyrchanova et al., 2008a*; *Kyrchanova et al., 2011*; *Kyrchanova et al., 2007*; *Kyrchanova et al., 2008b*; *Muravyova et al., 2001*).

Second, multimerized binding sites for generic polydactyl zinc finger proteins like Su(Hw), Pita, dCTCF, Zw5, and Zipic that are found associated with many fly boundaries are not only able to block enhancer/silencer action but can also mediate pairing interactions in insulator bypass and transvection assays (*Erokhin et al., 2021*; *Kyrchanova et al., 2008a*; *Zolotarev et al., 2016*). Moreover, as is the case for *homie* and *nhomie*, pairing interactions require the appropriate partners: multimerized sites for Zw5 do not pair with multimerized sites for dCTCF (*Erokhin et al., 2021*; *Kyrchanova et al., 2008a*; *Zolotarev et al., 2016*). This makes good sense, since these proteins form dimers/multimers, as do other fly boundary factors like BEAF, Mod(mdg4), the BEN-domain protein Insensitive, and the Elba complex (*Avva and Hart, 2016*; *Bonchuk et al., 2021*; *Bonchuk et al., 2011*; *Bonchuk et al., 2020*; *Fedotova et al., 2018*; *Fedotova et al., 2019*; *Fedotova et al., 2017*; *Hart et al., 1997*). The ability to multimerize means that boundaries that share binding sites for the same protein, that is, for Pita, Zipic, or CTCF, can be physically linked to each other by a single protein complex.

Third, *homie* and *nhomie* are not the only examples of fly boundary elements that engage in specific orientation-dependent physical pairing interactions. In addition to *gypsy*, *Mcp*, and *Fab-7*, *Mohana et al., 2023* identified nearly 60 meta-loops in which distant TADs are linked together by what appears in most instances to be orientation-dependent boundary pairing interactions. *Figure 9* shows the MicroC contact profiles generated for two such meta-loops in 12–16 hr embryos. As observed in our *homie* transgene experiments, the meta-loop in panel A is formed by specific and orientation-dependent boundary:boundary (blue:purple) pairing interactions that bring together distant TADs. The blue boundary splits the *Leukocyte-antigen-related like* (*Lar*) gene into a small upstream TAD that contains the most distal promoter (black arrowhead), and a larger downstream TAD whose endpoint is close to an internal *Lar* promoter (green arrowhead). The purple boundary is located ~600 kb away on the left arm of the 2nd chromosome, between two TADs, Pa and Pb, in a gene poor region. The pairing of the blue and purple boundaries generates two rectangles of enhanced contacts. The one on the upper left corresponds to interactions of the sequences in the small TAD just upstream of the blue boundary with sequences in the Pa TAD located upstream of the purple boundary. The other box on the lower right is generated by interactions between sequences in the TAD downstream of the blue boundary and sequences downstream of the purple boundary. Based on the locations of interacting TADs, the blue and purple boundaries pair with each other head-to-head, generating the circle-loop in the diagram.

The meta-loop in panel B links a boundary (blue arrow) to a boundary (purple arrow) ~5 Mb away on the right arm of the 3rd chromosome. The blue boundary is located between a small TAD that contains three genes encoding UDP-glycosyltransferases and a larger TAD that includes *CG10164* and the most distal *beat IV* (*beaten path IV*) promoter. The purple boundary separates a large TAD containing five genes (*Sid*, *CG33346*, *CG31050*, *CG14062,* and *CG9988*) from a small TAD that contains the most distal promoter of the *snu* (*snustorr*) ABC transporter. The blue and purple boundaries pair with each other head-to-tail, and this generates a stem-loop meta-loop (see diagram). In this pairing configuration, sequences upstream of the blue boundary come into contact with sequences downstream of the purple boundary. This generates a box of interactions between the small TAD that contains the UDP-glycosyltransferase gene cluster and the small TAD that contains the most distal *snu* promoter. (Another, weaker rectangular block of interactions links sequences in the TAD immediately upstream of the UDP-glycosyltransferase gene cluster, which contains *CG10175*, and the small TAD containing the distal *snu* promoter). A second, larger rectangular box is generated by contacts between sequences in the TADs downstream of the blue boundary and upstream of the purple boundary. Note that the positioning of the rectangular interaction boxes differs for circle-loops and stem-loops.

The pairing interactions in the *Lar* and *beat IV* meta-loops are much simpler than those seen for *homie* in the dual reporter. In both cases, there is only a single contact point (blue:purple boundaries). In contrast, as shown in *Figure 7* and in *Figure 6—figure supplements 1 and 2*, the interaction between the dual reporter *homie* and the *eve* TAD generates a 'stripe' that spans the *eve* TAD together with crosslinking events between TADs upstream and downstream of the dual reporters and TADs located to either side of the *eve* TAD. This more complicated interaction pattern is consistent with the viewpoint analysis (*Figure 8*), which shows that the transgene *homie* physically interacts with both *homie* and *nhomie* in the *eve* TAD.

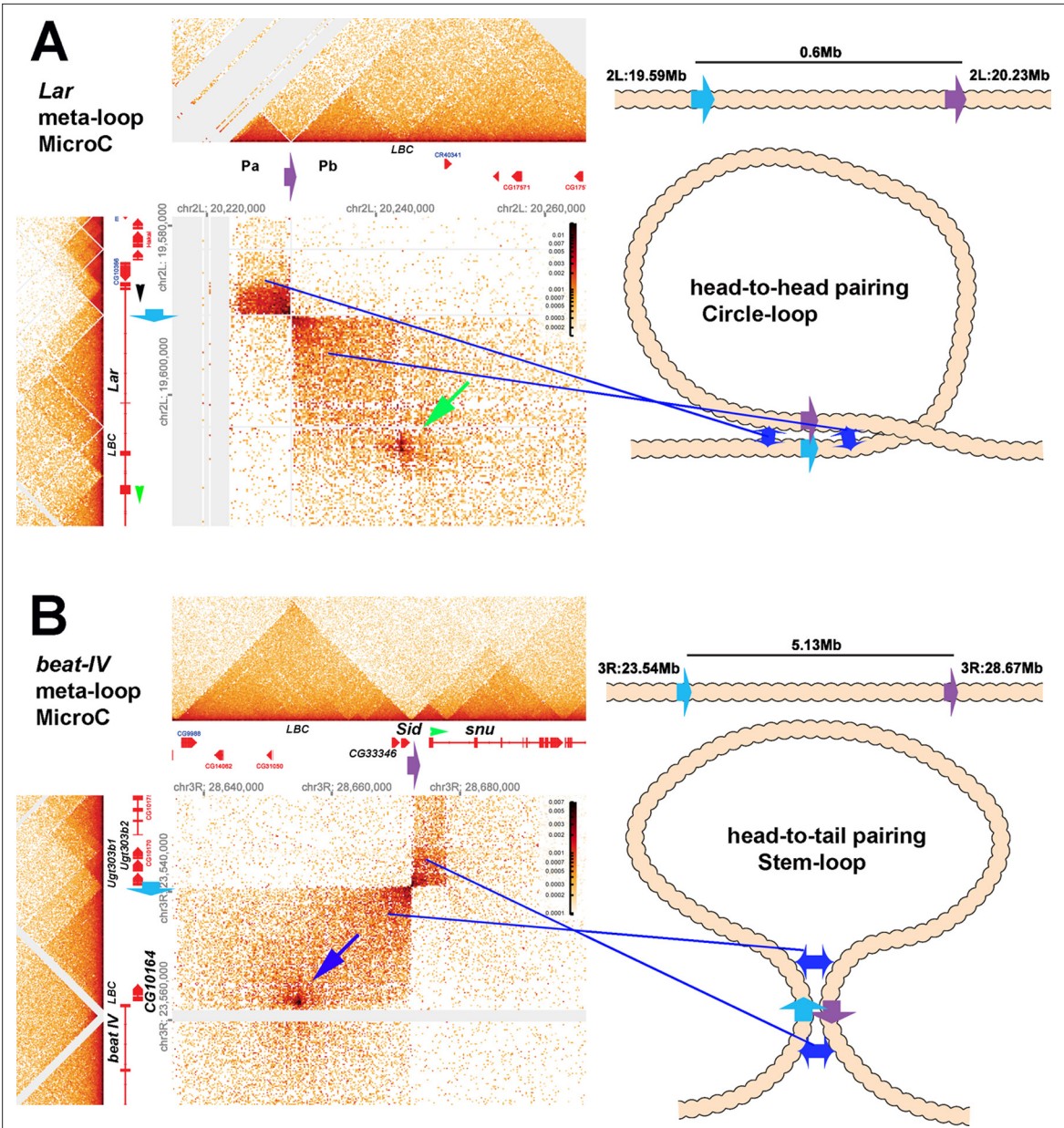

**Figure 9.** Circle-loop and stem-loop meta-loops. Distant boundary elements can find and pair with each other, forming large meta-loops. (**A**) Shows a circle-loop meta-loop formed by the head-to-head pairing of the blue (blue arrow) and purple (purple arrow) boundaries. As described in the text, the blue boundary separates a small TAD that contains the most distal *Lar* promoter (indicated by the black arrowhead) and a larger TAD that contains part of the *Lar* transcription unit (and is upstream of an internal *Lar* promoter: green arrowhead). The purple boundary is located between two TADS, Pa and Pb, in a gene-poor region of the 2nd chromosome approximately 600 kb away, downstream of the *Lar* gene. When the blue and purple boundaries pair, sequences in the small TAD upstream of the blue boundary come into contact with sequences in the Pa TAD upstream of the purple boundary. This interaction generates the dark rectangle on the upper left side of the interaction plot. Sequences downstream of the blue boundary in the TAD containing part of the *Lar* transcription unit are also brought into contact with sequences in the Pb TAD downstream of the purple boundary. This interaction generates the larger rectangular box on the bottom right side of the interaction plot. Shown in the diagram on the right is a schematic illustrating that the head-to-head pairing of the blue and purple boundaries generates a~600 kb circle-loop. In the circle-loop configuration, sequences in the TADs upstream of the blue and purple boundaries come into contact with each other, as do sequences downstream of the blue and purple boundaries (blue double arrows). (**B**) Shows a stem-loop meta-loop formed by the head-to-tail pairing of a boundary (blue arrow) located upstream of the *beat IV* gene with another boundary (purple arrow) located ~6 Mb away, between the *Sid* and *snu* genes. The blue boundary separates a small TAD containing three UDP-glycosyltransferase genes from a larger TAD containing the *CG10164* gene and the distal-most promoter of the *beat IV* gene. The purple boundary separates a ~20 kb TAD containing *Sid* and four other genes from a small TAD that contains the most distal *snu* promoter (green arrowhead). When the blue and purple boundaries pair with each other head-to-tail (see diagram on right), this interaction brings sequences in the

*Figure 9 continued on next page*

*Figure 9 continued*

large TAD downstream of the blue boundary into contact with sequences upstream of the purple boundary in the 20 kb *Sid* TAD, and this generates the large rectangular box to the lower left of interaction map. The stem-loop configuration also brings sequences upstream of blue boundary in the small TAD containing the three UDP-glycosyltransferase genes into contact with the TAD containing the distal-most *snu* promoter. This interaction gives a small rectangular box with a high contact density. Sequences in the TAD upstream of the UDP-glycosyltransferase TAD are also brought into contact with the small TAD containing the *snu* promoter. These two interacting 'zones' are indicated by blue double arrows in the diagram on right. In addition to boundary:boundary pairing, both meta-loops have a prominent dot in the larger rectangular boxes indicated by the green (panel A) and blue (panel B) arrows. Based on the ChIP signature of the sequences giving rise to these dots, they likely correspond to elements that are bound by the GAF-containing LBC complex, as indicated in the figure ('*LBC*' in both panels). LBC elements in other contexts function to bring enhancers into contact with promoters (*Batut et al., 2022*; *Kyrchanova et al., 2023*; *Kyrchanova et al., 2019a*; *Kyrchanova et al., 2019b*; *Levo et al., 2022*).

## Boundary pairing orientation and loop topology

Our analysis of the TAD organization in the *eve* neighborhood suggests that the predominant loop topology may be a circle-loop, not a stem-loop. Only the *eve* TAD (and perhaps the very small TAD TA) is clearly a stem-loop. It is topped by a distinctive volcano plume and is not overlaid by a hierarchy of LDC domains, indicating that it must be "insulated" from crosslinkable interactions with other, nearby TADs. By contrast, none of the TADs in the region spanning *eve* and the *hebe* gene have volcano plumes at their apex. Instead, all of these TADs are overlaid by a hierarchal organization of LDC domains (clouds) that diminish in crosslinking intensity as the number of in-between TADs increases. As discussed above, the contact pattern of the TADs to the left of *eve* are likely to arise from partner switching and/or head-to-head boundary pairing interactions that generate circle-loops. Pairing orientation versus loop topology is considered further in the accompanying paper (*Ke et al., 2024*).

## Orientation and long-distance regulatory interactions

Our experiments highlight an important constraint on long-distance regulatory interactions that are mediated not only by boundary elements like *homie*, *nhomie*, and the meta-loop boundaries in *Figure 9*, but also by the GAF-dependent (LBC) tethering and bypass elements (*Batut et al., 2022*; *Kyrchanova et al., 2023*; *Kyrchanova et al., 2018*; *Kyrchanova et al., 2019a*; *Kyrchanova et al., 2019b*; *Levo et al., 2022*; *Li et al., 2023*; *Mohana et al., 2023*), namely the orientation of the physical interactions that bring distant enhancers (or silencers) into close proximity with their gene targets. As we have shown here, the orientation of the physical interactions that bring distant sequences close together determines whether the enhancer will be able to activate its target gene. In principle, the effects of orientation could be mitigated by increasing (or decreasing) the distance between the boundary/tethering elements and either the enhancers (silencers) or the promoter. However, distance can be a critical, rate-limiting step in transcriptional activation. For example, live imaging experiments have shown that the frequency and amplitude of transcriptional bursts depend on the distance between the enhancer and target promoter. Nearly continuous bursting and high amplitude transcription were observed when the enhancer was located close the promoter (*Fukaya et al., 2016*). However, at distances as little as 6 kb, transcription became discontinuous with discrete bursts, while at 9 kb, the bursts were much less frequent, and the amplitude/output was substantially reduced (*Yokoshi et al., 2020*). If similar distance-dependent reductions in transcriptional efficiency are common, then it would not be possible to overcome the topological constraints on regulatory interactions by simply changing the distance between the boundary/tethering elements and the enhancers/promoters. While our studies do not directly address how chromosome architecture impacts regulatory interactions in vertebrates, it seems likely that long-distance regulatory interactions in vertebrates (*Tan et al., 2023*) will be subject to the same types of topological constraints as those that apply in flies.

## Materials and methods
### Plasmid construction and transgenic lines

The dual reporter constructs were described previously (*Fujioka et al., 2016*). In short, both reporters contain the *eve* basal promoter (–275 to +106 bp relative to *eve* start site), the *lacZ* (*eve-lacZ*) or *EGFP* (*eve-gfp*) coding region, and the *eve* 3' UTR (+1,300 to +1525 bp). The two reporters are divergently transcribed. The 367 bp *homie* fragment, or a 500 bp fragment from *lambda* phage DNA, was placed between the two reporters. For the *homie* fragment, it was inserted into the transgene in both

orientations, giving *eve-gfp:homie:eve-lacZ* (in *GhomieL* and *LeimohG*) and *eve-gfp:eimoh:eve-lacZ* (in *GeimohL* and *LhomieG*).

The –142 kb attP landing site was described previously (*Fujioka et al., 2009*). The –142 kb site contains two attP target sites for phiC31 recombinase-mediated cassette exchange (RMCE; *Bateman et al., 2006*) and *mini-white* as a marker. RMCE can result in the insertion of the transgene in either orientation, and all four possible insertions of the two transgenes were recovered. For two of these, *GeimohL* and *GhomieL*, the *eve-lacZ* reporter is on the *eve* side of the transgene *homie*, while *eve-gfp* is on the *hebe* enhancers side of the transgene *homie*. For the other two, *LeimohG* and *LhomieG*, the *eve-gfp* reporter is on the *eve* side of transgene *homie*, while *eve-lacZ* is on the *hebe* enhancers side of the transgene *homie*. RMCE events were identified by loss of *mini-white*, and the orientation of each insert was determined by PCR. All the new fly stocks described in this paper are available upon request.

## smFISH probe preparation

The sequences of target genes were obtained from FlyBase (flybase.org, *Gramates et al., 2022*). To design probes, the target gene sequences were submitted to the Biosearch Technologies Stellaris RNA FISH probe designer tool (biosearchtech.com), with parameters set to 20nt probes, 48 probes per gene, and mask level 5 for *Drosophila melanogaster*. The sequences of the probes used are listed in *Supplementary file 2*. The probes with 3' MdC(TEG-Amino) modifications were then ordered from Biosearch Technologies in 96-well plate format, cartridge purification and salt free conditions, and in 5 nmol scale. To prepare the fluorophore for probe labeling, the Sigma Atto NHS ester fluorophore (Atto 565 NHS or Atto 633 NHS) was dissolved in dimethylformamide at 5 mg/mL as the stock solution. For probe labeling, 1 nmol of each probe was combined and mixed with Atto NHS ester fluorophore at 1 mg/mL in 500 µL of 0.1 M sodium bicarbonate (pH 8.0–8.5) for 2 hr, and the probes were then precipitated with 0.5 M potassium acetate (pH 5.2) in 75% EtOH. Then, the probes were dried and resuspended in 50 µL of DEPC-treated $H_2O$. The probes were then injected into HPLC columns in which the percentage 0.1 M triethylammonium acetate varied from 7% to 30% with a flow rate of 1 mL/min. To monitor coupled probes during the HPLC purification, the detector was set to 260 nm for DNA and to the absorption wavelength of the coupled fluorophore. The coupled probes were collected when both 260 nm and the absorption wavelength channel were detected. The coupled probes were then dried and diluted to the appropriate concentration for smFISH experiments in DEPC-treated $H_2O$.

## smFISH

smFISH methods were adapted and modified from previous studies (*Little and Gregor, 2018*; *Trcek et al., 2017*). 100–200 flies were placed in a cage with an apple juice plate at the bottom of the cage. Early-stage embryos were collected for 7 hr, while for later-stage embryos, collection was overnight. Embryos from each plate were washed into collection mesh and dechorionated in bleach for 2 min, then fixed in 5 mL of 4% paraformaldehyde in 1 X PBS and 5 mL of heptane for 15 min with horizontal shaking. The paraformaldehyde was then removed and replaced with 5 mL methanol. The embryos were then devitellinized by vortexing for 30 s, and washed in 1 mL of methanol twice. Methanol was then removed and replaced by PTw (1 X PBS with 0.1% Tween-20) through serial dilution at 7:3, 1:1, and 3:7 methanol:PTw. The embryos were washed twice in 1 mL PTw and then twice in 1 mL smFISH wash buffer (4 X SSC, 35% formamide and 0.1% Tween-20). Embryos were incubated with ~5 nM coupled smFISH probes in hybridization buffer (0.1 g/mL dextran sulfate, 0.1 mg/mL salmon sperm ssDNA, 2 mM ribonucleoside vanadyl complex, 20 µg/mL RNAse-free BSA, 4 X SSC, 1% Tween-20 and 35% formamide) for 12–16 h. Embryos were then washed twice for 2 hr in 1 mL smFISH wash buffer, followed by 4 X washing in 1 mL PTw. For DAPI/Hoechst staining, the embryos were stained with 1 ug/mL DAPI or Hoechst in PTw, and washed 3 X with 1 mL PTw. Finally, the embryos were mounted for imaging on microscope slides with Aqua PolyMount and a #1.5 coverslip.

## Imaging, image analysis, and statistics

Embryos from smFISH were imaged using a Nikon A1 confocal microscope system with a Plan Apo 20 X/0.75 DIC objective. Z-stack images were taken at intervals of 2 µm, 4 X average, 1024 x 1024 resolution, and the appropriate laser power and gain were set for 405, 561, and 640 channels to

avoid overexposure. Images were processed using ImageJ, and the maximum projection was applied to each of the stack images. To measure the stripe intensity of early embryos, multi-channel images were first split into single channels and the stripe signal was highlighted and selected by the MaxEntropy thresholding method. For APR intensity measurements, the ROI tool was used to crop out the APR region of late-stage embryos. The cells with APR signal were also highlighted and selected by MaxEntropy thresholding. The particle measurement tool was used to measure the average intensity of all cells that had a signal. At the same time, the background signal (average intensity) was taken from cells without a signal in the same embryo. The relative intensity (signal to background) for each embryo was calculated using the stripe signal and background from the same embryo. To make comparisons between independent biological replicates, the average background signal of all embryos from each replicate was calculated. The relative intensity of each embryo from each replicate was normalized based on the average background signal of all embryos from same replicates. GraphPad Prism was used for data visualization and statistical analysis. To compare intensity from embryos in different groups, different signals from the same embryo (e.g. *lacZ* and *GFP*) were paired, and paired two-tailed t-tests were used to calculate p-values. All raw measurements and normalized data are included in *Supplementary file 3*.

## MicroC library construction

Embryos were collected on yeasted apple juice plates in population cages for 4 hr, incubated for 12 hr at 25°C, then subjected to fixation as follows. Embryos were dechorionated for 2 min in 3% sodium hypochlorite, rinsed with deionized water, and transferred to glass vials containing 5 mL PBST (0.1% Triton-X100 in PBS), 7.5 mL n-heptane, and 1.5 mL fresh 16% formaldehyde. Crosslinking was carried out at room temperature for exactly 15 min on an orbital shaker at 250 rpm, followed by addition of 3.7 mL 2 M Tris-HCl pH7.5 and shaking for 5 min to quench the reaction. Embryos were washed twice with 15 mL PBST and subjected to secondary crosslinking. Secondary crosslinking was done in 10 mL of freshly prepared 3 mM final DSG and ESG in PBST for 45 min at room temperature with passive mixing. The reaction was quenched by addition of 3.7 mL 2 M Tris-HCl pH7.5 for 5 min, washed twice with PBST, snap-frozen, and stored at –80°C until library construction.

Micro-C libraries were prepared as previously described (*Batut et al., 2022*) with the following modifications: 50 uL of 12–16 hr embryos were used for each biological replicate. Sixty U of MNase was used for each reaction to digest chromatin to a mononucleosome:dinucleosome ratio of 4. Libraries were barcoded, pooled, and subjected to paired-end sequencing on an Illumina Novaseq S1 100 nt Flowcell (read length 50 bases per mate, 6-base index read).

## MicroC data processing

MicroC data for *D. melanogaster* were aligned to custom genomes edited from the Berkeley *Drosophila* Genome Project (BDGP) Release 6 reference assembly (*dos Santos et al., 2015*) with BWA-MEM (*Li and Durbin, 2009*) using parameters **-S -P –5** M. Briefly, the custom genomes are simply insertions of the transgenic sequence into the –142 kb integration site, as predicted from perfect integration. These events were confirmed using PCR post-integration. The resultant BAM files were parsed, sorted, de-duplicated, filtered, and split with Pairtools (https://github.com/open2c/pairtools; *Goloborodko, 2024*) . We removed pairs where only half of the pair could be mapped, or where the MAPQ score was less than three. The resultant files were indexed with Pairix (https://github.com/4dn-dcic/pairix; *Lee, 2024*). The files from replicates were merged with Pairtools before generating 100 bp contact matrices using Cooler (*Abdennur and Mirny, 2020*). Finally, balancing and Mcool file generation was performed with Cooler's Zoomify tool.

Virtual 4C profiles were extracted from individual replicates using FAN-C (*Kruse et al., 2020*) at 400 bp resolution. The values were summed across replicates and smoothed across three bins (1.2 kb). Viewpoints were determined based on the most informative region for interpretation. Ultimately we decided to use 800 bp regions in the gene body of either *GFP* or *lacZ*, moving downstream from the *eve* promoter. However, similar results were obtained using viewpoints between transgenic *homie* and the promoters of either gene. For *homie* viewpoints, due to transgenic *homie* being masked, some loss of signal is expected at the viewpoint (*Figures 7D and 8*).

## Acknowledgements

We thank Gordon Grey for running the fly food facility at Princeton, members of the Lewis Sigler Genomics Core facility for their invaluable assistance with DNA sequencing, and Sergey Ryabichko for his help with smFISH probes. We would also like to thank members of MOL431 for creative input. Special thanks to Qing Liu for invaluable technical assistance, and to Olga Kyrchanova, Daria Chetverina, Maksim Erokhin, Pavel Georigev, Tsutomu Aoki, Girish Deshpande, Airat Ibragimov, Sergey Ryabichko, Yuri Pritykin and Alex Ostrin for stimulating discussions and sharing unpublished data.

## Additional information

### Competing interests

Xinyang Bing: currently employed by a biotech company; however, this author's experimental contributions to the paper were done prior to taking the biotech job. The other authors declare that no competing interests exist.

### Funding

| Funder | Grant reference number | Author |
|---|---|---|
| National Institute of General Medical Sciences | R35 GM118147 | Mike Levine |
| National Institute of Diabetes and Digestive and Kidney Diseases | U01 DK127429 | Mike Levine |
| National Institute of General Medical Sciences | R01 GM137062 | James B Jaynes |
| New Jersey Commission on Cancer Research | COCR23PDF011 | Wenfan Ke |
| Histochemical Society | Keystone Grant | Wenfan Ke |
| National Institute of General Medical Sciences | R35 GM126975 | Paul Schedl |

The funders had no role in study design, data collection and interpretation, or the decision to submit the work for publication.

### Author contributions

Xinyang Bing, Wenfan Ke, Miki Fujioka, Data curation, Formal analysis, Investigation, Methodology; Amina Kurbidaeva, Sarah Levitt, Investigation; Mike Levine, Supervision, Funding acquisition; Paul Schedl, Conceptualization, Supervision, Writing - original draft, Writing - review and editing; James B Jaynes, Conceptualization, Supervision

### Author ORCIDs

Xinyang Bing ⓘ https://orcid.org/0000-0001-6789-1918
Wenfan Ke ⓘ https://orcid.org/0000-0002-7047-5445
Paul Schedl ⓘ https://orcid.org/0000-0001-5704-2349
James B Jaynes ⓘ http://orcid.org/0000-0001-7943-794X

Reviewer #1 (Public Review): https://doi.org/10.7554/eLife.94070.3.sa1
Reviewer #2 (Public Review): https://doi.org/10.7554/eLife.94070.3.sa2
Reviewer #3 (Public Review): https://doi.org/10.7554/eLife.94070.3.sa3
Author response https://doi.org/10.7554/eLife.94070.3.sa4

## Additional files

### Supplementary files

• Supplementary file 1. Statistical analysis of reporter expression in the *LeimohG* and *LhomieG*

transgene inserts.
- Supplementary file 2. smFISH oligo probes.
- Supplementary file 3. Imaging data.
- MDAR checklist

### Data availability

Sequence data are available at GEO GSE263229. Confocal images are available on Open Science Framework at https://doi.org/10.17605/OSF.IO/VNGAK. Raw measurements are in *Supplementary file 3*.

The following datasets were generated:

| Author(s) | Year | Dataset title | Dataset URL | Database and Identifier |
|---|---|---|---|---|
| Bing X, Bing X, Ke W, Fujioka M, Kurbidaeva A, Levitt S, Levine M, Schedl P, Jaynes J | 2024 | Chromosome Structure I: Loop extrusion or boundary:boundary pairing? | https://www.ncbi.nlm.nih.gov/geo/query/acc.cgi?acc=GSE263229 | NCBI Gene Expression Omnibus, GSE263229 |
| Ke W | 2024 | Chromosome structure in Drosophila is determined by boundary pairing not loop extrusion | https://osf.io/vngak/ | Open Science Framework, vngak |

The following previously published datasets were used:

| Author(s) | Year | Dataset title | Dataset URL | Database and Identifier |
|---|---|---|---|---|
| Bing X, Bing X, Batut P, Levine M | 2022 | Genome organization controls transcriptional dynamics during development | https://www.ncbi.nlm.nih.gov/geo/query/acc.cgi?acc=GSE171396 | NCBI Gene Expression Omnibus, GSE171396 |
| Bing X, Bing X, Levo M, Raimundo J, Levine M | 2022 | Transcriptional Coupling of Distant Regulatory Genes in Living Embryos | https://www.ncbi.nlm.nih.gov/geo/query/acc.cgi?acc=GSE173518 | NCBI Gene Expression Omnibus, GSE173518 |

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

# Appendix 1

**Appendix 1—key resources table**

| Reagent type (species) or resource | Designation | Source or reference | Identifiers | Additional information |
|---|---|---|---|---|
| Gene (*Drosophila melanogaster*) | eve | Flybase | FBgn0000606 | |
| Gene (*D. melanogaster*) | hebe | Flybase | FBgn0033448 | |
| Genetic reagent (*D. melanogaster*) | eve-lacZ:homie:eve-gfp | *Fujioka et al., 2016* | | |
| Genetic reagent (*D. melanogaster*) | eve-lacZ:eimoh:eve-gfp | *Fujioka et al., 2016* | | |
| Genetic reagent (*D. melanogaster*) | eve-gfp:homie:eve-lacZ | *Fujioka et al., 2016* | | |
| Genetic reagent (*D. melanogaster*) | eve-gfp:eimoh:eve-lacZ | *Fujioka et al., 2016* | | |
| Chemical compound, drug, solution | n-Heptane | Fisher Chemical | O3008-4 | |
| Chemical compound, drug, solution | Paraformaldehyde 20% solution, EM Grade | Electron Microscopy Sciences | 15,713 S | |
| Chemical compound, drug, solution | Formaldehyde, 16%, methanol free, Ultra Pure | Polysciences Inc | 18814–10 | |
| Chemical compound, drug, solution | PBS - Phosphate-Buffered Saline (10 X) pH 7.4, RNase-free | Thermo Fisher | AM9624 | |
| Chemical compound, drug, solution | Tween 20 | Sigma | P1379 | |
| Chemical compound, drug, solution | Triton X-100 | Bio-Rad | 161–0407 | |
| Chemical compound, drug, solution | Tris base | Sigma | 11814273001 | |
| Chemical compound, drug, solution | Methanol | Fisher Chemical | 203403 | |
| Chemical compound, drug, solution | SSC, 20 X | Thermo Fisher | 15557044 | |
| Chemical compound, drug, solution | Formamide | Thermo Fisher | 17899 | |
| Chemical compound, drug, solution | Dextran sulfate | Sigma | D8906 | |
| Chemical compound, drug, solution | Salmon Sperm DNA | Thermo Fisher | AM9680 | |
| Chemical compound, drug, solution | Ribonucleoside Vanadyl Complex | NEB | S1402S | |
| Chemical compound, drug, solution | Nuclease-free BSA | Sigma | 126609 | |
| Chemical compound, drug, solution | Triethylammonium Acetate | Sigma | 625718 | |
| Chemical compound, drug, solution | dGTP (100 MM) | VWR | 76510–208 | |
| Chemical compound, drug, solution | dTTP (100 MM) | VWR | 76510–224 | |

*Appendix 1 Continued on next page*

*Appendix 1 Continued*

| Reagent type (species) or resource | Designation | Source or reference | Identifiers | Additional information |
|---|---|---|---|---|
| Chemical compound, drug, solution | Lonza NuSieve 3:1 Agarose | Thermo Fisher | BMA50090 | |
| Other | T4 DNA ligase | NEB | M0202L | enzyme |
| Chemical compound, drug, solution | Biotin-11-dCTP | Jena Bioscience | NU-809-BIOX | |
| Chemical compound, drug, solution | Biotin-14-dATP | Jena Bioscience | NU-835-BIO14 | |
| Commercial assay or kit | Qubit dsDNA HS Assay Kit | Life Technologies Corp. | Q32851 | |
| Chemical compound, drug, solution | Atto 633 NHS ester | Sigma | 01464 | |
| Chemical compound, drug, solution | Phase Lock Gel, QuantaBio - 2302830, Phase Lock Gel Heavy | VMR | 10847–802 | |
| Commercial assay or kit | NEBNext Ultra II DNA Library Prep Kit for Illumina | NEB | E7645S | |
| Commercial assay or kit | Ampure Xp 5 ml Kit | Thermo Fisher | NC9959336 | |
| Commercial assay or kit | Hifi Hotstart Ready Mix | Thermo Fisher | 501965217 | |
| Commercial assay or kit | Dynabeads MyOne Streptavidin C1 | Life Technologies Corp. | 65001 | |
| Chemical compound, drug, solution | cOmplete, EDTA-free Protease Inhibitor Cocktail | Sigma | 11873580001 | |
| Chemical compound, drug, solution | N,N-Dimethylformamide | Sigma | 227056 | |
| Chemical compound, drug, solution | Potassium acetate solution | Sigma | 95843 | |
| Chemical compound, drug, solution | DSG (disuccinimidyl glutarate) | Thermo Fisher | PI20593 | |
| Other | T4 Polynucleotide Kinase - 500 units | NEB | M0201S | enzyme |
| Other | DNA Polymerase I, Large (Klenow) Fragment - 1000 units | NEB | M0210L | enzyme |
| Commercial assay or kit | End-it DNA End Repair Kit | Thermo Fisher | NC0105678 | |
| Other | Proteinase K recomb. 100 mg | Sigma | 3115879001 | enzyme |
| Other | Nuclease Micrococcal (s7) | Thermo Fisher | NC9391488 | enzyme |
| Chemical compound, drug, solution | EGS (ethylene glycol bis(succinimidyl succinate)) | Thermo Fisher | PI21565 | |
| Commercial assay or kit | Atto 565 NHS ester | Sigma | 72464 | |
| Sequence-based reagent | WK_LacZ_probeset | Biosearch Technologies | Custom FISH probe set | see ***Supplementary file 2*** |

*Appendix 1 Continued*

| Reagent type (species) or resource | Designation | Source or reference | Identifiers | Additional information |
|---|---|---|---|---|
| Sequence-based reagent | WK_GFP_probeset | Biosearch Technologies | Custom FISH probe set | see *Supplementary file 2* |
| Sequence-based reagent | WK_eve_probeset | Biosearch Technologies | Custom FISH probe set | see *Supplementary file 2* |
| Commercial assay or kit | NEBNext Multiplex Oligos for Illumina | NEB | E7335S | |
| Commercial assay or kit | Dig RNA labeling mixture | Roche | 11277073910 | |
| Commercial assay or kit | T7 RNA polymerase | Roche | 10881767001 | |
| Commercial assay or kit | NEB IN solution | Roche | 11383213001 | |
| Commercial assay or kit | BCIP X-PHOS IN solution | Roche | 11383221001 | |
| Commercial assay or kit | ant-digoxigenin-ap | Roche | 11093274910 | |
| Software, algorithm | Fiji (ImageJ) | *Schindelin et al., 2012* | fiji.sc | |
| Software, algorithm | NIS element | Nikon | https://microscope.healthcare.nikon.com/products/software/nis-elements | |
| Software, algorithm | GraphPad Prism 8 | GraphPad Software | https://www.graphpad.com/ | |
| Software, algorithm | HiGlass | *Kerpedjiev et al., 2018* | https://higlass.io/app | |
| Software, algorithm | bwa | *Li and Durbin, 2009* | https://bio-bwa.sourceforge.net/ | |
| Software, algorithm | samtools | GitHub/open source | https://samtools.github.io/ | |
| Software, algorithm | pairsamtools | *Goloborodko et al., 2024* | https://github.com/mirnylab/pairsamtools | |
| Software, algorithm | pairix | *Lee, 2024* | https://github.com/4dn-dcic/pairix | |
| Software, algorithm | cooler | *Abdennur and Mirny, 2020*; *Abdennur, 2016* | https://github.com/open2c/cooler | |
| Software, algorithm | Miniconda | Anaconda | https://docs.anaconda.com/miniconda.html | |
| Software, algorithm | Snakemake | GitHub/open source | https://snakemake.github.io/ | |

