## [Editor Report · eLife assessment]

This **valuable** work presents elegant experimental data from the *Drosophila* embryo supporting the notion that interactions among specific loci, called boundary elements, contribute to topologically associated domain (TAD) formation and gene regulation. The evidence supporting boundary:boundary pairing as a determinant of 3D structures is **compelling**; however, an inability to deplete loop extruders formally leaves open a possible contribution of loop extrusion. This study will be of interest to the nuclear structure community, particularly those using *Drosophila* as a model.

---

## [Referee Report · Reviewer #1 (Public Review)]

Summary:

The authors addressed how long-range interactions between boundary elements are established and influence their function in enhancer specificity. Briefly, the authors placed two different reporters separated by a boundary element. They inserted this construct ectopically ~140 kb away from an endogenous locus that contains the same boundary element. The authors used expression patterns driven by nearby enhancers as an output to determine which enhancers the reporters interact with. They complemented this analysis with 3D DNA contact mapping. The authors found that the orientation of the boundary element determined which enhancers each reporter interacted with. They proposed that the 3D interaction topology, whether being circular or stem configuration, distinguished whether the interaction was cohesin mediated or through an independent mechanism termed pairing.

Strengths:

The transgene expression assays are built upon prior knowledge of the enhancer activities. The 3D DNA contacts confirm that transgene expression correlates with the contacts. Using 4 different orientations covers all combinations of the reporter genes and the boundary placement.

Weaknesses:

The interpretation of the data as a refusal of loop extrusion playing a role in TAD formation is not warranted, as the authors did not deplete the loop extruders to show that what they measure is independent. As the authors show, the single long DNA loop mediated by cohesin loop extrusion connecting the ectopic and endogenous boundary is clearly inconsistent with the results, therefore the main conclusion of the paper that the 3D topology of the boundary elements a consequence of pairing is strong. However, the loop extrusion and pairing are not mutually exclusive models for the formation of TADs. Loop-extruding cohesin complexes need not make a 140 kb loop, multiple smaller loops could bring together the two boundary elements, which are then held together by pairing proteins that can make circular topologies.

---

## [Referee Report · Reviewer #2 (Public Review)]

In Bing et al, the authors analyze micro-C data from NC14 fly embryos, focusing on the eve locus, to assess different models of chromatin looping. They conclude that fly TADs are less consistent with conventional cohesin-based loop extrusion models and instead rely more heavily on boundary-boundary pairings in an orientation-dependent manner.

Overall, I found the manuscript to be interesting and thought-provoking. However, this paper reads much more like a perspective than a research article. Considering the journal is aimed at the general audience, I strongly suggest the authors spend some time editing their introduction to the most salient points as well as organizing their results section in a more conventional way with conclusion-based titles. It was very difficult to follow the authors' logic throughout the manuscript as written. It was also not clear as written which experiments were performed as part of this study and which were reanalyzed but published elsewhere. This should be made clearer throughout.

It has been shown several times that *Drosophila* Hi-C maps do not contain all of the features (frequent corner peaks, stripes, etc.) observed when compared to mammalian cells. Considering these features are thought to be products of extrusion events, it is not an entirely new concept that *Drosophila* domains form via mechanisms other than extrusion. That being said, the authors' analyses do not distinguish between the formation and the maintenance of domains. It is not clear to this reviewer why a single mechanism should explain the formation of the complex structures observed in static Hi-C heatmaps from a population of cells at a single developmental time point. For example, how can the authors rule out that extrusion initially provides the necessary proximity and possibly the cis preference of contacts required for boundary-boundary pairing whereas the latter may more reflect the structures observed at maintenance? Future work aimed at analyzing micro-C data in cohesin-depleted cells might shed additional light on this.

Additional mechanisms at play include compartment-level interactions driven by chromatin states. Indeed, in mammalian cells, these interactions often manifest as a "plume" on Hi-C maps similar to what the authors attribute to boundary interactions in this manuscript. How do the chromatin states in the neighboring domains of the eve locus impact the model if at all?

How does intrachromosomal homolog pairing impact the models proposed in this manuscript (Abed et al. 2019; Erceg et al., 2019). Several papers recently have shown that somatic homolog pairing is not uniform and shows significant variation across the genome with evidence for both tight pairing regions and loose pairing regions. Might loose pairing interactions have the capacity to alter the cis configuration of the eve locus?

In summary, the transgenic experiments are extensive and elegant and fully support the authors' models. However, in my opinion, they do not completely rule out additional models at play, including extrusion-based mechanisms. Indeed, my major issue is the limited conceptual advance in this manuscript. The authors essentially repeat many of their previous work and analyses. The authors make no attempt to dissect the mechanism of this process by modifying extrusion components directly. Some discussion of Rollins et al., 1999 on the discovery of Nipped-B and its role in enhancer-promoter communication should also be made to reconcile their conclusions in the proposed absence of extrusion events.

---

## [Referee Report · Reviewer #3 (Public Review)]

Bing et al. attempt to address fundamental mechanisms of TAD formation in *Drosophila* by analyzing gene expression and 3D conformation within the vicinity of the eve TAD after insertion of a transgene harboring a Homie insulator sequence 142 kb away in different orientations. These transgenes along with spatial gene expression analysis were previously published in Fujioka et al. 2016, and the underlying interpretations regarding resulting DNA configuration in this genomic region were also previously published. This manuscript repeats the expression analysis using smFISH probes in order to achieve more quantitative analysis, but the main results are the same as previously published. The only new data are the Micro-C and an additional modeling/analysis of what they refer to as the 'Z3' orientation of the transgenes. The rest of the manuscript merely synthesizes further interpretation with the goal of addressing whether loop extrusion may be occurring or if boundary:boundary pairing without loop extrusion is responsible for TAD formation. The authors conclude that their results are more consistent with boundary:boundary pairing and not loop extrusion; however, most of this imaging data seems to support both loop extrusion and the boundary:boundary models. This manuscript lacks support, especially new data, for its conclusions. Furthermore, there are many parts of the manuscript that are difficult to follow. There are some minor errors in the labelling of the figures that if fixed would help elevate understanding. Lastly, there are several major points that if elaborated on, would potentially be helpful for the clarity of the manuscript.

Major Points:

(1) The authors suggest and attempt to visualize in the supplemental figures, that loop extrusion mechanisms would appear during crosslinking and show as vertical stripes in the micro-C data. In order to see stripes, a majority of the nuclei would need to undergo loop extrusion at the same rate, starting from exactly the same spots, and the loops would also have to be released and restarted at the same rate. If these patterns truly result from loop extrusion, the authors should provide experimental evidence from another organism undergoing loop extrusion.

(2) On lines 311-314, the authors discuss that stem-loops generated by cohesin extrusion would possibly be expected to have more next-next-door neighbor contacts than next-door neighbor contacts and site their models in Figure 1. Based on the boundary:boundary pairing models in the same figure would the stem-loops created by head-to-tail pairing also have the same phenotype? Making possible enrichment of next-next-door neighbor contacts possible in both situations? The concepts in the text are not clear, and the diagrams are not well-labeled relative to the two models.

(3) The authors appear to cite Chen et al., 2018 as a reference for the location of these transgenes being 700nM away in a majority of the nuclei. However, the exact transgenes in this manuscript do not appear to have been measured for distance. The authors could do this experiment and include expression measurements.

(4) The authors discuss the possible importance of CTCF orientation in forming the roadblock to cohesin extrusion and discuss that Homie orientation in the transgene may impact Homie function as an effective roadblock. However, the Homie region inserted in the transgene does not contain the CTCF motif. Can the authors elaborate on why they feel the orientation of Homie is important in its ability to function as a roadblock if the CTCF motif is not present? Trans-acting factors responsible for Homie function have not been identified and this point is not discussed in the manuscript.

(5) The imaging results seem to be consistent with both boundary:boundary interaction and loop extrusion stem looping.

(6) The authors suggest that the eveMa TAD could only be formed by extrusion after the breakthrough of Nhomie and several other roadblocks. Additionally, the overall long-range interactions with Nhomie appear to be less than the interactions with endogenous Homie (Figures 7, 8, and supplemental 5). Is it possible that in some cases boundary:boundary pairing is occurring between only the transgenic Homie and endogenous Homie and not including Nhomie?

(7) In Figure 4E, the GFP hebe expression shown in the LhomieG Z5 transgenic embryo does not appear in the same locations as the LlambdaG Z5 control. Is this actually hebe expression or just a background signal?

(8) Figure 6- The LhomieG Z3 late-stage embryo appears to be showing the ventral orientation of the embryo rather than the lateral side of the embryo as was shown in the previous figure. Is this for a reason? Additionally, there are no statistics shown for the Z3 transgenic images. Were these images analyzed in the same way as the Z5 line images?

(9) Do the Micro-C data align with the developmental time points used in the smFISH probe assays?

---

## [Author Response]

The following is the authors’ response to the original reviews.

**Reviewer #1 (Public Review)**:Summary:The authors addressed how long-range interactions between boundary elements are established and influence their function in enhancer specificity. Briefly, the authors placed two different reporters separated by a boundary element. They inserted this construct ectopically ~140 kb away from an endogenous locus that contains the same boundary element. The authors used expression patterns driven by nearby enhancers as an output to determine which enhancers the reporters interact with. They complemented this analysis with 3D DNA contact mapping. The authors found that the orientation of the boundary element determined which enhancers each reporter interacted with. They proposed that the 3D interaction topology, whether being circular or stem configuration, distinguished whether the interaction was cohesin mediated or through an independent mechanism termed pairing.Strengths:The transgene expression assays are built upon prior knowledge of the enhancer activities. The 3D DNA contacts confirm that transgene expression correlates with the contacts. Using 4 different orientations covers all combinations of the reporter genes and the boundary placement.Weaknesses:The interpretation of the data as a refusal of loop extrusion playing a role in TAD formation is not warranted, as the authors did not deplete the loop extruders to show that what they measure is independent.

**(1.1)** To begin with, our findings do not exclude the possibility that cohesin loop extrusion has some sort of role in the formation or maintenance of TADs in flies or other aspects of chromosome structure. On the other hand, it clearly is not determinative in defining the end-points of TADs or in generating the resulting topology (stem-loop or circle-loop). Our main point, which we feel we have established unequivocally, is that it can’t explain many essential features of TADs or chromosome loops (see below) in *Drosophila*. This reviewer agrees with this point in their next paragraph (below). We also think that the loop extrusion model’s general acceptance as **THE** driving force behind TAD formation in mammals is unwarranted and not fully consistent with the available data, as explained below.

As to the reviewer’s specific point regarding depletion of loop extruders, we first note that completely eliminating factors encoding cohesin subunits in fly embryos isn’t readily feasible. As cohesin is essential starting at the beginning of embryonic development, and is maternally deposited, knockdowns/depletions would likely be incomplete and there would always be some remaining activity. As long as there is some residual activity – and no disruption in TAD formation is observed – this experimental test would be a failure. In addition, any defects that are observed might arise not from a failure in TAD formation via loop extrusion but rather because the rapid mitotic cycles would be disrupted. A far better approach would be to deplete/knockdown cohesin subunits in tissue culture cells, as there is no requirement for the cells to undergo embryonic development. Moreover, since cell division is relatively slow, the depletion would likely eliminate much if not all of the activity before a checkpoint is reached.

While a drastic depletion of cohesin is not feasible in our model organism, we would draw the reviewer’s attention to an experiment of this type which has already been done in mammalian tissue culture cells by Goel et al. (2023). Unlike most Hi-C studies in mammals, the authors used region capture MicroC (RCMC). In contrast to published genome-wide mammalian MicroC experiments (c.f., Hsieh et al. 2020; Krietenstein et al. 2020) which require large bin sizes to visualize mammalian “TADs”, the resolution of the experiments in Goel et al. (2023) is similar to the resolution in our MicroC experiments (200-400 bp). Author response image 1 is a MicroC contact prepared from the data in Goel et al. (2023), showing the *Pdm1g* locus on chromosome 5 before and after RAD21 depletion. The contact map visualizes a ~150 kb DNA segment using a bin size of 200 bp.

In the Goel et al. (2023) experiment (see Author response image 1), there was a 97% reduction in the amount of RAD21. However, as can be seen by comparing the contact profiles above and below the diagonal, there is little or no difference in TAD organization (dark triangles on the diagonal) after cohesin depletion when individual TADs are visualized with a bin size of 200 bp. These results would indicate that mammalian TADs *do not* require cohesin. In addition, TAD:TAD interactions (e.g., purple asterisks) are the same with or without cohesin. Likewise, TAD interactions between next-next door neighbors (red asterisks), next-next-next door neighbors (light blue asterisks), etc., are not disrupted by loss of cohesin.

Note also that the weak 45^o^ stripes connecting different TADs (c.f., blue/green arrowheads) are still present after RAD21 depletion. In the most popular version of the loop-extrusion model, cohesin loads at a site(s) somewhere in the TAD-to-be, and then extrudes both strands until it bumps into CTCF roadblocks. As illustrated in Figure 2–figure supplement 2, this mechanism generates a vertical stripe originating at the cohesin loading site and extending until cohesin bumps into the left or right roadblock, at which point the stripe transitions into 45^o^ stripe that ends when cohesin bumps into the other roadblock. While 45^o^ stripes are visible, there is no hint of a vertical stripe. This suggests that the mechanism for generating stripes, if it is an active mechanism (rather than passive diffusion) may be quite different. The 45^o^ stripes must be generated by a factor(s) that is anchored to one of the TAD boundaries and pulls through a chromosomal segment that can extend through several TADs (e.g., the green and blue arrowheads link next door TADs). However, this factor, whatever it is, ***is not*** cohesin. The reason for this is that the 45^o^ stripes are present both before and after RAD21 depletion. Moreover, if one were to imagine that the stripes represent a process involved in TAD formation, this process does not require cohesin (see Author response image 1).

**Author response image 1. sa4fig1:** 

[Author response image 1 is generated from data available from Goel et al., 2023.]

There is another observation that is inconsistent with the cohesin loop extrusion/CTCF roadblock model for TAD formation/maintenance. CTCF is not found at all of the TAD boundaries in this mouse chromosomal DNA segment. This would suggest that there are other DNA binding proteins that have chromosomal architectural functions besides CTCF. In flies, many of the chromosomal architectural proteins are, like CTCF, polydactyl zinc finger (PZF) proteins (Bonchuk et al. 2021; Bonchuk et al. 2022; Fedotova et al. 2017). These include Su(Hw), CTCF, Pita, Zipic and CLAMP. The PZF family in flies is quite large. There are ~250 different PZF genes, and since only a handful of these have been characterized, it seems likely that additional members of this family will have architectural functions. Thus far, only one boundary protein, CTCF, has received attention in studies on mammalian chromosome architecture. As the mammalian genome is much larger and more complicated than the fly genome, it is difficult to believe that CTCF is the sole chromosomal architectural protein in mammals. In this respect, it is worth noting that there are ~800 members of the PZF family in mammalian genomes (Fedotova et al. 2017).

Goel et al. (2023) did observe alterations in the contact profiles after RAD21 depletion when they visualized the *Ppm1g* region at much lower resolution (bin sizes of 5 kb and 1 kb). This is shown in Author response image 2, for a 1 kb bin size that visualizes a region spanning ~600 kb. The individual TADs that are seen with bin sizes of 200 bp are no longer visible. Instead they merge into larger TAD neighborhoods (c.f., Author response image 9). At this level of resolution, depletion of RAD21 impacts the contact map. One change is a reduction in the frequency of crosslinking events between distant sequences within a TAD neighborhood. This can be seen by comparing the region within the two boxed areas (blue lines for WT and purple lines for the RAD21 knockdown). As indicated by the green double arrow, crosslinking events between sequences distant from the diagonal are less frequent when RAD21 is depleted. Another change is the yield of contact “dots” that are visualized with a bin size of 1 kb. The contact dots indicated by the green asterisks are barely visible after RAD21 depletion. while the effects on the contact dots indicated by the blue asterisks are not clear-cut.

It worth noting that, with the exception of the high resolution experiments in Goel et al., all of the other studies on cohesin (and CTCF) have examined the effects on contact maps within (and between) large neighborhoods (bin sizes >1 kb). In most cases, these large neighborhoods are likely to be composed of many individual TADs like those seen in Author response image 1 above and in Fig. 2 of our paper. We also observe larger neighborhoods in the fly genome, though they do not appear to be as large as those in mammals. Our experiments do not address what role cohesin might have in facilitating contacts between more distant TADs located within the same neighborhoods, or between TADs in different neighborhoods, or whether loop extrusion is involved.

**Author response image 2. sa4fig2:** 

[Author response image 2 is generated from data available from Goel et al., 2023.]

We would also note that the *Drosophila* DNA segment in Fig. 2C contains 35 different genes, while the mammalian DNA segment shown in Fig. 1 has only 9. Thus, in this part of the fly genome, Pol II genes are more densely packed than in the mammalian DNA segment. Much of the fly genome is also densely packed, and the size of individual TADs will be likely be smaller, on average, than in mammals. Nevertheless, the MicroC profiles are not all that different. As is also common in flies, each TAD in the *Ppm1g* region only encompasses one or two genes. Note also that there are no volcano triangles with plumes as would be predicted for TADs that have a stem-loop topology.

**Author response image 3. sa4fig3:** *Abd-B* regulatory domains.

In fact, as shown in Author response image 3, the high-resolution contact profile for the *Ppm1g* region shows a strong resemblance to that observed for the parasegment-specific regulatory domains in the *abd-A* and *Abd-B* region of the *Drosophila* bithorax complex (BX-C). Each of the regulatory domains (*iab-3*, *iab-4*, *iab-5*, *iab-6*, *iab-7*, and *iab-8*) are bracketed by boundary elements, and they assemble into a series of discrete TADs that are thought to have a circle-loop topology. There are no volcano triangles with plumes. Instead, each TAD physically interacts with its immediate neighbors, next-next door neighbors, and so on. For *iab-6*, the first LDC domains are 5-6 and 6-7 (see labels within Author response image 3), while the second LDC domains are 4-6 and 5-8. This is similar to the pattern of decreasing contacts between neighbors seen in the *Ppm1g* regions of mouse chromosome 5.

As the authors show, the single long DNA loop mediated by cohesin loop extrusion connecting the ectopic and endogenous boundary is clearly inconsistent with the results, therefore the main conclusion of the paper that the 3D topology of the boundary elements a consequence of pairing is strong. However, the loop extrusion and pairing are not mutually exclusive models for the formation of TADs. Loop-extruding cohesin complexes need not make a 140 kb loop, multiple smaller loops could bring together the two boundary elements, which are then held together by pairing proteins that can make circular topologies.

**(1.2)** In the pairing model, distant boundaries bump into each other (by random walks or partially constrained walks), and if they are “compatible” they pair with each other, typically in an orientation-dependent manner. As an alternative, the reviewer argues that cohesin need not make one large 140 kb loop. Instead it could generate a series of smaller loops (presumably corresponding to the intervening TADs). These smaller loops would bring *homie* in the transgene in close proximity to the *eve* locus so that it could interact with the endogenous *homie* and *nhomie* elements in the appropriate orientation, and in this way only one of the reporters would be ultimately activated.

There are two problems with the idea that cohesin-dependent loop extrusion brings transgene *homie* into contact with *homie/nhomie* in the *eve* locus by generating a series of small loops (TADs). The first is the very large distances over which specific boundary:boundary pairing interactions can occur. The second is that boundary:boundary pairing interactions can take place not only in *cis*, but also in *trans*.

We illustrate these points with several examples. Author response image 4 shows an experiment in which attP sites located ~2 Mb apart were used to insert two different transgenes, one containing a *lacZ* reporter and the other containing the *eve* anal plate enhancer (AP) (Fujioka et al. 2016). If the *lacZ* reporter and the AP transgenes also contain *homie*, the AP enhancer can activate *lacZ* expression (panel A, top left). On the other hand, if one of the transgenes has *lambda* DNA instead of *homie*, no regulatory interactions are observed. In addition, as is the case in our experiments using the -142 kb platform, orientation matters. In the combination on the top left, the *homie* boundary is pointing away from both the *lacZ* reporter and the AP enhancer. Since *homie* pairs with itself head-to-head, pairing brings the AP enhancer into contact with the *lacZ* reporter. A different result is obtained for the transgene pair in panel A on the top right. In this combination, *homie* is pointing away from the *lacZ* reporter, while it is pointing towards the AP enhancer. As a consequence, the reporter and enhancer are located on opposite sides of the paired *homie* boundaries, and in this configuration they are unable to interact with each other.

**Author response image 4. sa4fig4:** Long-distance (2 Mb) *cis* regulation mediated by *eve* insulators.

[Author response image 4 is reproduced from Figure 7 from Fujioka et al., 2016.]

On the top left of panel B, the *homie* element in the AP enhancer transgene was replaced by a *nhomie* boundary oriented so that it is pointing towards the enhancer. Pairing of *homie* and *nhomie* head-to-tail brings the AP enhancer in the *nhomie* transgene into contact with the *lacZ* reporter in the *homie* transgene, and it activates reporter expression. Finally, like *homie*, *nhomie* pairs with itself head-to-head, and when the *nhomie* boundaries are pointing towards both the AP reporter and the *lacZ* reporter, reporter expression is turned on (Author response image 4, panel B, top right).

Long-distance boundary-dependent pairing interactions by the bithorax complex *Mcp* boundary have also been reported in several papers. Author response image 5 is from Muller et al. (1999). It shows the pattern of regulatory interactions (in this case PRE-dependent “pairing-sensitive silencing”) between transgenes that have a *mini-white* reporter, the *Mcp* and *scs’* boundaries and a PRE that is located close to *Mcp*. In this experiment flies carrying transgenes inserted at the indicated sites on the left and right arms of the 3^rd^ chromosome were mated in pairwise combinations, and their *trans-*heterozygous progeny examined for pairing-sensitive silencing of the *mini-white* reporter.

**Author response image 5. sa4fig5:** 

[Author response image 5 is reproduced from Figure 6 from Muller et al., 1999, with permission from Genetics Society of America. It is not covered by the CC-BY 4.0 license, and further reproduction of this figure would need permission from the copyright holder.]

Two examples of long-distance pairing-sensitive silencing mediated by *Mcp/scs’* are shown in Author response image 6. The transgene inserts in panel A are *w#12.43* and *ff#10.5*. *w#12.43* is inserted close to the telomere of 3R at 99B. *ff10.5* is inserted closer to the middle of 3R at 91A. The estimate distance between them is 11.3 Mb. The transgene inserts in panel B are *ff#10.5* and *ff#11.102*. *ff#11.102* is inserted at 84D, and the distance between them is 11 Mb. Normally, the eye color phenotype of the *mini-white* reporter is additive: homozygyous inserts have twice as dark eye color as hemizygous inserts, while in *trans-*heterozygous flies the eye color would be the sum of the two different transgenes. However, when a PRE is present and the transgene can pair, silencing is observed. In panel A, the t*rans-*heterozygous combination has a lighter eye color than either of the parents. In panel B, the *trans-*heterozygous combination is darker than one of the parents (*ff#10.5*) but much lighter than the other (*ff#11.102*).

**Author response image 6. sa4fig6:** 

[Author response image 6 is reproduced from Figure 5 from Muller et al*.*, 1999, with permission from Genetics Society of America. It is not covered by the CC-BY 4.0 license, and further reproduction of this panel would need permission from the copyright holder.]

All ten of the transgenes tested in Author response image 5 were able to engage in long distance (>Mbs) *trans*-regulatory interactions; however, likely because of how the chromosome folds on the Mb scale (e.g., the location of meta-loops: see (2.1) and Author response image 10 below) not all of the possible pairwise silencing interactions are observed. The silencing interactions shown in Author response image 5 and Author response image 6 are between transgenes inserted on different homologs. *Mcp/scs'*-dependent silencing interactions can also occur in *cis*. Moreover, just like the *homie* and *nhomie* experiments described above, Muller et al. (Muller et al. 1999) found that *Mcp* could mediate long-distance activation of *mini-white* and *yellow* by their respective enhancers.

The pairing-sensitive activity of the PRE associated with the *Mcp* boundary is further enhanced when the *mini-white* transgene has the *scs* boundary in addition to *Mcp* and *scs’*. In the experiment shown in Author response image 7, the pairing-sensitive silencing interactions of the *Mcp/scs’/scs* transgene are between transgenes inserted on *different* chromosomes. Panel A shows pairing-sensitive silencing between *w#15.60*, which is on the X chromosome, and *w#15.102*, which is on the 2^nd^ chromosome. Panel B shows pairing-sensitive silencing between the 2^nd^ chromosome insert *w#15.60* and a transgene, *w#15.48*, which is inserted on the 3^rd^ chromosome.

**Author response image 7. sa4fig7:** 

[Author response image 7 is reproduced from Figure 8 from Muller et al., 1999, with permission from Genetics Society of America. It is not covered by the CC-BY 4.0 license, and further reproduction of this panel would need permission from the copyright holder.]

The long-distance *trans* and *cis* interactions described here are not unique to *Mcp/scs’/scs*. Precisely analogous results have been reported by Sigrist and Pirrotta (1997) for the *gypsy* boundary when the *bxd* PRE was included in the transgene. Also like the *Mcp*-containing transgenes in Muller et al. (1999), Sigrist and Pirrotta observed pairing-sensitive silencing between *gypsy bxd*PRE transgenes inserted on different chromosomes. Similar long-distance (Mb) interactions have been reported for *Fab-7* (Bantignies et al. 2003; Li et al. 2011). Also like the *homie* and *nhomie* experiments described above, Muller et al. (1999) found that *Mcp* could mediate long-distance activation of *mini-white* and *yellow* by their respective enhancers. In addition, there are examples of “naturally occurring” long-distance regulatory and/or physical interactions. One would be the regulatory/physical interactions between the p53 enhancer upstream of *reaper* and *Xrp1* which was described by Link et al. (2013). Another would be the nearly 60 meta-loops identified by Mohana et al. (2023).

Like *homie* at -142 kb, the regulatory interactions (pairing-sensitive silencing and enhancer activation of reporters) reported in Muller et al. (1999) involve direct physical interactions between the transgenes. Vazquez et al. (2006) used the *lacI*/*lacO* system to visualize contacts between distant *scs/Mcp/scs’*-containing transgenes in eye imaginal discs. As indicated in RTable 1 lines 4-7, when both transgenes have *Mcp* and were inserted on the same chromosome, they colocalized in *trans-*heterozygotes (single dot) in 94% to 97% of the disc nuclei in the four pairwise combinations they tested. When the transgenes both lacked *Mcp* (Author response image 8, line 1), co-localization was observed in 4% of the nuclei. When *scs/Mcp/scs’*-containing transgenes on the 2^nd^ and 3^rd^ chromosome were combined (Author response image 8, line 8), colocalization was observed in 96% of the nuclei. They also showed that four different *scs/Mcp/scs’* transgenes (two at the same insertion site but on different homologs, and two at different sites on different homologs) co-localized in 94% of the eye imaginal disc nuclei (Author response image 8, line 9). These pairing interactions were also found to be stable over several hours. Similar co-localization experiments together with 3C were reported by Li et al. (2011).

**Author response image 8. sa4fig8:** 

[Author response image 8 is reproduced from Table 3 from Vazquez et al., 2006.]

The *de novo* establishment of *trans* interactions between compatible boundary elements has been studied by Lim et al. (2018). These authors visualized transvection (enhancer activation of a MS2 loop reporter in *trans*) mediated by the *gypsy* insulator, *homie* and *Fab-8* in NC14 embryos. When both transgenes shared the same boundary element, transvection/physical pairing was observed in a small subset of embryos. The interactions took place after a delay, and increased in frequency as the embryo progressed into NC14. As expected from previous studies, transvection was specific: it was not observed when the transgenes had different boundaries. For *homie*, it was also orientation-dependent. It was observed when *homie* was orientated in the same direction in both transgenes, but not when *homie* was orientated in opposite directions in the two transgenes.

While one could imagine that loop extrusion-dependent compaction of the chromatin located between *eve* and the transgene at -142 kb into a series of small loops (the intervening TADs) might be able to bring *homie* in the transgene close to *homie/nhomie* in the *eve* locus, there is no plausible cohesin-based loop-extrusion scenario that would bring transgenes inserted at sites 6 Mb, 11 Mb, on different sides of the centromere, or at opposite ends of the 3^rd^ chromosome together so that the distant boundaries recognize their partners and physically pair with each other. Nor is there a plausible cohesin-based loop extrusion mechanism that could account for the fact that most of the documented long-distance interactions involve transgenes inserted on different homologs. This is not to mention the fact that long-distance interactions are also observed between boundary-containing transgenes inserted on different chromosomes.

In fact, given these results, one would logically come to precisely the opposite conclusion. If boundary elements inserted Mbs apart, on different homologs and on different chromosomes can find each other and physically pair, it would be reasonable to think that the same mechanism (likely random collisions) is entirely sufficient when they are only 142 kb apart.

Yet another reason to doubt the involvement or need for cohesin-dependent loop extrusion in bringing the transgene *homie* in contact with the *eve* locus comes from the studies of Goel et al. (2023). They show that cohesin has no role in the formation of TADs in mammalian tissue culture cells (Author response image 1 above). So if TADs in mammals aren’t dependent on cohesin, there would not be a good reason to think at this point that the loops (TADs) that are located between *eve* and the transgene are generated by, or even strongly dependent on, cohesin-dependent loop extrusion.

It is also important to note that even if loop-extrusion were to contribute to chromatin compaction in this context and make the looping interactions that lead to orientation-specific pairing more efficient, the role of loop extrusion in this model is not determinative of the outcome, it is merely a general compaction mechanism. This is a far cry from the popular concept of loop extrusion as being THE driving force determining chromosome topology at the TAD level.

**Reviewer #2 (Public Review):**
In Bing et al, the authors analyze micro-C data from NC14 fly embryos, focusing on the eve locus, to assess different models of chromatin looping. They conclude that fly TADs are less consistent with conventional cohesin-based loop extrusion models and instead rely more heavily on boundary-boundary pairings in an orientation-dependent manner.Overall, I found the manuscript to be interesting and thought-provoking. However, this paper reads much more like a perspective than a research article. Considering eLife is aimed at the general audience, I strongly suggest the authors spend some time editing their introduction to the most salient points as well as organizing their results section in a more conventional way with conclusion-based titles. It was very difficult to follow the authors' logic throughout the manuscript as written. It was also not clear as written which experiments were performed as part of this study and which were reanalyzed but published elsewhere. This should be made clearer throughout.It has been shown several times that *Drosophila* Hi-C maps do not contain all of the features (frequent corner peaks, stripes, etc.) observed when compared to mammalian cells. Considering these features are thought to be products of extrusion events, it is not an entirely new concept that *Drosophila* domains form via mechanisms other than extrusion.

**(2.1)** While there are differences between the Hi-C contact profiles in flies and mammals, these differences likely reflect in large part the bin sizes used to visualize contact profiles. With the exception of Goel et al. (2023), most of the mammalian Hi-C studies have been low resolution restriction enzyme-based experiments, and required bin sizes of >1 kb or greater to visualize what are labeled as “TADs.” In fact, as shown by experiments in Goel et al. (see Author response image 1 and Author response image 2 above), these are not actually TADs, but rather a conglomeration of multiple TADs into a series of TAD neighborhoods. The same is true for the MicroC experiments of Krietenstein et al. and Hsieh et al. on human and mouse tissue culture cells (Hsieh et al. 2020; Krietenstein et al. 2020). This is shown in Author response image 9. In this image, we have compared the MicroC profiles generated from human and mouse tissue culture cells with fly MicroC profiles at different levels of resolution.

For panels A-D, the genomic DNA segments shown are approximately 2.8 Mb, 760 kb, 340 kb, and 190 kb. For panels E-H, the genomic DNA segments shown are approximately 4.7 Mb, 870 kb, 340 kb and 225 kb. For panels I-L, the genomic DNA segments shown are approximately 3 Mb, 550 kb, 290 kb and 175 kb.

As reported for restriction enzyme-based Hi-C experiments, a series of stripes and dots are evident in mammalian MicroC profiles. In the data from Krietenstein et al., two large TAD “neighborhoods” are evident with a bin size of 5 kb, and these are bracketed by 45^o^ stripes (A: black arrows). At 1 kb (panel B), the 45^o^ stripe bordering the neighborhood on the left no longer defines the edge of the neighborhood (blue arrow: panel B), and both stripes become discontinuous (fuzzy dots). At 500 (panel C) and 200 bp (panel D) bin sizes, the stripes largely disappear (black arrows) even though they were the most prominent feature in the TAD landscape with large bin sizes. At 200 bp, the actual TADs (as opposed to the forest) are visible, but weakly populated. There are no stripes, and only one of the TADs has an obvious “dot” (green asterisk: panel C).

Large TAD neighborhoods bordered by stripes are also evident in the Hsieh et al. data set in Author response image 9 panels E and F (black arrows in E and F and green arrow in F). At 400 bp resolution (panel G), the narrow stripe in panel F (black arrows) becomes much broader, indicating that it is likely generated by interactions across one or two small TADs that can be discerned at 200 bp resolution. The same is true for the broad stripe indicated by the green arrows in panels F, G and H. This stripe arises from contacts between the TADs indicated by the red bar in panels G and H and the TADs to the other side of the volcano triangle with a plume (blue arrow in panel H). As in flies, we would expect that this volcano triangle topped by a plume corresponds to a stem-loop. However, the resolution is poor at 200 bp, and the profiles of the neighboring TADs are not very distinct.

**Author response image 9. sa4fig9:** MicroC profiles at different bin sizes.

[Author response image 9 is adapted from Krietenstein et al 2020; Hsieh et al. 2020.]

For the fly data set, stripes can be discerned when analyzed at 800 bp resolution (see arrows in Author response image 10); however, these stripes are flanked by regions of lower contact, and represent TAD-TAD interactions. At 400 bp, smaller neighborhoods can be discerned, and these neighborhoods exhibit a complex pattern of interaction with adjacent neighborhoods. With bin sizes of 200 bp, individual TADs are observed, as are TAD-TAD interactions like those seen near *eve*. Some of the TADs have dots at their apex, while others do not—much like what is seen in the mammalian MicroC studies.

*Stripes:* As illustrated in Author response image 9, panels A-D and E-H, the continuous stripes seen in low resolution mammalian studies (>1 kb bins) would appear to arise from binning artefacts. At high resolution where single TADs are visible, the stripes seem to be generated by TAD-TAD interactions, and not by some type of “extrusion” mechanism. This is most clearly seen for the volcano with plume TAD in Author response image 9 G and H. While stripes in Author response image 9 disappear at high resolution, this is not always true. There are stripes that appear to be “real” in Author response image 1 for the TADs in the *Ppm1g* region, and in Author response image 3 for the *Abd-B* regulatory domain TADs. Since the stripes in the *Ppm1g* region are unaffected by RAD21 depletion, some other mechanism must be involved (c.f. Shidlovskii et al. 2021).

**Author response image 10. sa4fig10:** MicroC profiles at different bin sizes.

*Dots:* The high resolution images in Author response image 9, panels D and H, show that, like *Drosophila* (Author response image 10, panel L), mammalian TADs don’t always have a “dot” at the apex of the triangle. This is not surprising. In the MicroC procedure, fixed chromatin is digested to mononucleosomes with MNase. Since most TAD boundaries in flies, and presumably also in mammals, are relatively large (150-400 bp) nuclease hypersensitive regions, extensive MNase digestion will typically reduce the boundary element sequences to oligonucleotides.

In flies, the only known sequences (at least to date) that end up giving dots (like those seen in Author response image 3) are bound by a large (>1,000 kd) GAF-containing multiprotein complex called LBC. In the *Abd-B* region of BX-C, the LBC binds to two ~180 bp sequences in *Fab-7* (dHS1 and HS3: (Kyrchanova et al. 2018; Wolle et al. 2015)), and to the centromere proximal (CP) side of *Fab-8*. The LBC elements in *Fab-7* (dHS1) and *Fab-8* (CP) have both blocking and boundary bypass activity (Kyrchanova et al. 2023; Kyrchanova et al. 2019a; Kyrchanova et al. 2019b; Postika et al. 2018). Elsewhere, LBC binds to the *bx* and *bxd* PREs in the *Ubx* regulatory domains, to two PREs upstream of *engrailed*, to the *hsp70* promoter, the *histone H3-H4* promoters, and the *eve* promoter (unpublished data). Based on ChIP signatures, it likely binds to most PREs/tethering elements in the fly genome (Batut et al. 2022; Li et al. 2023). Indirect end-labeling experiments (Galloni et al. 1993; Samal et al. 1981; Udvardy and Schedl 1984) indicate that the LBC protects an ~150-180 bp DNA segment from MNase digestion, which would explain why LBC-bound sequences are able to generate dots in MicroC experiments. Also unlike typical boundary elements, the pairing interactions of the LBC elements we’ve tested appear to be orientation-independent (unpublished data).

The difference in MNase sensitivity between typical TAD boundaries and LBC-bound elements is illustrated in the MicroC of the *Leukocyte-antigen-related-like* (*Lar*) meta-loop in Author response image 11, panels A and B. Direct physical pairing of two TAD boundaries (blue and purple) brings two TADs encompassing the 125 kb *lar* gene into contact with two TADs in a gene poor region 620 kb away. This interaction generates two regions of greatly enhanced contact: the two boxes on either side of the paired boundaries (panel A). Note that like transgene *homie* pairing with the *eve* boundaries, the boundary pairing interaction that forms the *lar* meta-loop is orientation-dependent. In this case the TAD boundary in the *Lar* locus pairs with the TAD boundary in the gene poor region head-to-head (arrow tip to arrow tip), generating a circle-loop. This circle-loop configuration brings the TAD upstream of the blue boundary into contact with the TAD upstream of the purple boundary. Likewise, the TAD downstream of the blue boundary is brought into contact with the TAD downstream of the purple boundary.

In the MicroC procedure, the sequences that correspond to the paired boundaries are not recovered (red arrow in Author response image 11, panel B). This is why there are vertical and horizontal blank stripes (red arrowheads) emanating from the missing point of contact. Using a different HiC procedure (dHS-C) that allows us to recover sequences from typical boundary elements (Author response image 11, panels C and D), there is a strong “dot” at the point of contact which corresponds to the pairing of the blue and purple boundaries.

**Author response image 11. sa4fig11:** *Lar* metaloop. (A, B) MicroC. (C, D) dHS-C.

There is a second dot (green arrow) within the box that represents physical contacts between sequences in the TADs downstream of the blue and purple boundaries. This dot is resistant to MNase digestion and is visible both in the MicroC and dHS-C profiles. Based on the ChIP signature of the corresponding elements in the two TADs downstream of the blue and purple boundaries, this dot represents paired LBC elements.

That being said, the authors' analyses do not distinguish between the formation and the maintenance of domains. It is not clear to this reviewer why a single mechanism should explain the formation of the complex structures observed in static Hi-C heatmaps from a population of cells at a single developmental time point. For example, how can the authors rule out that extrusion initially provides the necessary proximity and possibly the cis preference of contacts required for boundary-boundary pairing whereas the latter may more reflect the structures observed at maintenance?

**(2.2)** The MicroC profiles shown in Fig. 2 of our paper were generated from nuclear cycle (NC) 14 embryos. NC14 is the last nuclear cycle before cellularization (Foe 1989). After the nuclei exit mitosis, S-phase begins, and because satellite sequences are late replicating in this nuclear cycle, S phase lasts 50 min instead of only 4-6 min during earlier cycles (Shermoen et al. 2010). So unlike MicroC studies in mammals, our analysis of chromatin architecture in NC14 embryos likely offers the best opportunity to detect any intermediates that are generated during TAD formation. In particular, we should be able to observe evidence of cohesin linking the sequences from the two extruding strands together (the stripes) as it generates TADs *de novo*. However, there are no vertical stripes in the *eve* TAD as would be expected if cohesin entered at a few specific sites somewhere within the TAD and extruded loops in opposite directions synchronously, nor are their stripes at 45^o^ as would be expected if it started at *nhomie* or *homie* (see Figure 2–figure supplement 2.). We also do not detect cohesin-generated stripes in any of the TADs in between *eve* and the attP site at -142 kb. Note that in some models, cohesin is thought to be continuously extruding loops. After hitting the CTCF roadblocks, cohesin either falls off after a short period and starts again, or it breaks through one or more TAD boundaries, generating the LDC domains. In this dynamic model, stripes of crosslinked DNA generated by the passing cohesin complex should be observed throughout the cell cycle. They are not.

As for formation versus maintenance, and the possible involvement of cohesin loop extrusion in the former, but not the latter: This question was indirectly addressed in point (1.2) above. In this point we described multiple examples of specific boundary:boundary pairing interactions that take place over Mbs, in *cis* and in *trans* and even between different chromosomes. These long-distance interactions don’t preexist; instead they must be established *de novo*, and then maintained. This process was actually visualized in the studies of Lim et al. (Lim et al. 2018) on the establishment of *trans* boundary pairing interactions in NC14 embryos. There is no conceivable mechanism by which cohesin-based loop extrusion could *establish* the interactions in *trans* that have been documented in many studies on fly boundary elements. Also, as noted above, it seems unlikely that it is necessary for long-range interactions in *cis*. A more plausible scenario is that cohesin entrapment helps to stabilize these long-distance interactions after they are formed. If this were true, then one could argue that cohesin might also function to maintain TADs after boundaries have physically paired with their neighbors in *cis*. However, the RAD21 depletion experiments of Goel et al. (2023) (see Author response image 1 above) would rule out an essential role for cohesin in maintaining TADs after boundary:boundary pairing. In short, while we cannot formally rule out that loop extrusion might help bring sequences closer together to increase their chance of pairing, neither the specificity of that pairing, nor its orientation can be explained by loop extrusion. Furthermore, since pairing in *trans* cannot be facilitated by loop extrusion, invoking it as potentially important for boundary-boundary pairing in *cis* can only be described as a potential mechanism in search of a function, without clear evidence in its favor.

On the other hand, the apparent loss of contacts between TADs within large multi-TAD neighborhoods (see Author response image 2) would suggest that there is some sort of decompaction of neighborhoods after RAD21 depletion. It is possible that this might stress interactions that span multiple TADs as is the case for *homie* at -142, or for the other examples described in (1.2) above. This kind of involvement of cohesin might or might not be associated with a loop-extrusion mechanism.

Future work aimed at analyzing micro-C data in cohesin-depleted cells might shed additional light on this.

**(2.3)** This experiment has been done by Goel et al. (2023) in mammalian tissue culture cells. They found that TADs, as well as local TAD neighborhoods, are not disrupted/altered by RAD21 depletion (see Author response image 1 above and our response to point (1.1) of reviewer #1).

Additional mechanisms at play include compartment-level interactions driven by chromatin states. Indeed, in mammalian cells, these interactions often manifest as a "plume" on Hi-C maps similar to what the authors attribute to boundary interactions in this manuscript. How do the chromatin states in the neighboring domains of the eve locus impact the model if at all?

**(2.4)** Chromatin states have been implicated in driving compartment level interactions. Compartments, as initially described, were large, often Mb-sized chromosomal segments. These compartments share similar chromatin marks/states, and are thought to merge via co-polymer segregation. They were visualized using large multi-kb bin sizes. In the studies reported here, we use bin sizes of 200 bp to examine a DNA segment of less than 200 kb that is subdivided into a dozen or so small TADs. Several of the TADs contain more than one transcription unit, and they are expressed in quite different patterns. Thus, they might be expected to have different “chromatin states” at different points in development and in different cells in the organism. However, as can be seen by comparing the MicroC patterns in our paper that are shown in Fig. 2 with Fig. 7 and Figure 6–figure supplements 1 and 2, the TAD organization in NC14 and 12-16 hr embryos is for the most part quite similar. There is no indication that these small TADs are participating in liquid phase compartmentalization that depends upon shared chromatin/transcriptional states in NC14, and then again in 12-16 hr embryos.

In NC14 embryos, *eve* is expressed in 7 stripes, while it is potentially active throughout much of the embryo. In fact, the initial pattern in early cycles is quite broad and is then refined during NC14. In 12-16 hr embryos, the *eve* gene is silenced by the PcG system in all but a few cells in the embryo. However, here again the basic structure of the TAD, including the volcano plume, looks quite similar at these different developmental stages.

As for the suggestion that the plume topping the *eve* volcano triangle is generated because the TADs flanking the *eve* TAD share chromatin states and coalesce via some sort of phase separation: this model has been tested directly in Ke et al. (Ke et al. 2024). In Ke et al., we deleted the *nhomie* boundary and replaced it with either *nhomie* in the reverse orientation or *homie* in the forward orientation. According to the compartment model, changing the orientation of the boundaries so that the topology of the *eve* TAD changes from a stem-loop to a circle-loop should have absolutely no effect on the plume topping the *eve* volcano triangle. The TADs flanking the *eve* TAD would still be expected to share the same chromatin states, and so would still coalesce via phase transition. However, this is not what is observed. The plume disappears and is replaced by “clouds” on both sides of the *eve* TAD. The clouds arise because the *eve* TAD bumps into the neighboring TADs when the topology is a circle-loop.

We would also note that “compartment-level” interactions would not explain the findings presented in Author response image 4 – Author response image 7, in Table 1, or in Author response image 10. It is clear that the long-distance (Mb) interactions observed for *Mcp*, *gypsy*, *Fab-7*, *homie*, *nhomie* and the blue and purple boundaries in Author response image 10 arise by the physical pairing of TAD boundary elements. This fact is demonstrated directly by the MicroC experiments in Fig. 7 and Figure 6–figure supplements 1 and 2, and by the MicroC and dHS-C experiments in Author response image 10. There is no evidence for any type of “compartment/phase separation” driving these specific boundary-pairing interactions.

In fact, given the involvement of TAD boundaries in meta-loop formation, one might begin to wonder whether some of the “compartment level interactions” are generated by the specific pairing of TAD boundary elements, rather than by “shared chromatin states”. For example, the head-to-head pairing of the blue and purple boundaries generates a *Lar* meta-loop that has a circle-loop topology. As a consequence, sequences upstream of the blue and purple boundary come into contact, generating the small dark rectangular box on the upper left side of the contact map. Sequences downstream of the blue and purple boundary also come into contact, and this generates the larger rectangular box in the lower right side of the contact map. A new figure, Fig. 9, shows that the interaction pattern flips (lower left and top right) when the meta-loop has a stem-loop topology. If these meta-loops are visualized using larger bin sizes, the classic “compartment” patchwork pattern of interactions emerges. Would the precise patchwork pattern of “compartmental” interactions involving the four distant TADs that are linked in the two meta-loops shown in Fig. 9 persist as is if we deleted one of the TAD boundaries that forms the meta-loop? Would the precise patchwork pattern persist if we inverted one of the meta-loop boundaries so that we converted the topology of the loop from a circle-loop to a stem-loop or vice versa? We haven’t used MicroC to compare the compartment organization after deleting or inverting a meta-loop TAD boundary; however, a comparison of the MicroC pattern in WT in Fig. 1C with that for the *homie* transgenes in Fig. 7, Figure 6–figure supplements 1 and 2, and Figure 8–figure supplement 1 indicates (a) that novel patterns of TAD:TAD interactions are generated by this mini-meta-loop, and (b) that the patterns of TAD:TAD interaction depend upon loop topology. Were these novel TAD:TAD interactions generated instead by compartment-level interactions/shared chromatin states, they should be evident in WT as well (Fig. 1). They are not.

How does intrachromosomal homolog pairing impact the models proposed in this manuscript (Abed et al. 2019; Erceg et al., 2019). Several papers recently have shown that somatic homolog pairing is not uniform and shows significant variation across the genome with evidence for both tight pairing regions and loose pairing regions. Might loose pairing interactions have the capacity to alter the cis configuration of the eve locus?

**(2.5)** At this point, it is not entirely clear how homolog pairing impacts the *cis* configuration/MicroC contact maps. We expect that homolog pairing is incomplete in the NC14 embryos we analyzed; however, since replication of *eve* and the local neighborhood is likely complete, sister chromosomes should be paired. So we are likely visualizing the 3D organization of paired TADs.

In summary, the transgenic experiments are extensive and elegant and fully support the authors' models. However, in my opinion, they do not completely rule out additional models at play, including extrusion-based mechanisms. Indeed, my major issue is the limited conceptual advance in this manuscript. The authors essentially repeat many of their previous work and analyses.

**(2.6)** In our view, the current paper makes a number of significant contributions that go well beyond those described in our 2016 publication. These are summarized below.

A) While our 2016 paper used transgenes inserted in the -142 kb attP site to study pairing interactions of *homie* and *nhomie*, we didn’t either consider or discuss how our findings might bear on the loop extrusion model. However, since the loop extrusion model is currently accepted as established fact by many labs working on chromosome structure, it is critically important to devise experimental approaches which test the predictions of this particular model. One approach would be to deplete cohesin components; however, as discussed in (1.1), our experimental system is not ideal for this type of approach. On the other hand, there are other ways to test the extrusion model. Given the mechanism proposed for TAD formation—extruding a loop until cohesin bumps into CTCF/boundary road blocks—it follows that only two types of loop topologies are possible: stem-loop and unanchored loop. The loop extrusion model, as currently conceived, can’t account for the two cases in this study in which the reporter on the wrong side of the *homie* boundary from the *eve* locus is activated by the *eve* enhancers. In contrast, our findings are completely consistent with orientation-specific boundary:boundary pairing.

B) In the loop-extrusion model, cohesin embraces both of the extruded chromatin fibers, transiently bringing them into close proximity. As far as we know, there have been no (high resolution) experiments that have actually detected these extruding cohesin complexes during TAD formation. In order to have a chance of observing the expected signatures of extruding cohesin complexes, one would need a system in which TADs are being formed. As described in the text, this is why we used MicroC to analyze TADs in NC14 embryos. We do not detect the signature stripes that would be predicted (see Figure 2–figure supplement 2) by the current version of the loop extrusion model.

C) Reporter expression in the different -142 kb transgenes provides only an indirect test of the loop extrusion and boundary:boundary pairing models for TAD formation. The reporter expression results need to be confirmed by directly analyzing the pattern of physical interactions in each instance. While we were able to detect contacts between the transgenes and *eve* in our 2016 paper, the 3C experiments provided no information beyond that. By contrast, the MicroC experiments in the current paper give high resolution maps of the physical contacts between the transgene and the *eve* TAD. The physical contacts track completely with reporter activity. Moreover, just as is the case for reporter activity, the observed physical interactions are inconsistent with the loop extrusion model.

D) Genetic studies in Muller et al. (1999) and imaging in Vazquez et al. (2006) suggested that more than two boundaries can participate in pairing interactions. Consistent with these earlier observations, viewpoint analysis indicates the transgene *homie* interacts with both *eve* boundaries. While this could be explained by transgene *homie* alternating between *nhomie* and *homie* in the *eve* locus, this would require the remodeling of the *eve* TAD each time the pairing interaction switched between the three boundary elements. Moreover, two out of the three possible pairing combinations would disrupt the *eve* TAD, generating an unanchored loop (c.f., the *lambda* DNA TAD in Ke et al. 2024). However, the MicroC profile of the *eve* TAD is unaffected by transgenes carrying the *homie* boundary. This would suggest that like *Mcp*, the pairing interactions of *homie* and *nhomie* might not be exclusively pairwise. In this context is interesting to compare the contact profiles of the *lar* meta-loop shown in Author response image 10 with the different -142 kb *homie* inserts. Unlike the *homie* element at -142 kb, there is clearly only a single point of contact between the blue and purple boundaries.

E) Chen et al. (2018) used live imaging to link physical interactions between a *homie* containing transgene inserted at -142 kb and the *eve* locus to reporter activation by the *eve* enhancers. They found that the reporter was activated by the *eve* enhancers only when it was in “close proximity” to the *eve* gene. “Close proximity” in this case was 331 nM. This distance is equivalent to ~1.1 kb of linear duplex B form DNA, or ~30 nucleosome core particles lined up in a row. It would not be possible to ligate two DNAs wrapped around nucleosome core particles that are located 330 nM apart in a fixed matrix. Since our MicroC experiments were done on embryos in which the gene is silent in the vast majority of cells, it is possible that the *homie* transgene only comes into close enough proximity for transgene nucleosome: *eve* nucleosome ligation events when the *eve* gene is off. Alternatively, and clearly more likely, distance measurements using imaging procedures that require dozens of fluorescent probes may artificially inflate the distance between sequences that are actually close enough for enzymatic ligation.

F) The findings reported in Goel et al. (2023) indicate that mammalian TADs don’t require cohesin activity; however, the authors do not provide an alternative mechanism for TAD formation/stability. Here we have suggested a plausible mechanism.

The authors make no attempt to dissect the mechanism of this process by modifying extrusion components directly.

**(2.7)** See point (1.1).

Some discussion of Rollins et al. on the discovery of Nipped-B and its role in enhancer-promoter communication should also be made to reconcile their conclusions in the proposed absence of extrusion events.

**(2.8)** The reason why reducing *nipped-B* activity enhances the phenotypic effects of *gypsy*-induced mutations is not known at this point; however, the findings reported in Rollins et al. (1999) would appear to argue against an extrusion mechanism for TAD formation.

Given what we know about enhancer blocking and TADs, there are two plausible mechanisms for how the Su(Hw) element in the *gypsy* transposon blocks enhancer-promoter interactions in the *gypsy*-induced mutants studied by Rollins et al. First, the Su(Hw) element could generate two new TADs through pairing interactions with boundaries in the immediate neighborhood. This would place the enhancers in one TAD and the target gene in another TAD. Alternatively, the studies of Sigrist and Pirrotta (1997), as well as several publications from Victor Corces’ lab, raise the possibility that the Su(Hw) element in *gypsy*-induced mutations is pairing with *gypsy* transposons inserted elsewhere in the genome. This would also isolate enhancers from their target genes. In either case, the loss of *nipped-B* activity increases the mutagenic effects of Su(Hw) element presumably by strengthening its boundary function. If this is due to a failure to load cohesin on to chromatin, this would suggest that cohesin normally functions to *weaken* the boundary activity of the Su(Hw) element, i.e., disrupting the ability of Su(Hw) elements to interact with either other boundaries in the neighborhood or with themselves. Were this a general activity of cohesin (to *weaken* boundary activity), one would imagine that cohesin normally functions to disrupt TADs rather than generate/stabilize TADs.

An alternative model is that Nipped-B (and thus cohesin) functions to stabilize enhancer-promoter interactions within TADs. In this case, loss of Nipped-B would result in a destabilization of the weak enhancer-promoter interactions that can still be formed when *gypsy* is located between the enhancer and promoter. In this model, the loss of these weak interactions in *nipped-b* mutants would appear to increase the “blocking” activity of the *gypsy* element. However, this alternative model would also provide no support for the notion that Nipped-B and cohesin function to promote TAD formation.

**Reviewer #3 (Public Review):**
Bing et al. attempt to address fundamental mechanisms of TAD formation in *Drosophila* by analyzing gene expression and 3D conformation within the vicinity of the eve TAD after insertion of a transgene harboring a Homie insulator sequence 142 kb away in different orientations. These transgenes along with spatial gene expression analysis were previously published in Fujioka et al. 2016, and the underlying interpretations regarding resulting DNA configuration in this genomic region were also previously published. This manuscript repeats the expression analysis using smFISH probes in order to achieve more quantitative analysis, but the main results are the same as previously published. The only new data are the Micro-C and an additional modeling/analysis of what they refer to as the 'Z3' orientation of the transgenes. The rest of the manuscript merely synthesizes further interpretation with the goal of addressing whether loop extrusion may be occurring or if boundary:boundary pairing without loop extrusion is responsible for TAD formation. The authors conclude that their results are more consistent with boundary:boundary pairing and not loop extrusion; however, most of this imaging data seems to support both loop extrusion and the boundary:boundary models. This manuscript lacks support, especially new data, for its conclusions.

**(3.1)** The new results/contributions of our paper are described in (2.6) above. Although there are (two) *homie* transgene configurations that give expression patterns that would be consistent with the loop extrusion model, this is not quite the same as strong evidence supporting loop extrusion. On the contrary, key aspects of the expression data are *entirely inconsistent* with loop extrusion, and they thus rule out the possibility that loop extrusion is sufficient to explain the results. Moreover, the conclusions drawn from the expression patterns of the four transgenes are back up by the MicroC contact profiles—profiles that are not consistent with the loop extrusion model. Further, as documented above, loop extrusion is not only unable to explain the findings reported in this manuscript, but also the results from a large collection of published studies on fly boundaries. Since all of these boundaries function in TAD formation, there is little reason to think that loop extrusion makes a significant contribution at the TAD level in flies. Given the results reported by Goel et al. 2023, one might also have doubts about the role of loop extrusion in the formation/maintenance of mammalian TADs.

To further document these points, we’ve included a new figure (Fig. 9) that shows two meta-loops. Like the loops seen for *homie*-containing transgenes inserted at -142 kb, meta-loops are formed by the pairing of distant fly boundaries. As only two boundaries are involved, the resulting loop topologies are simpler than those generated when transgene *homie* pairs with *nhomie* and *homie* in the *eve* locus. The meta-loop in panel B is a stem-loop. While a loop with this topology could be formed by loop extrusion, cohesion would have to break through dozens of intervening TAD boundaries and then somehow know to come to a halt at the blue boundary on the left and the purple boundary on the right. However, none of the mechanistic studies on either cohesin or the mammalian CTCF roadblocks have uncovered activities of either the cohesin complex or the CTCF roadblocks that could explain how cohesin would be able to extrude hundreds of kb and ignore dozens of intervening roadblocks, and then stop only when it encounters the two boundaries that form the *beat-IV* meta-loop. The meta-loop in panel A is even more problematic in that it is a circle-loop--a topology that can’t be generated by cohesin extruding a loop until comes it into contact with CTCF roadblocks on the extruded strands.

Furthermore, there are many parts of the manuscript that are difficult to follow. There are some minor errors in the labelling of the figures that if fixed would help elevate understanding. Lastly, there are several major points that if elaborated on, would potentially be helpful for the clarity of the manuscript.Major Points:(1) The authors suggest and attempt to visualize in the supplemental figures, that loop extrusion mechanisms would appear during crosslinking and show as vertical stripes in the micro-C data. In order to see stripes, a majority of the nuclei would need to undergo loop extrusion at the same rate, starting from exactly the same spots, and the loops would also have to be released and restarted at the same rate. If these patterns truly result from loop extrusion, the authors should provide experimental evidence from another organism undergoing loop extrusion.

**(3.2)** We don’t know of any reports that actually document cohesion extrusion events that are forming TADs (TADs as defined in our paper, in the RCMC experiments of Goel et al. (2023), Author response image 1 in response (1.1), or in the high-resolution images from the MicroC data of Krietenstein et al. (2020) and Hsieh et al. (2020) that are shown in Author response image 9 D and H in response (2.1)). However, an extruding cohesin complex would be expected to generate vertical and 45^o^ stripes, because it transiently brings together the two chromatin strands, as illustrated by the broken zipper in Figure 2–figure supplement 2 of our paper. While stripes generated by cohesin forming a TAD have not to our knowledge ever been observed, Author response image 1 above (from Fig. 4 in Goel et al. 2023) shows 45^o^ stripes outlining TADs and connecting neighboring TADs. These stripes are visible with or without RAD21.

In some versions of the loop-extrusion model, cohesin extrudes a loop until it comes to a halt at both boundaries, where it then remains, holding the loop together. In this model, the extrusion event would occur only once per cell cycle. This is the reason we selected NC14 embryos, as they should provide by far the best opportunity to visualize cohesindependent TAD formation. However, the expected stripes generated by cohesin embrace of both strands of the extruding loop were not evident. Other, newer versions of the loop extrusion model are much more dynamic: cohesin extrudes the loop, coming to a halt at the two boundaries, but either doesn’t remain stably bound, or breaks through one or both boundaries. In the former case, the TAD needs to be re-established by another extrusion event, while in the latter case, LDC domains are generated. In this dynamic model, we should also be able to observe vertical and 45^o^ stripes (or stripes leaning to one side or the other of the loading site, if the extrusion rates aren’t equal on both fibers) in NC14 embryos, corresponding to the formation of TADs and LDC domains. However, we don’t.

(2) On lines 311-314, the authors discuss that stem-loops generated by cohesin extrusion would possibly be expected to have more next-next-door neighbor contacts than next-door neighbor contacts and site their models in Figure 1. Based on the boundary:boundary pairing models in the same figure would the stem-loops created by head-to-tail pairing also have the same phenotype? Making possible enrichment of next-next-door neighbor contacts possible in both situations? The concepts in the text are not clear, and the diagrams are not well-labeled relative to the two models.

**(3.3)** Yes, we expect that stem-loops formed by cohesin extrusion or head-to-tail pairing would behave in a similar manner. They could be stem-loops separated by unanchored loops as shown in Fig. 1B and E. Alternatively, adjacent loops could be anchored to each other (by cohesin/CTCF road blocks or by pairing interactions) as indicated in Fig. 1C and F. In stem-loops generated either by cohesin extrusion or by head-to-tail pairing, next-next door neighbors should interact with each other, generating a plume above the volcano triangle. In the case of circle-loops, the volcano triangle should be flanked by clouds that are generated when the TAD bumps into both next-door neighbors. In the accompanying paper, we test this idea by deleting the *nhomie* boundary and then (a) inserting *nhomie* back in the reverse orientation, or (b) by inserting *homie* in the forward orientation. The MicroC patterns fit with the predictions that were made in this paper.

(3) The authors appear to cite Chen et al., 2018 as a reference for the location of these transgenes being 700nM away in a majority of the nuclei. However, the exact transgenes in this manuscript do not appear to have been measured for distance. The authors could do this experiment and include expression measurements.

**(3.4)** The transgenes used in Chen et al. are modified versions of a transgene used in Fujioka et al. (2016) inserted into the same attP site. When we visualize reporter transcription in NC14 embryos driven by the *eve* enhancers using smFISH, HCR-FISH or DIG, only a subset of the nuclei at this stage are active. The number of active nuclei we detect is similar to that observed in the live imaging experiments of Chen et al. The reason we cited Chen et al. (2018) was that they found that proximity was a critical factor in determining whether the reporter was activated or not in a given nucleus. The actual distance they measured wasn’t important. Moreover, as we discussed in response (2.6) above, there are good reasons to think that the “precise” distances measured in live imaging experiments like those used in Chen et al. are incorrect. However, their statements are certainly correct if one considers that a distance of ~700 nM or so is “more distant” relative to a distance of ~300 nM or so, which is “closer.”

(4) The authors discuss the possible importance of CTCF orientation in forming the roadblock to cohesin extrusion and discuss that Homie orientation in the transgene may impact Homie function as an effective roadblock. However, the Homie region inserted in the transgene does not contain the CTCF motif. Can the authors elaborate on why they feel the orientation of Homie is important in its ability to function as a roadblock if the CTCF motif is not present? Trans-acting factors responsible for Homie function have not been identified and this point is not discussed in the manuscript.

We discussed the “importance” of CTCF orientation in forming roadblocks because one popular version of the cohesin loop extrusion/CTCF roadblock model postulates that CTCF must be oriented so that the N-terminus of the protein is facing towards the oncoming cohesin complex, otherwise it won’t be able to halt extrusion on that strand. When *homie* in the transgene is pointing towards the *eve* locus, the reporter on the other side (farther from *eve*) is activated by the *eve* enhancers. One possible way to explain this finding (if one believes the loop extrusion model) is that when *homie* is inverted, it can’t stop the oncoming cohesin, and it runs past the *homie* boundary until it comes to a stop at a properly oriented boundary farther away. In this case, the newly formed loop would extend from the boundary that stopped cohesin to the *homie* boundary in the *eve* locus, and would include not only the distal reporter, but also the proximal reporter. If both reporters are in the same loop with the *eve* enhancers (which they would have to be given the mechanism of TAD formation by loop extrusion), both reporters should be activated. They are not.

For the boundary pairing model, the reporter that will be activated will depend upon the orientation of the pairing interaction—which can be either head-to-head or head-to-tail (or both: see discussion of LBC elements in (2.1)). For an easy visualization of how the orientation of pairing interactions is connected to the patterns of interactions between sequences neighboring the boundary, please look at Fig. 9. This figure shows two different meta-loops. In panel A, head-to-head pairing of the blue and purple boundaries brings together, on the one hand, sequences upstream of the blue and purple boundary, and on the other hand, sequences downstream of the blue and purple boundaries. In the circle loop configuration, the resulting rectangular boxes of enhanced contact are located to the upper left and lower right of the contact map. In panel B, the head-to-tail pairing of the blue and purple boundary changes how sequences upstream and downstream of the blue and purple boundaries interact with each other. Sequences upstream of the blue boundary interact with sequences downstream of the purple boundary, and this gives the rectangular box of enhanced interactions on the top right and lower left. Sequences downstream of the blue boundary interact with sequences upstream of the purple boundary, and this gives the rectangular box of enhanced contact on the lower left.

CTCF: Our analysis of the *homie* boundary suggests that CTCF contributes little to its activity. It has an Su(Hw) recognition sequence and CP190 “associated” sequence. Mutations in both compromise boundary activity (blocking and -142 kb pairing). Gel shift experiments and ChIP data indicate there are half a dozen or more additional proteins that associate with the 300 bp *homie* fragment used in our experiments.

Orientation of CTCF or other protein binding sites: The available evidence suggests that orientation of the individual binding sites is not important (Kyrchanova et al. 2016; Lim et al. 2018). Instead, it is likely that the order of binding sites affects function.

(5) The imaging results seem to be consistent with both boundary:boundary interaction and loop extrusion stem looping.

It is not clear whether the reviewer is referring to the different patterns of reporter expression—which clearly don’t fit with the loop-extrusion model in the key cases that distinguish the two models—or the live imaging experiments in Chen et al. (2018).

(6) The authors suggest that the eveMa TAD could only be formed by extrusion after the breakthrough of Nhomie and several other roadblocks. Additionally, the overall long-range interactions with Nhomie appear to be less than the interactions with endogenous Homie (Figures 7, 8, and Figure 6–figure supplements 1 and 2). Is it possible that in some cases boundary:boundary pairing is occurring between only the transgenic Homie and endogenous Homie and not including Nhomie?

Yes, it is possible. On the other hand, the data that are currently available supports the idea that transgene *homie* usually interacts with endogenous *homie* and *nhomie* at the same time. This is discussed in (2.6) D above. The viewpoints indicate that crosslinking occurs more frequently to *homie* than to *nhomie*. This could indicate that when there are only pairwise interactions, these tend to be between *homie* and *homie*. Alternatively, this could also be explained by a difference in relative crosslinking efficiency.

(7) In Figure 4E, the GFP hebe expression shown in the LhomieG Z5 transgenic embryo does not appear in the same locations as the LlambdaG Z5 control. Is this actually hebe expression or just a background signal?

The late-stage embryos shown in E are oriented differently. For *GlambdaL*, the embryo is oriented so that *hebe*-like reporter expression on the ventral midline is readily evident. However, this orientation is not suitable for visualizing *eve* enhancer-dependent expression of the reporters in muscle progenitor cells. For this reason, the stage 12-16 *GeimohL* embryo in E is turned so that the ventral midline isn’t readily visible in most of the embryo. As is the case in NC14 embyros, the *eve* enhancers drive *lacZ* but not *gfp* expression in the muscle progenitor cells.

(8) Figure 6- The LhomieG Z3 (now *LeimohG*) late-stage embryo appears to be showing the ventral orientation of the embryo rather than the lateral side of the embryo as was shown in the previous figure. Is this for a reason? Additionally, there are no statistics shown for the Z3 transgenic images. Were these images analyzed in the same way as the Z5 line images?

The *LeimohG* embryo was turned so that the *hebe* enhancer-dependent expression of *lacZ* is more clearly visible. While the *eve* enhancer-dependent expression of *lacZ* in the muscle progenitor cells isn’t visible with this orientation, *eve* enhancer-dependent expression in the anal plate is.

(9) Do the Micro-C data align with the developmental time points used in the smFISH probe assays?

The MicroC data align with the smFISH images of older embryos: stages 14-16.

**Recommendations for the authors:**

**Reviewer #1 (Recommendations For The Authors):**
This was a difficult paper to review. It took me several hours to understand the terminology and back and forth between different figures to put it together. It might be useful to put the loop models next to the MicroC results and have a cartoon way of incorporating which enhancers are turning on which reporters.I also found the supercoiled TAD models in Figure 1 not useful. These plectoneme-type of structures likely do not exist, based on the single-cell chromosome tracing studies, and the HiC structures not showing perpendicular to diagonal interactions between the arms of the plectonemes.

We wanted to represent the TAD as a coiled 30nM fiber, as they are not likely to resemble the large loops like those shown in Fig. 1A, D, and G. One striking feature of most TADs is that sequences within the TADs tend to bump into each other and get crosslinked. For some of the larger TADs (without internal substructures) the frequency of contact can drop off a bit at larger distance, otherwise the contract frequency within TADs seems pretty uniform. So one might imagine that a TAD resembles a plasmid DNA, where all sequences can come into contact by some type of sliding mechanism.

There are no stripes emerging from homies, which is consistent with the pairing model, but there seem to be stripes from the eve promoter. I think these structures may be a result of both the underlying loop extruders + pairing elements.

There are internal structures in the *eve* TAD that link the upstream region of the *eve* promoter to the *eve* PRE and sequences in *nhomie*. All three of these sequences are bound by LBC. Each of the regulatory domains in BX-C also have LBC elements and, as shown in Author response image 3, you can see stripes connecting some of these LBC elements to each other. Since the stripes that Goel et al. (2023) observed in their RCMC analysis of *Ppm1g* didn’t require cohesin, how these stripes are generated (active: e.g, a chromatin remodeler or passive: e.g., the LBC complex has non-specific DNA binding activity that can be readily crosslinked as the chromatin fiber slides past) isn’t clear.

The authors say there are no TADs that have "volcano plumes" but the leftmost TAD TA appears to have one. What are the criteria for calling the plumes? I am also not clear why there is a stripe off the eve volcano. It looks like homie is making a "stripe" loop extrusion type of interaction with the next TAD up. Is this maybe cohesin sliding off the left boundary?

The reviewer is correct, the left-most TAD TA appears to have a plume. We mentioned TA seems to have a plume in the original text, but it was inadvertently edited out.

Two different types of TAD«»TAD interactions are observed. In the case of *eve*, the TADs to either side of *eve* interact more frequently with each other than they do with *eve*. This generates a “plume” above the *eve* volcano triangle. The TADs that comprise the *Abd-B* regulatory domains (see Author response image 3) are surrounded by clouds of diminishing intensity. Clouds at the first level represent interactions with both next-door neighbors; clouds at the second level represent interactions with both next-next-door neighbors; clouds at the third level represent interactions with next-next-next door neighbors. The *Abd-B* TADs are close to the same size, so that interactions with neighbors are relatively simple. However, this is not always the case. When there are smaller TADs near larger TADs the pattern of interaction can be quite complicated. An example is indicated by the red bar in Author response image 9.

The authors state "In the loop-extrusion model, a cohesin complex initiating loop extrusion in the eve TAD must break through the nhomie roadblock at the upstream end of the eve TAD. It must then make its way past the boundaries that separate eve from the attP site in the hebe gene, and come to a halt at the homie boundary associated with the lacZ reporter." Having multiple loops formed by cohesin would also bring in the 142kb apart reporter and homie. Does cohesin make 140 kb long loops in flies?

A mechanism in which cohesin brings the reporter close to the *eve* TAD by generating many smaller loops (which would be the intervening TADs) was discussed in (1.2).

Figure 5 title mistakes the transgene used?

This has been fixed.

In figure 6, the orientation of the embryos does not look the same for the late-stage panels. So it was difficult to tell if the eve enhancer was turning the reporter on.

Here, we were focusing mainly on the AP enhancer activation of the reporter, as this is most easily visualized. It should be clear from the images that the appropriate reporter is activated by the AP enhancer for each of the transgene inserts.

It is not clear to me why the GFP makes upstream interactions (from the 4C viewpoint) in GhomileL Z5 but not in LhomieGZ5? Corresponding interactions for Figure 6–figure supplements 1 and 2 are not the same. That is, LacZ in the same place and with the same homie orientation does not show a similar upstream enrichment as the GFP reporter does.

We are uncertain as to whether we understand this question/comment. In GhomieL Z5 (now *GhomieL*), the *lacZ* reporter is on the *eve* side of the *homie* boundary, while *gfp* is on the *hebe* enhancer side of the *homie* boundary. Since *homie* is pointing away from *gfp*, pairing interactions with *homie* and *nhomie* in the *eve* locus bring the *eve* enhancers in close proximity with the *gfp* reporter. This is what is seen in Fig. 7, panel D, lower trace. In LhomieG Z5 (now *GeimohL*), the *lacZ* reporter is again on the *eve* side of the *homie* boundary, while *gfp* is on the *hebe* enhancer side of the *homie* boundary. However, in this case, *homie* is inverted so that it points away from *lacZ* (and towards *gfp*). In this orientation, pairing brings the *lacZ* reporter into contact with the *eve* enhancers. This is what is seen in the upper trace in Fig. 7, panel D.

The orientation of the transgene is switched in Figure 6–figure supplements 1 and 2. For these “Z3” transgenes (now called *LeimohG* and *LhomieG*), the *gfp* reporter is on the *eve* side of *homie*, while the *lacZ* reporter is on the *hebe* enhancer side of *homie*. The interactions between the reporters and *eve* are determined by the orientation of *homie* in the transgene. When *homie* is pointing away from *gfp* (as in *LeimohG*), *gfp* is activated, and that is reflected in the trace in Figure 6–figure supplement 1. When *homie* is pointing away from *lacZ*, *lacZ* is activated, and this is reflected (though not as cleanly as in other cases) in the trace in Figure 6–figure supplement 2.

I did not see a data availability statement. Is the data publicly available? The authors also should consider providing the sequences of the insertions, or provide the edited genomes, in case other researchers would like to analyze the data.

The data have now been deposited.

**Reviewer #3 (Recommendations For The Authors):**
Minor Points:(1) There is an inconsistency in the way that some of the citations are formatted. Some citations have 'et al' italicized while others do not. It seems to be the same ones throughout the manuscript. Some examples: Chetverina et al 2017, Chetverina et al 2014, Cavalheiro et al 2021, Kyrchanova et al 2008a, Muravyova et al 2001.

This has been fixed.

(2) Pita is listed twice in line 48.

This has been fixed.

(3) Line 49, mod(mdg4)67.2 is written just as mod(mdg4). The isoform should be indicated.

This refers to all Mod isoforms.

(4) Homie and Nhomie are italicized throughout the manuscript and do not need to be.

This is the convention used previously.

(5) The supplemental figure captions 1 and 2 in the main document are ordered differently than in the supplemental figures file. This caused it to look like the figures are being incorrectly cited in lines 212-214 and 231-232.

This has been fixed.

(6) Is the correct figure being cited in line 388-389? The line cites Figure 6E when mentioning LlambdaG Z5; however, LlambdaG Z5 is not shown in Figure 6.

This has been fixed.

(7) Section heading 'LhomieG Z5 and GhomieL Z5' could be renamed for clarity. GhomieL Z5 results are not mentioned until the next section, named 'GhomieL Z5'.

This has been fixed.

(8) Can the authors provide better labeling for control hebe expression? This would help to determine what is hebe expression and what is background noise in some of the embryos in Figures 4-6.

Author response image 12 shows expression of the *lacZ* reporter in *GiemohL* and *GlambdaL*. For the *GlambdaL* transgene, the *hebe* enhancers drive *lacZ* expression in 12-16 hr embryos. Note that *lacZ* expression is restricted to a small set of quite distinctive cells along the ventral midline. *lacZ* is also expressed on the ventral side of the *GiemohL* embryo (top panel). However, their locations are quite different from those of the *lacZ*-positive cells in the *GlambdaL* transgene embryo. These cells are displaced from the midline, and are arranged as pairs of cells in each hemisegment, locations that correspond to *eve*-expressing cells in the ventral nerve cord. The *eve* enhancers also drive *lacZ* expression elsewhere in the *GiemohL* embryo, including the anal plate and dorsal muscle progenitor cells (seen most clearly in the lower left panel).

**Author response image 12. sa4fig12:** *lacZ* expression in *GiemohL* and *GlambdaL* embryos.

(9) The Figure 5 title is labeled with the wrong transgene.

This has been fixed.

(10) Heat map scales are missing for Figures 7 and Figure 6–figure supplements 1 and 2.

This has been fixed.

(11) Did the authors check if there was a significant difference in the expression of GFP and lacZ from lambda control lines to the Homie transgenic lines?

Yes. Statistical analysis was added in Table Supplemental 1.

(12) The Figure 7 title references that these are Z3 orientations, however, it is Z5 orientations being shown.

This has been fixed.

(13) The virtual 4C data should include an axis along the bottom of the graphs for better clarity. An axis is missing in all 4C figures.

This has been fixed.

References for Author Response to Reviewers

Bantignies F, Grimaud C, Lavrov S, Gabut M, Cavalli G. 2003. Inheritance of Polycomb-dependent chromosomal interactions in *Drosophila*. Genes Dev. 17(19):2406-2420.

Batut PJ, Bing XY, Sisco Z, Raimundo J, Levo M, Levine MS. 2022. Genome organization controls transcriptional dynamics during development. Science. 375(6580):566-570.

Bonchuk A, Boyko K, Fedotova A, Nikolaeva A, Lushchekina S, Khrustaleva A, Popov V, Georgiev P. 2021. Structural basis of diversity and homodimerization specificity of zinc-finger-associated domains in *Drosophila*. Nucleic Acids Res. 49(4):2375-2389.

Bonchuk AN, Boyko KM, Nikolaeva AY, Burtseva AD, Popov VO, Georgiev PG. 2022. Structural insights into highly similar spatial organization of zinc-finger associated domains with a very low sequence similarity. Structure. 30(7):1004-1015.e1004.

Chen H, Levo M, Barinov L, Fujioka M, Jaynes JB, Gregor T. 2018. Dynamic interplay between enhancer–promoter topology and gene activity. Nat Genet. 50(9):1296.

Fedotova AA, Bonchuk AN, Mogila VA, Georgiev PG. 2017. C2H2 zinc finger proteins: The largest but poorly explored family of higher eukaryotic transcription factors. Acta Naturae. 9(2):47-58.

Foe VE. 1989. Mitotic domains reveal early commitment of cells in *Drosophila* embryos. Development. 107(1):1-22.

Fujioka M, Mistry H, Schedl P, Jaynes JB. 2016. Determinants of chromosome architecture: Insulator pairing in *cis* and in *trans*. PLoS Genet. 12(2):e1005889.

Galloni M, Gyurkovics H, Schedl P, Karch F. 1993. The *bluetail* transposon: Evidence for independent cis‐regulatory domains and domain boundaries in the bithorax complex. The EMBO Journal. 12(3):1087-1097.

Goel VY, Huseyin MK, Hansen AS. 2023. Region capture micro-c reveals coalescence of enhancers and promoters into nested microcompartments. Nat Genet. 55(6):1048-1056.

Hsieh TS, Cattoglio C, Slobodyanyuk E, Hansen AS, Rando OJ, Tjian R, Darzacq X. 2020. Resolving the 3D landscape of transcription-linked mammalian chromatin folding. Mol Cell. 78(3):539-553.e538.

Ke W, Fujioka M, Schedl P, Jaynes JB. 2024. Stem-loop and circle-loop TADs generated by directional pairing of boundary elements have distinct physical and regulatory properties. eLife 2024;13:RP94114*.* DOI: https://doi.org/10.7554/eLife.94114.

Krietenstein N, Abraham S, Venev SV, Abdennur N, Gibcus J, Hsieh TS, Parsi KM, Yang L, Maehr R, Mirny LA et al. 2020. Ultrastructural details of mammalian chromosome architecture. Mol Cell. 78(3):554-565.e557.

Kyrchanova O, Ibragimov A, Postika N, Georgiev P, Schedl P. 2023. Boundary bypass activity in the *Abdominal-B* region of the *Drosophila* bithorax complex is position dependent and regulated. Open Biol. 13(8):230035.

Kyrchanova O, Kurbidaeva A, Sabirov M, Postika N, Wolle D, Aoki T, Maksimenko O, Mogila V, Schedl P, Georgiev P. 2018. The bithorax complex *iab-7* Polycomb response element has a novel role in the functioning of the *Fab-7* chromatin boundary. PLoS Genet. 14(8):e1007442.

Kyrchanova O, Mogila V, Wolle D, Deshpande G, Parshikov A, Cleard F, Karch F, Schedl P, Georgiev P. 2016. Functional dissection of the blocking and bypass activities of the *Fab-8* boundary in the *Drosophila* bithorax complex. PLoS Genet. 12(7):e1006188.

Kyrchanova O, Sabirov M, Mogila V, Kurbidaeva A, Postika N, Maksimenko O, Schedl P, Georgiev P. 2019a. Complete reconstitution of bypass and blocking functions in a minimal artificial *Fab-7* insulator from *Drosophila* bithorax complex. Proceedings of the National Academy of Sciences.201907190.

Kyrchanova O, Wolle D, Sabirov M, Kurbidaeva A, Aoki T, Maksimenko O, Kyrchanova M, Georgiev P, Schedl P. 2019b. Distinct elements confer the blocking and bypass functions of the bithorax *Fab-8* boundary. Genetics. Genetics.302694.302019.

Li H-B, Muller M, Bahechar IA, Kyrchanova O, Ohno K, Georgiev P, Pirrotta V. 2011. Insulators, not Polycomb response elements, are required for long-range interactions between Polycomb targets in *Drosophila melanogaster*. Mol Cell Biol. 31(4):616-625.

Li X, Tang X, Bing X, Catalano C, Li T, Dolsten G, Wu C, Levine M. 2023. GAGA-associated factor fosters loop formation in the *Drosophila* genome. Mol Cell. 83(9):1519-1526.e1514.

Lim B, Heist T, Levine M, Fukaya T. 2018. Visualization of transvection in living *Drosophila* embryos. Mol Cell. 70(2):287-296. e286.

Link N, Kurtz P, O'Neal M, Garcia-Hughes G, Abrams JM. 2013. A p53 enhancer region regulates target genes through chromatin conformations in cis and in trans. Genes Dev. 27(22):2433-2438.

Mohana G, Dorier J, Li X, Mouginot M, Smith RC, Malek H, Leleu M, Rodriguez D, Khadka J, Rosa P et al. 2023. Chromosome-level organization of the regulatory genome in the *Drosophila* nervous system. Cell. 186(18):3826-3844.e3826.

Muller M, Hagstrom K, Gyurkovics H, Pirrotta V, Schedl P. 1999. The *Mcp* element from the *Drosophila melanogaster* bithorax complex mediates long-distance regulatory interactions. Genetics. 153(3):1333-1356.

Postika N, Metzler M, Affolter M, Müller M, Schedl P, Georgiev P, Kyrchanova O. 2018. Boundaries mediate long-distance interactions between enhancers and promoters in the *Drosophila* bithorax complex. PLoS Genet. 14(12):e1007702.

Rollins RA, Morcillo P, Dorsett D. 1999. *Nipped-b*, a *Drosophila* homologue of chromosomal adherins, participates in activation by remote enhancers in the *cut* and *Ultrabithorax* genes. Genetics. 152(2):577-593.

Samal B, Worcel A, Louis C, Schedl P. 1981. Chromatin structure of the histone genes of *D. melanogaster*. Cell. 23(2):401-409.

Shermoen AW, McCleland ML, O'Farrell PH. 2010. Developmental control of late replication and S-phase length. Curr Biol. 20(23):2067-2077.

Shidlovskii YV, Bylino OV, Shaposhnikov AV, Kachaev ZM, Lebedeva LA, Kolesnik VV, Amendola D, De Simone G, Formicola N, Schedl P et al. 2021. Subunits of the Pbap chromatin remodeler are capable of mediating enhancer-driven transcription in *Drosophila* . Int J Mol Sci. 22(6).

Sigrist CJ, Pirrotta V. 1997. Chromatin insulator elements block the silencing of a target gene by the *Drosophila* Polycomb response element (PRE) but allow *trans* interactions between PREs on different chromosomes. Genetics. 147(1):209-221.

Udvardy A, Schedl P. 1984. Chromatin organization of the 87A7 heat shock locus of *Drosophila melanogaster*. J Mol Biol. 172(4):385-403.

Vazquez J, Muller M, Pirrotta V, Sedat JW. 2006. The *Mcp* element mediates stable long-range chromosome-chromosome interactions in *Drosophila* . Molecular Biology of the Cell. 17(5):2158-2165.

Wolle D, Cleard F, Aoki T, Deshpande G, Schedl P, Karch F. 2015. Functional requirements for *Fab-7* boundary activity in the bithorax complex. Mol Cell Biol. 35(21):3739-3752.